# Inferring language dispersal patterns with velocity field estimation

Sizhe Yang[1], Xiaoru Sun [2,3], Li Jin [1,2] ✉ & Menghan Zhang [4,5] ✉

Reconstructing the spatial evolution of languages can deepen our understanding of the demic diffusion and cultural spread. However, the phylogeographic approach that is frequently used to infer language dispersal patterns has limitations, primarily because the phylogenetic tree cannot fully explain the language evolution induced by the horizontal contact among languages, such as borrowing and areal diffusion. Here, we introduce the language velocity field estimation, which does not rely on the phylogenetic tree, to infer language dispersal trajectories and centre. Its effectiveness and robustness are verified through both simulated and empirical validations. Using language velocity field estimation, we infer the dispersal patterns of four agricultural language families and groups, encompassing approximately 700 language samples. Our results show that the dispersal trajectories of these languages are primarily compatible with population movement routes inferred from ancient DNA and archaeological materials, and their dispersal centres are geographically proximate to ancient homelands of agricultural or Neolithic cultures. Our findings highlight that the agricultural languages dispersed alongside the demic diffusions and cultural spreads during the past 10,000 years. We expect that language velocity field estimation could aid the spatial analysis of language evolution and further branch out into the studies of demographic and cultural dynamics.

Over the past 10,000 years, substantial demic diffusions and cultural spreads have occurred among human populations along with the intensification of agricultural techniques[1–6]. They were also accompanied by the origins and dispersals of language families and groups worldwide[3,7–10]. Given that humans are carriers of languages that are in turn carriers of cultures, technical advances in human genetics have enabled us to trace the complex demographic dynamics of different language-speaking populations[11–13]. On the other hand, the history of language evolution can provide striking insights into the origins and spreads of cultural innovations that may not be reflected in archaeological records[11,12]. A synthesis of linguistic, genetic, and archaeological evidence was therefore proposed to comprehensively depict the prehistory of human activities, although evidence from different disciplines is often far from reaching a reasonable consensus[14,15]. The challenge to achieving this synthesis is to establish the spatiotemporal alignment of interdisciplinary evidence drawn from limited data and fragmented historical accounts[3,16–18]. Fortunately, recent advances in human genetic and archaeological studies have shown similar prehistoric pictures of demic and cultural diffusions globally[1–3]. From a linguistic perspective, an increasing number of phylogenetic studies have provided comprehensive temporal evidence of language evolution to infer prehistoric population

[1]State Key Laboratory of Genetic Engineering, Center for Evolutionary Biology, and Collaborative Innovation Center for Genetics and Development, School of Life Sciences, Fudan University, Shanghai 200438, China. [2]Human Phenome Institute, Fudan University, Shanghai 200438, China. [3]Ministry of Education Key Laboratory of Contemporary Anthropology, Department of Anthropology and Human Genetics, School of Life Sciences, Fudan University, Shanghai 200438, China. [4]Institute of Modern Languages and Linguistics, Fudan University, Shanghai 200433, China. [5]Research Institute of Intelligent Complex Systems, Fudan University, Shanghai 200433, China. ✉e-mail: lijin@fudan.edu.cn; mhzhang@fudan.edu.cn

activities[8,10,19,20]. Relying on the estimated language divergence time, these studies have also examined several hypotheses about prehistoric population activities, such as the language/farming dispersal hypothesis[3,8,19,21]. Nevertheless, studying the spatial evolution of languages remains another great challenge in interdisciplinary alignment[3,12,22].

The spatial evolution of languages has been frequently modelled using the phylogeographic approach[7,9,23,24], which consists of two major aspects (Supplementary Fig. 1). The first one is to obtain a phylogenetic tree based on the linguistic traits (e.g., lexicons) to demonstrate the observed linguistic relatedness[7,9,23,24] (Supplementary Fig. 1). Linguistic relatedness is shaped by the diachronic evolution of linguistic traits, which can be represented by branching patterns within the phylogenetic tree[25]. The branching patterns mirror the evolutionary trajectories of linguistic traits in languages after they diverged from their most recent common ancestor (MRCA). For example, the shorter branch linking two languages indicates fewer diachronic distinctions between their linguistic traits, resulting in a higher linguistic relatedness between them[26,27]. Accordingly, the phylogenetic tree can reflect both the observed linguistic relatedness and the diachronic evolutionary trajectories of linguistic traits that shape it (Supplementary Fig. 1). The second aspect is to transform the diachronic evolutionary trajectories of linguistic traits into language dispersal trajectories (Supplementary Fig. 1). This step is implemented by projecting the phylogenetic tree into geographic space based on the correlation between linguistic relatedness among languages and their geographic proximity[7,9,28,29]. With this projection, the branches within the phylogenetic tree in the geographic space thus constitute the language dispersal trajectories (Supplementary Fig. 1)[7,9,30].

Nevertheless, the phylogenetic tree is an ideal model for representing linguistic relatedness that exclusively captures vertical divergence but ignores horizontal contact[31,32]. In linguistic reality, horizontal contact such as language borrowing and areal diffusion can be substantially found in multilingual areas[33]. Moreover, it can also be identified in different-level linguistic systems of lexicon, grammar, and sound[34–37]. On these grounds, the phylogeographic approach would pose some limitations when linguistic relatedness cannot be well interpreted by the family-tree model. Fortunately, recent technical advances in velocity field estimation provide alternative opportunities to circumvent these limitations. The velocity field can be visualised as a collection of arrows estimated by a specific dynamic model[38]. The directions of the arrows constitute a set of continuously changing trajectories, enabling us to outline the spatiotemporal dynamics of natural or social phenomena such as atmospheric circulations[39] (e.g., water vapour transport), biomolecular processes[40] (e.g., RNA transcription), demic diffusions[41] (e.g., human mobility), and cultural spreads[42] (e.g., Neolithic culture propagation).

Noting these advantages, we here introduced a novel computational approach, namely language velocity field estimation (LVF)[43], to infer the language dispersal pattern, including dispersal trajectories and centre (Fig. 1; see details in Methods). Similar to the phylogeographic approach, the LVF also consists of two major aspects but without involving the phylogenetic tree (see details in Supplementary Notes section 1.1). The first is to establish a velocity field to depict the diachronic evolutionary trajectories of linguistic traits that shape the observed linguistic relatedness. This velocity field functions like the phylogenetic tree but additionally captures the attribution of horizontal contact. The second is to project this velocity field into the geographic space based on the correlation between linguistic relatedness and language geography. It resembles the geographic projection of the phylogenetic tree that facilitates outlining the language dispersal trajectories in geographic space. In simulated validations, we verified the computational effectiveness and robustness of our LVF using 1000 simulated datasets where the dispersal patterns are

given[44]. Specifically, the effectiveness of the LVF was validated based on the difference between estimated and given patterns, and its robustness was evaluated under different parametric settings. In empirical applications, we employed this verified LVF to infer the dispersal patterns of four agricultural languages which are Indo-European, Sino-Tibetan, Bantu, and Arawak languages. Subsequently, we investigated the interdisciplinary alignments between the agricultural language dispersals and the known demic and cultural diffusions drawn from genetic and archaeological evidence. To illustrate the methodological advantages of LVF, we additionally conducted comprehensive simulated and empirical model comparisons between the LVF and other prevailing approaches, such as the phylogeographic approach.

## Results

### Overview of the language velocity field estimation

Akin to the phylogeographic approach, the LVF can be used to infer the language dispersal pattern through the diachronic evolution of linguistic traits (Fig. 1; see details in Methods). Its implementation involves two major aspects, corresponding to the two aspects in the computational procedure of the phylogeographic approach but without involving the phylogenetic tree (Supplementary Fig. 1; see details in Supplementary Notes section 1.1).

The first aspect is to establish the diachronic evolutionary trajectories of linguistic traits that determine linguistic relatedness among language samples, which comprises three steps. The first step is to conduct the principal component analysis (PCA)-based distance to represent linguistic relatedness among language samples (Fig. 1a, b). The linguistic relatedness among language samples is represented by their Euclidean distances in the PC space. Specifically, the linguistic traits are rearranged into two principal components (PC1 and PC2) using PCA (Fig. 1b). Accordingly, each language sample can be visualised based on its PC1 and PC2 values in the two-dimensional PC space. In this PC space, a shorter Euclidean distance between two language samples indicates a higher linguistic relatedness resulting from either vertical divergence or horizontal contact.

In parallel to the first step, the second step is to establish a dynamic model that consists of ordinary differential equations to reconstruct the past state of each linguistic trait for each language sample (Fig. 1a, c, d). This model is similar to the covarion model that is widely utilised for modelling linguistic trait evolution[8,45,46] (Fig. 1d1; see details in Methods). By measuring the differences between the present and past states of linguistic traits for each language sample, we can obtain a high-dimensional velocity vector that exhibits how the linguistic traits for each language sample evolved into their current states (Fig. 1d2; see details in Methods and Supplementary Notes section 1.1 and section 1.2). Therefore, we can derive a collection of velocity vectors that constitute a velocity field in high-dimensional space. This high-dimensional velocity field can illustrate the diachronic evolutionary trajectories of linguistic traits in the observed language samples.

The third step is to project the high-dimensional velocity field into the two-dimensional PC space. This process delineates the diachronic evolutionary trajectories of linguistic traits that shape the linguistic relatedness among language samples (Fig. 1e; see details in Methods and Supplementary Notes section 1.1). This projection is implemented by simultaneously mapping the past and present states of linguistic traits for each language sample into the PC space (Fig. 1e1). The velocity vector for each language sample in PC space can be visualised as an arrow connecting the present and past trait states of that language sample scaled by the reconstruction time (Fig. 1e1). This arrow visually illustrates how the linguistic traits in each language sample evolved from the past states into the present states within PC space. Accordingly, the collection of arrows within the PC space presents the diachronic evolutionary trajectories of linguistic traits, which visualises

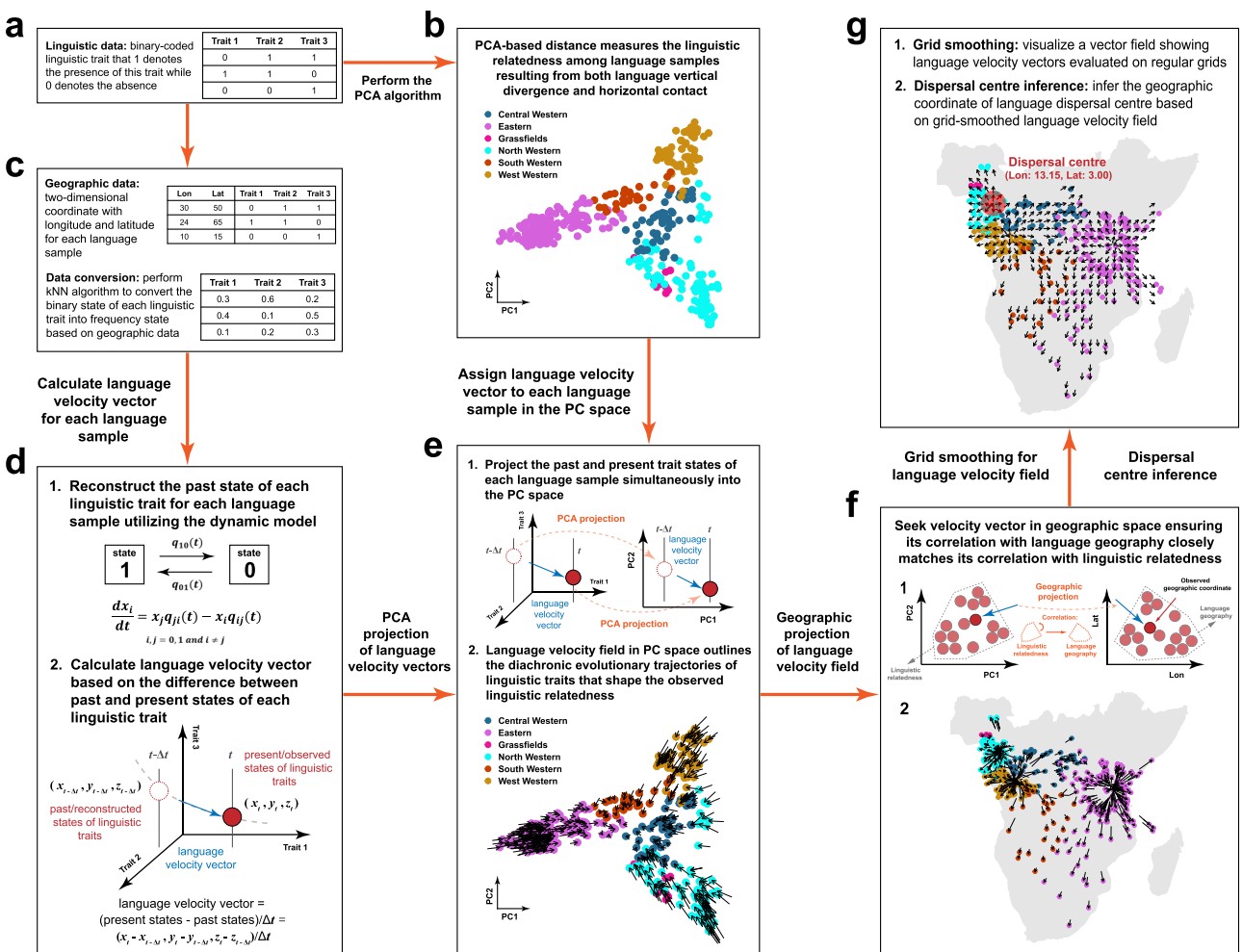

**Fig. 1 | Schematic overview of the LVF for inferring the dispersal trajectories and centre of languages.** The computational procedure of the LVF comprises two major steps. Subfigures (**a**) to (**e**) illustrate the first step, which is to estimate a velocity field within the PC space to outline the diachronic evolutionary trajectories of linguistic traits that shape the observed linguistic relatedness. Subfigures (**f**) to (**g**) illustrate the second step, which is to project the velocity field from PC space into geographic space. Within the velocity field in geographic space, the directions of the velocity vectors compose a set of continuously changing trajectories that delineate from where the language samples diffused to their current locations. These procedures are exemplified using the Bantu language family. Comprehensive insights into the underlying principles and computational steps can be found in the Methods, as well as Supplementary Notes and Supplementary Methods. The grey base world map used in Subfigures (**f**) to (**g**) is generated using the map function of the maps package in R (4.3.1). The Source Data and Codes for generating Fig. 1 are available.

the formation of the observed linguistic relatedness among language samples (Fig. 1e2).

The second aspect is to transform the diachronic evolutionary trajectories of linguistic traits that shape observed linguistic relatedness into language dispersal trajectories (Fig. 1f, g). This is accomplished with kernel projection[40]. This kernel projection enables us to map the velocity vectors from PC space into geographic space based on the correlation between linguistic relatedness and language geography (Fig. 1f; see details in Methods). In particular, the rationale of kernel projection is to estimate the velocity vector for each language sample in geographic space, ensuring its correlation with linguistic relatedness among language samples in PC space can be optimally maintained (Fig. 1f1). The vector directions in geographic space thus constitute a set of trajectories that suggest how and from where (i.e., dispersal trajectories and centre) the language samples dispersed into their current locations (Fig. 1f2; see details in Methods and Supplementary Notes section 1.3). In particular, the geographic location surrounded by velocity vectors that point radially outwards in all directions could be a potential dispersal centre (Fig. 1g; see details in Methods).

## Simulated validations of the language velocity field estimation
To validate the feasibility of our LVF, we sourced 1000 simulated datasets from Wichmann and Rama's work[44] (see details in Supplementary Notes section 2). Each dataset consisted of 20 simulated language samples with 306 binary-coded linguistic traits generated by a given phylogenetic tree. For each dataset, the geographic coordinates of the simulated language samples were generated by applying the random walk model to the phylogenetic tree assigned with a given dispersal centre. In other words, the geographic coordinate of the dispersal centre was provided in each dataset. Accordingly, these simulated datasets can serve as benchmarks or baselines for the validation of LVF.

In this study, we validated the effectiveness of the LVF by comparing the differences between the given and inferred dispersal centres in longitude and latitude under different parametric settings. In practice, the simulated results showed that the inferred dispersal centres under different parametric settings were not significantly different from the given dispersal centre in longitude and latitude (Supplementary Fig. 2; see details in Methods and Supplementary Notes section 2). This suggests the high effectiveness of the LVF in inferring the language dispersal pattern. On the other hand,

we also validated the robustness of the LVF by measuring the cosine similarities among velocity fields that were estimated under different parametric settings. The simulated results showed that there should be no significant difference among the velocity fields within either high-dimensional or two-dimensional spaces under different parametric settings (Supplementary Fig. 3; see details in Methods and Supplementary Notes section 2). This indicates the high robustness of the LVF in inferring the language dispersal pattern.

## Empirical applications of the language velocity field estimation

We collected four empirical cases of language families and groups that encompass a total of 692 language samples. They are the Indo-European[7], Sino-Tibetan[8], Bantu[9], and Arawak[24] languages (Supplementary Table 1). These languages have been suggested to be associated with the developments and spreads of ancient agricultural or Neolithic cultures[3] (Fig. 2a; see details in Supplementary Discussion section 1). For each case, we applied the LVF to estimate a velocity field in geographic space to illustrate the language dispersal trajectories

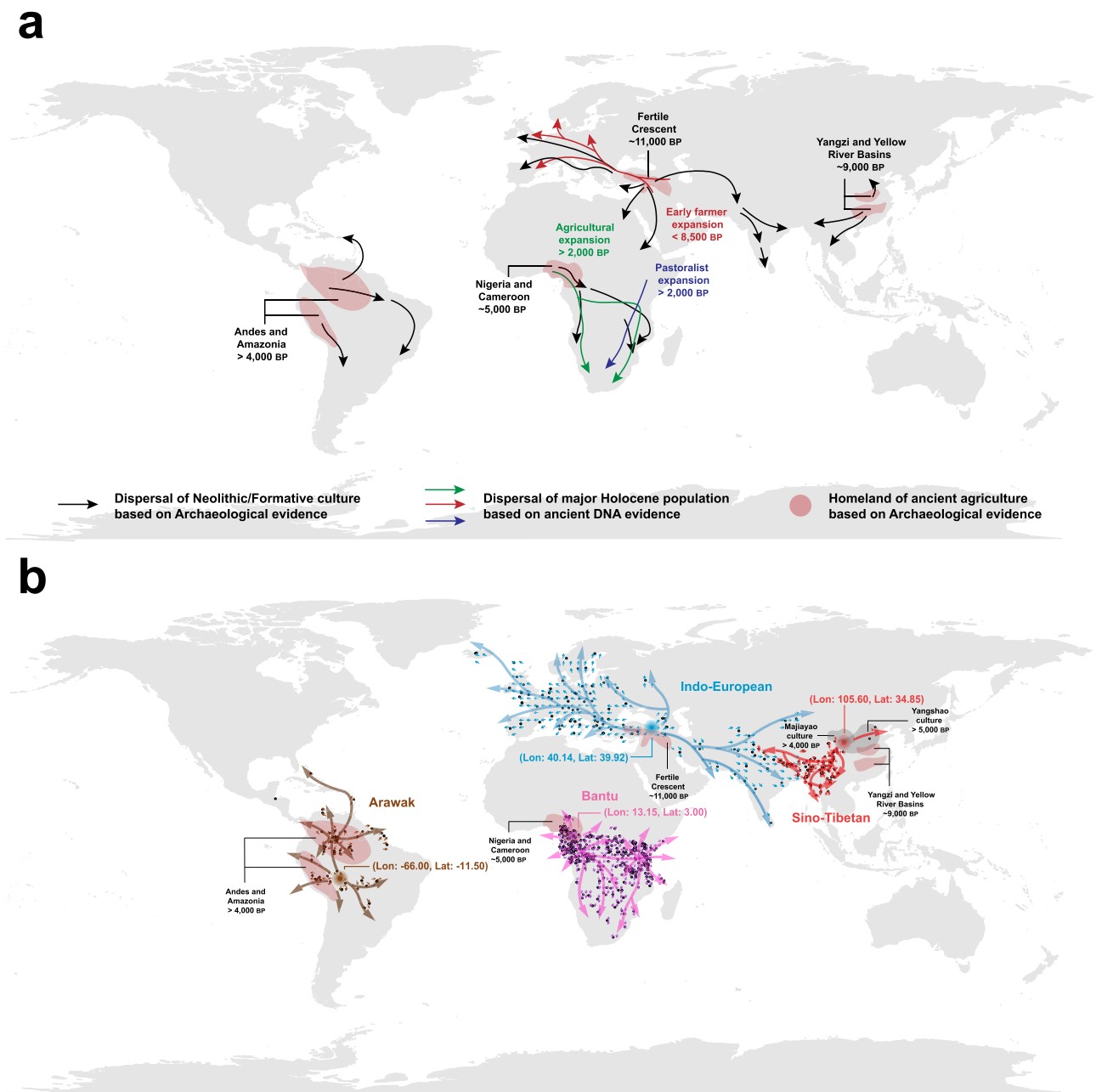

**Fig. 2 | The homelands and dispersals of ancient agriculture, Neolithic cultures, Holocene populations, and language families and groups. a** The homelands of ancient agriculture and the dispersal routes of Neolithic/Formative cultures and Holocene populations proposed by previous studies[2–4] based on archaeological and ancient DNA evidence. The pale red polygon denotes the known ancient agricultural homeland. The black arrow signifies the dispersal trajectory of the Neolithic/Formative culture. The coloured arrow represents the dispersal trajectory of the major Holocene population. **b** The velocity fields of four language families and groups. The coloured dot denotes the geographical position of each observed language sample. The coloured small arrow represents the velocity vector which has been grid-smoothed and normalised for better visualisation. The larger coloured schematic arrow, summarised based on the velocity vectors, renders the general language dispersal trajectory. The pale grey polygon signifies the known geographic range of the Neolithic culture. The coloured concentric circle represents the language dispersal centre inferred by the LVF. The grey base world map is generated using the map function of the maps package in R (4.3.1). The Source Data and Codes for generating Fig. 2 are available.

(Fig. 2b and Supplementary Figs. 4, 5). Specifically, across the Eurasian continent, we observed that Indo-European languages expanded geographically westwards into Europe and eastwards into the Indian peninsula. In Asia, the reconstructed trajectories showed the Sino-Tibetan expansion westwards into the Tibet Plateau, southwards into mainland Southeast Asia, and eastwards into the coastal areas. In Africa, the Bantu language dispersal exhibited a series of eastwards and southwards expansions. In South America, the Arawak languages spread from the Amazon basin to the Caribbean and across the lowlands. Our results of the spatial reconstructions are primarily consistent with previous studies of language evolution[7–9,24,47], and largely favoured by the known evidence of demic diffusions and cultural spreads of Holocene populations (Fig. 2a)[1–3,5,6].

Based on the estimated velocity field in geographic space, we further inferred the dispersal centre for each language case (Fig. 2b and Supplementary Fig. 6 and Supplementary Table 2; see details in Methods and Supplementary Notes section 1.3). Notably, the inferred dispersal centres of these four agricultural languages were adjacent to the known ancient agricultural or Neolithic homelands (Fig. 2b). Specifically, the inferred dispersal centre of Indo-European languages was located in the Fertile Crescent which is the earliest ancient agricultural homeland in the world (Fig. 2b)[3,4]. This observation favours the Anatolia origin hypothesis[7] of Indo-European languages rather than the alternative competing hypothesis of Pontic steppe region origin[48]. Moreover, in the case of Sino-Tibetan languages, their dispersal centre was inferred to be located in the Gansu Province of China (Fig. 2b). This centre is situated within the geographic ranges of the Yangshao (7000-5000 years BP) and/or Majiayao (5500-4000 years BP) Neolithic cultures[8] in the ancient agricultural homeland of China, the Yellow River plains[3,4]. This result supports the Northern origin hypothesis that Sino-Tibetan languages originated from the Yellow River plains in northern China[8,47,49]. For Bantu languages, the inferred dispersal centre was located in the southern Cameroon (Fig. 2b). This area is geographically adjacent to the known ancient agricultural homeland of Africa in eastern Nigeria and western Cameroon[3]. In addition, the LVF showed that the dispersal of Arawak languages could originate from the northern lowlands of Bolivia in the upper Madeira River basin (Fig. 2b). This area is an important homeland of ancient agriculture in lowland South America[4,50,51]. With four language cases, the LVF conformed to the spatial alignments of agricultural language dispersal patterns with human population activities drawn from ancient human genomes and archaeological materials[2–6] (Fig. 2a). Without loss of generality, we additionally varied different parametric settings for the LVF to infer the dispersal patterns of these four language families and groups. Our results showed that the inferred dispersal patterns remained robust across different parametric settings (Supplementary Figs. 7–10).

## Comparisons between the language velocity field estimation and phylogeographic approach

The LVF shares a similar theoretical foundation as the phylogeographic approach, which includes reconstructing the language dispersal pattern through the diachronic evolution of linguistic traits. However, these two approaches employ distinct strategies to fulfil this foundation. The primary distinction revolves around the representation of linguistic relatedness. To represent linguistic relatedness, our LVF conducts the PCA-based distance, whereas the phylogeographic approach relies on the phylogenetic tree. The PCA-based distance can capture the linguistic relatedness arising from both vertical divergence and horizontal contact. In contrast, the phylogenetic tree can solely capture the linguistic relatedness raised by vertical divergence. Accordingly, we raised the speculation regarding the consistency and inconsistency between these two approaches. Specifically, on the one hand, if the linguistic relatedness can be well reflected by the family-tree model, the LVF and phylogeographic approach should perform

similarly. On the other hand, if linguistic relatedness bears a significant horizontal influence, their performances could show considerable differences. To verify this speculation, utilising both the simulated and empirical datasets, we made comprehensive simulated and empirical comparisons between the LVF and phylogeographic approach. In this study, the phylogeographic approach was performed using the geographical model (PhyloG) in the BayesTraits programme[52]. The dispersal centres of four empirical cases inferred by LVF and phylogeographic approach are visualised in Fig. 3a, and the statistical comparison results are visualised in Figs. 3b1–3b5 and summarised in Fig. 3b6.

We first conducted the simulated comparison between the LVF and phylogeographic approach. In this comparison, the simulated datasets served as benchmarks to validate the performance between the LVF and phylogeographic approach when the linguistic relatedness can be adequately interpreted by the family-tree model. The reason is that the simulated datasets are generated by a given phylogenetic tree, meaning that the linguistic relatedness among simulated language samples can be regarded as being solely raised by vertical divergence. In other words, the linguistic relatedness among simulated language samples can be accurately explained by the family-tree model.

This simulated comparison involves two aspects. In the first aspect, we compared the performance between the LVF and phylogeographic approach by examining the differences between their estimated dispersal centres in longitude and latitude using 1000 simulated datasets. The results showed that there were no significant differences between the dispersal centres inferred by these two approaches in longitude and latitude (Lat: $p$ value = 0.85; Lon: $p$ value = 0.36; Fig. 3b1). In the second aspect, we examined the explanatory power of PCA-based distance and phylogenetic tree for linguistic relatedness among simulated language samples. Specifically, we computed three types of relatedness matrixes for simulated language samples, which are the overall relatedness matrix, PCA-based relatedness matrix, and tree-based relatedness matrix (see details in Methods). The overall relatedness matrix contains the Manhattan distance between each language sample pair, reflecting their overall relatedness arising from either divergence or contact. The PCA-based relatedness matrix encompasses the PCA-based Euclidean distance between each language sample pair within PC space, quantifying their relatedness due to either divergence or contact. The tree-based relatedness matrix includes the phylogenetic distance between each language sample pair on the phylogenetic tree, measuring their relatedness solely raised by vertical divergence. The statistical results of the Mantel test[53] showed that both PCA-based ($R^2 = 0.90$, $p$ value = 0.001; Fig. 3b6) and tree-based ($R^2 = 0.93$, $p$ value = 0.001; Fig. 3b6) relatedness matrixes were significantly correlated with the overall relatedness matrix. This indicates that both PCA-based distance and the phylogenetic tree show similar and high explanatory power for the linguistic relatedness that solely arises from vertical divergence. This therefore results in identical performance between the LVF and phylogeographic approach.

We next conducted the empirical comparison between the LVF and phylogeographic approach. In this comparison, the empirical datasets were utilised to validate the performance between the LVF and phylogeographic approach when linguistic relatedness bears a significant influence from horizontal contact. This empirical comparison involves three aspects. In the first aspect, we assessed the degree of the influence of horizontal contact on linguistic relatedness within these empirical datasets. This assessment is implemented by the delta score which is a widely utilised metric to quantify the degree of likeness between the language phylogenetic topology and the tree topology[54]. The larger value of the delta score implies that linguistic relatedness bears a larger influence of horizontal contact and cannot be well explained by the family-tree model[54]. Given that the phylogenetic topology of simulated language samples is highly compatible

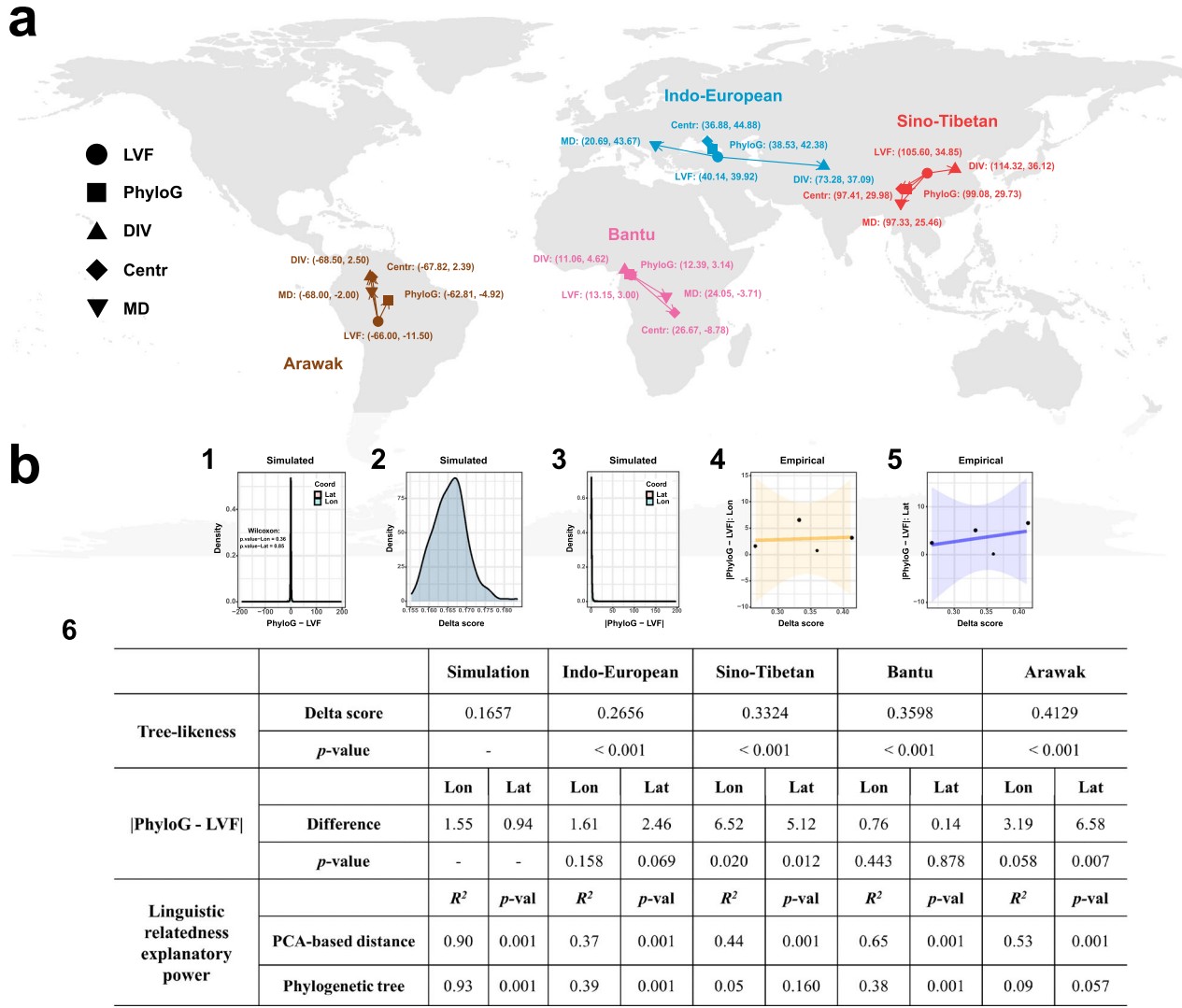

**Fig. 3 | Comparison between LVF and other spatial reconstruction approaches.**
**a** The geographic coordinates (Lon, Lat) of dispersal centres for each case inferred by five approaches: language velocity field estimation (LVF), phylogeographic approach (PhyloG), diversity approach (DIV), centroid approach (Centr), and minimal distance approach (MD). (b1) Density plot displaying differences in longitude and latitude between the dispersal centres inferred by LVF and PhyloG using 1000 simulated datasets. $p$ value is calculated by the two-sided Wilcoxon rank-sum test. (b2) Density plot showing the delta score distribution of simulated language samples (one-sided 95% CI = [0.1553, 0.1727]), estimated from 200 bootstrap resamplings. (b3) Density plot illustrating absolute differences in longitude and latitude between dispersal centres inferred by LVF and PhyloG using 1000 simulated datasets (Lat: mean = 0.94, one-sided 95% CI = [$4 \times 10^{-4}$, 2.82]; Lon: mean = 1.55, one-sided 95% CI = [$5 \times 10^{-5}$, 3.55]). (b4) Linear relation between the delta score and the absolute difference between dispersal centres in longitude estimated from LVF and PhyloG. The orange ribbon denotes the 95% CI. (b5) Linear relation between the delta score and the absolute difference between dispersal centres in latitude estimated from LVF and PhyloG. The blue ribbon denotes the 95% CI. (b6) Table displaying statistical test results for three indexes: delta score, absolute estimated difference between LVF and PhyloG, and linguistic relatedness explanatory power of PCA-based distance and phylogenetic tree. For the delta score, the $p$ value is calculated using the one-sided bootstrap test. For the absolute estimated difference, the $p$ value is calculated using the one-sided Monto-Carlo Simulation test. For linguistic relatedness explanatory power of PCA-based distance or phylogenetic tree, the $p$ value is calculated using the Mantel test. For all tests, statistical significance is indicated by $p$ value < 0.05. The grey base world map used in Subfigure (a) is generated using the map function of the maps package in R (4.3.1). The Source Data and Codes for generating Fig.3 are available.

with the tree topology, the delta score of simulated language samples can serve as the baseline for the linguistic relatedness that can be well interpreted by the family-tree model (one-sided 95% CI = [0.1553, 0.1727]; Fig. 3b2). With this baseline, the statistical examinations indicated that the language phylogenetic topologies within these four empirical datasets all significantly deviated from the tree topology (Indo-European: delta-score = 0.2656, $p$ value < 0.001; Sino-Tibetan: delta-score = 0.3324, $p$ value < 0.001; Bantu: delta-score = 0.3598, $p$ value < 0.001; Arawak: delta-score = 0.4129, $p$ value < 0.001; Fig. 3b6). This result suggests that the linguistic relatedness among the language samples within these empirical datasets bears significant influences from horizontal contact.

In the second aspect, we examined the differences between the dispersal centres in longitude and latitude inferred by the LVF and phylogeographic approach for each empirical case. It is noted that these two approaches perform similarly within the simulated datasets (Figure 3b1). Therefore, the absolute differences among the dispersal centres in longitude and latitude estimated by the LVF and phylogeographic approach within the simulated comparison can serve as baselines for empirical comparison (Fig. 3b3). These baselines quantify the absolute estimated differences between these two approaches in longitude and latitude when they exhibit similar performance (Lat: mean = 0.94, one-sided 95% CI = [$4 \times 10^{-4}$, 2.82]; Lon: mean = 1.55, one-sided 95% CI = [$5 \times 10^{-5}$, 3.55]; Fig. 3b3). With these baselines, the

statistical tests showed the significant differences between the dispersal centres estimated by the LVF and phylogeographic approach in the cases of Sino-Tibetan (Lat: diff = 5.12, $p$ value = 0.012; Lon: diff = 6.52, $p$ value = 0.020; Fig. 3b6) and Arawak languages (Lat: diff = 6.58, $p$ value = 0.007; Lon: diff = 3.19, $p$ value = 0.058; Fig. 3b6) but not in the Indo-European (Lat: diff = 2.46, $p$ value = 0.069; Lon: diff = 1.61, $p$ value = 0.158; Fig. 3b6) and Bantu (Lat: diff = 0.14, $p$ value = 0.878; Lon: diff = 0.76, $p$ value = 0.443; Fig. 3b6) languages. Despite these two approaches exhibiting identical performance in Indo-European and Bantu languages, we found that the differences between their estimated dispersal centres in longitude and latitude would increase as the delta score increased (Figs. 3b4, 3b5). This suggests that the stronger horizontal contact influence on linguistic relatedness would lead to a larger distinction between the performances of the LVF and the phylogeographic approach.

In the third aspect, we assessed the explanatory power of PCA-based distance and the phylogenetic tree for linguistic relatedness within empirical cases. We calculated the overall relatedness, PCA-based relatedness, and tree-based relatedness matrixes for the language samples within each case. Utilising the Mantel test[53], we observed that both PCA-based and tree-based relatedness matrixes exhibited significant correlations with the overall relatedness matrix in Indo-European (PCA-based distance: $R^2$ = 0.37, $p$ value = 0.001; phylogenetic tree: $R^2$ = 0.39, $p$ value = 0.001; Fig. 3b6) and Bantu languages (PCA-based distance: $R^2$ = 0.65, $p$ value = 0.001; phylogenetic tree: $R^2$ = 0.38, $p$ value = 0.001; Fig. 3b6). This indicates that both PCA-based distance and phylogenetic tree are highly explanatory for the linguistic relatedness within Bantu and Indo-European languages. This results in identical performance between the LVF and phylogeographic approach within the Bantu and Indo-European languages. However, only the PCA-based relatedness matrix significantly correlated with the overall relatedness matrix while the tree-based relatedness matrix did not within the Sino-Tibetan (PCA-based distance: $R^2$ = 0.44, $p$ value = 0.001; phylogenetic tree: $R^2$ = 0.05, $p$ value = 0.160; Fig. 3b6) and Arawak languages (PCA-based distance: $R^2$ = 0.53, $p$ value = 0.001; phylogenetic tree: $R^2$ = 0.09, $p$ value = 0.057; Fig. 3b6). This implies that only PCA-based distance manifests a high explanatory power for linguistic relatedness while the phylogenetic tree does not within the Sino-Tibetan and Arawak languages. It hence leads to the distinct performance between the LVF and phylogeographic approach within the Sino-Tibetan and Arawak languages.

According to the simulated and empirical comparisons, we confirm that the key distinction between the LVF and phylogeographic approach is rooted in their distinctive explanatory power for linguistic relatedness. Once the family-tree model is adequately explanatory for linguistic relatedness, the LVF and phylogeographic approach could exhibit similar performance. In contrast, a notable distinction between these two approaches could appear if the family-tree model cannot adequately reflect the linguistic relatedness. Moreover, such distinction could increase while the explanatory power of the family-tree model for linguistic relatedness decreases. Importantly, the genetic and archaeological evidence largely favoured the estimated results of the LVF within empirical cases but partially supported that of the phylogeographic approach. This suggests that the LVF may be more reliable than the phylogeographic approach when linguistic relatedness can be less explained by the family-tree model. Accordingly, the LVF can be regarded as an extension of the phylogeographic approach by relaxing its tree topology assumption of linguistic relatedness.

### Comparisons between the language velocity field estimation and other phylogeny-free baseline approaches

Apart from the phylogeographic approach, we also compared the LVF to the other three phylogeny-free approaches. They are the diversity (DIV), centroid (Centr), and minimal distance (MD) approaches[44,55] (see details in Methods). These approaches rest upon completely distinct

theoretical foundations from the LVF and phylogeographic approach. Specifically, the diversity approach postulates that the dispersal centre should be situated in the area encompassing the greatest linguistic diversity[44,55]. Linguistic diversity refers to the degree of distinctions among the linguistic traits of languages in a certain area[44,56], where a higher value implies greater distinctions (see details in Supplementary Discussion section 2). The centroid approach postulates that the centre of the polygon formed by the extension of current language geographic locations should be the dispersal centre[44]. The minimal distance approach posits that the location of the language that exhibits the smallest average geographic distance to the other languages should be the dispersal centre[44]. We applied these three basic approaches to the four empirical cases. The results showed that the dispersal centres inferred by the LVF exhibited significant differences from those inferred by these three approaches (Fig. 3a). This highlights the fundamental distinction between the LVF and these phylogeny-free approaches.

## Discussion

Aligning the spatiotemporal evidence of linguistics, genetics, and archaeology can be beneficial for comprehensively uncovering prehistoric human activities[14,15,57]. In this study, we proposed a computational approach, the LVF, to infer the language dispersal pattern without relying on the phylogenetic tree (see details in Supplementary Notes section 1). With 1000 simulated datasets, we validated the effectiveness and robustness of our LVF (see details in Supplementary Notes section 2). Utilising this verified LVF, we reconstructed the dispersal patterns of four prominent agricultural languages: Indo-European, Sino-Tibetan, Bantu, and Arawak languages. Our findings highlight that agricultural languages dispersed along with demic diffusions and cultural spreads in the past 10,000 years[5,6].

Compared to the phylogeographic and three phylogeny-free approaches, our LVF may exhibit some methodological advantages in empirical applications (see details in Supplementary Discussion section 3). In contrast to the phylogeographic approach, the LVF can be used independently of the phylogenetic tree and accounts for the histories of both language vertical divergence and horizontal contact. Therefore, the application of the LVF can be flexibly extended into structural features such as grammar and sound[58,59], rather than just limited to lexicons (e.g., lexical cognate). Compared to lexicons, structural features usually exhibit more complex evolutionary processes, such as contact-induced changes and convergence[32]. These processes could not be completely modelled by the family-tree model[32–34,36]. Therefore, the LVF may allow for the utilisation of various linguistic traits (i.e., lexicon, grammar, and sound) to infer the dispersal patterns of languages whose linguistic relatedness cannot be described by the family-tree model (e.g., Chinese dialects[60], Indo-Aryan languages across India[61], and Oceanic languages across Pacific settlements[62]). In contrast to the other three phylogeny-free approaches, the LVF can be used to infer the dispersal pattern of languages when they exhibit unbalanced diversity or nonuniform dispersal rates across geographic space shaped by other factors such as sampling bias and population migration[63] (see details in Supplementary Discussion section 2).

With these methodological advantages, the dispersal patterns of four empirical cases inferred by the LVF can be largely supported by interdisciplinary evidence. Nevertheless, the origins and dispersals of some of these empirical cases remain controversial[7–9,24,47,48]. For the origin and dispersal of Indo-European languages, Bouckaert et al. (2012)[7] supported the Anatolia hypothesis by revealing that Indo-European languages originated in Anatolia approximately 7000-10,000 years ago. In contrast, Chang et al. (2015)[48] declared that Indo-European languages originated approximately 6000 years ago, which strongly supported the steppe hypothesis, by reanalysing datasets provided by Bouckaert et al. (2012). Note that Chang et al. (2015)

modified several time calibrations according to linguistic evidence but did not apply phylogeographic reconstruction. Therefore, the homeland of Indo-European languages remains controversial. Following these two works, we utilised the same dataset provided by Bouckaert et al. (2012). Our spatial reconstruction of Indo-European languages supported the same Anatolia hypothesis as Bouckaert et al. (2012). For Sino-Tibetan origin and dispersal, recent phylogenetic studies dated the divergence time of the Sino-Tibetan languages to the Neolithic period[8,47,49] and supported the Northern origin hypothesis. Therefore, the homeland of Sino-Tibetan languages should be located in northern China. However, spatial reconstruction has not been rigorously implemented to infer the homeland of the Sino-Tibetan languages thus far. Nevertheless, corroborated with previous temporal evidence, our LVF indeed revealed the Sino-Tibetan dispersal centre situated in the location at the upper Yellow River plains, Northern China.

Despite several methodological advantages, we must note that the LVF should not be considered completely superior to other approaches, especially the phylogeographic approach. In contrast, the LVF can be viewed as the extension of the phylogeographic approach by relaxing its tree topology assumption of linguistic relatedness. However, if linguistic relatedness can be well illustrated by the family-tree model, the phylogeographic approach should be a better solution than the LVF, because the phylogenetic tree is a more accurate representation of linguistic relatedness than the PCA-based distance. Nevertheless, we still believe that the LVF can serve as a useful compensation for the phylogeographic approach when the family-tree model cannot adequately capture linguistic relatedness. Moreover, several improvements in the LVF should be further accomplished in the future (see details in Supplementary Discussion section 3), such as estimating the period of language dispersal and adjusting the estimation bias raised by the sampling bias of language geographic distribution (Supplementary Fig. 11; see details in Supplementary Notes section 1.3.2). Overall, we still anticipate that the LVF could aid the spatial analysis of language evolution and branch out into other interdisciplinary fields such as genetics and archaeology.

## Methods

### Linguistic data
Our linguistic datasets are sourced from the public lexical datasets of four language families and groups. They contain several lexical words following a specific wordlist such as the Swadesh 100 or 200 wordlists[64]. These words have been well coded as different lexical cognates by previous linguistic experts. Each lexical word contains several cognates that manifest the same meaning and systematic sound correspondences. For calculation, each cognate has been further recoded as a new binary-coded linguistic trait, where 1 signifies the presence of this cognate in a language, while 0 signifies the absence (Fig. 1a). Therefore, our linguistic datasets encompasses 5995 linguistic traits across 103 Indo-European language samples[7], 949 linguistic traits across 109 Sino-Tibetan language samples[8], 3859 linguistic traits across 420 Bantu language samples[9], and 693 linguistic traits across 60 Arawak language samples[24]. Additionally, each language sample is also assigned a geographic coordinate in terms of longitude and latitude (Fig. 1c).

### Imputation of missing values
We first removed linguistic traits with > 75% missing values. Then, we used the mode-value imputation approach to impute missing values of the remaining linguistic traits. To evaluate the efficiency of the mode-value imputation approach, we employed the metric of cosine similarity[65] to measure the similarity between the velocity fields estimated with and without mode-value imputation (see details in Supplementary Methods section 1.4.1). Similarly, we also evaluated the consensus of the velocity fields estimated under three imputation approaches: frequency-value imputation, zero-value imputation, and

mode-value imputation. We also utilised Procrustes analysis[66,67] to examine the consistency among PC values of linguistic traits imputed by these three approaches (Supplementary Fig. 7 and Supplementary Table 3). All the evaluations showed that the imputation of missing values would not affect the estimation of the velocity field.

### Conversion of the binary-coded trait into a frequency trait
The LVF necessitates linguistic traits ranging from 0 to 1. We employed the $k$-nearest neighbours ($k$-NN) algorithm to convert each binary-coded linguistic trait into a frequency trait ranging from 0 to 1 (Fig. 1c). First, we selected $k$ language samples that are geographically nearest to a given language sample (including itself)[68,69]. Second, we calculated the frequencies of each linguistic trait exhibiting state 1 and state 0 within these $k$ language samples. Since the sum of different state frequencies for each linguistic trait equals one in these $k$ language samples, the frequency of state 0 for each linguistic trait can be determined once given the frequency of state 1. Accordingly, there is no difference in converting the binary state of a linguistic trait into the frequency of either state 1 or state 0. In this study, for each language sample, we converted the binary value of each linguistic trait into its frequency of exhibiting state 1 (hereafter state frequency) in its $k$ nearest language samples. For this conversion, we set $k = 10$, which has been verified in both simulated and empirical validations (Supplementary Figs. 2, 3, and 8). In practice, the state frequency of a linguistic trait can also be computed by disregarding missing values of that trait within the $k$-nearest language samples.

### The dynamic model for linguistic trait evolution
We proposed three model assumptions regarding the evolution of linguistic traits. First, each linguistic trait can undergo multiple transitions among different states with heterogeneous rates during evolution. Second, the variation of each linguistic trait within a language can be influenced by neighbouring languages. Particularly, such influence could arise from the competition among different states of the same linguistic trait possessed by neighbouring languages. Third, each trait state holds a specific sociolinguistic prestige in a certain area. This prestige reflects the social opportunities or convenience afforded to an individual who speaks the language with this trait state. A state with higher prestige will occur more frequently in future generations, while a state with lower prestige will correspondingly decline. Accordingly, the prestige of a state can be measured by its probability of being inherited by future generations.

According to these assumptions, we proposed a simple dynamic model as Eq. (1) derived from the Abrams-Strogatz (AS) model[70] (Fig. 1d1). The AS model simulates two-language competition, where one language with higher prestige will persist while one with lower prestige will decline. Accordingly, the AS model shares a similar rationale with our model, which can be utilised to demonstrate linguistic trait evolution.

$$\begin{cases} \frac{dx_0^i}{dt} = x_1^i q_{10}(x_0^i, s_0^i) - x_0^i q_{01}(x_1^i, s_1^i) \\ \frac{dx_1^i}{dt} = x_0^i q_{01}(x_1^i, s_1^i) - x_1^i q_{10}(x_0^i, s_0^i) \end{cases} \quad (1)$$

Here, $x_1^i$ denotes the frequency of state 1 (state frequency) for linguistic trait $i$, while $x_0^i = 1 - x_1^i$ denotes the frequency of state 0 for linguistic trait $i$. $s_1^i$ and $s_0^i$ signify the prestige of state 1 and state 0 for linguistic trait $i$, respectively. Following our previous study[71], the prestige of state $j$ for trait $i$ ($s_j^i$, $j = 0$ or 1) could be redefined as the inheritance rate signifying the probability that trait $i$ with state $j$ remains in state $j$ after a unit of time (one generation). $q_{uv}(x_v^i, s_v^i) = s_v^i x_v^i$ ($u, v = 0, 1$ or $1, 0$) denotes the transition rate from state $u$ to state $v$. Our dynamic model is akin to the covarion model[45,46] that is extensively used to model trait evolution in phylo-linguistics (e.g., phylogenetic studies of Indo-European[7,19,48] and Sino-Tibetan languages[8,47]). The

rationale of the covarion model also posits that each linguistic trait can undergo multiple transitions between gain and loss (i.e., from state 0 to state 1 or from state 1 to state 0), and shift between fast and slow evolutionary rates[8,48]. This model aligns with linguists' intuition that different linguistic traits should experience distinct evolutionary processes[8].

## The estimation of the prestige parameter

To estimate the prestige parameter in our dynamic model, we devised a parametric estimation principle derived from the DNA substitution model in Genetics proposed by Felsenstein[72] (see details in Supplementary Notes section 1.2). This DNA substitution model rests upon the Poisson process, postulating that each base can undergo multiple transitions to other bases (e.g., A transiting to T or C transiting to G) with a heterogeneous rate during DNA evolution[72]. This is analogous to our model assumption that each linguistic trait can undergo multiple shifts between gain and loss with a heterogeneous rate during evolution. Accordingly, we also used the Poisson process to model the gain and loss of each linguistic trait. The prestige parameter can be estimated using Eq. (2) (see details in Supplementary Methods section 1.1.1).

$$\begin{cases} s_1^i = e^{-\lambda} + (1 - e^{-\lambda})\,\pi_1^i \\ s_0^i = e^{-\lambda} + (1 - e^{-\lambda})\,\pi_0^i \end{cases} \tag{2}$$

Here, $s_j^i$ denotes the prestige of state $j$ ($j = 0$ or 1) for trait $i$. $\lambda$ is the mutation rate of the Poisson process, which signifies the number of mutations occurring per unit of time in expectation. Following the definition utilised in phylogenetic studies, a unit of time is defined as the period during which linguistic traits in a language undergo one mutation[25,73] (see details in Supplementary Notes section 1.2.1). Accordingly, we set $\lambda = 1$, which implies that linguistic traits will experience one mutation in a unit of time in expectation. This setting $\lambda = 1$ has been verified in both simulated and empirical validations (Supplementary Figs. 2, 3, and 9). $\pi_j^i$ ($j = 0$ or 1) denotes the transition probability that a transition will result in any current state of trait $i$ eventually being replaced with state $j^{72,74}$. In this study, we set $\pi_j^i$ as the frequency of state $j$ for trait $i$ within all the language samples following previous studies[75]. Since we usually lack temporal information regarding linguistic traits, we considered that this parametric setting of $\pi_j^i$ could facilitate the Poisson process to better interpret the formation of the observed state distribution in trait $i$.

## Reconstruct the past state frequency for each linguistic trait

We reconstructed the past state frequency for each linguistic trait using Eq. (3) which is the analytical solution of Eq. (1) (Fig. 1d2; see details in Supplementary Methods section 1.1.2).

$$x_1^i(-m) = \left[1 + \left(\frac{1}{x_1^i(0)} - 1\right) e^{(s_1^i - s_0^i)m}\right]^{-1} \tag{3}$$

Here, $x_1^i(0)$ signifies the state frequency (frequency of state 1) of trait $i$ at present, while $x_1^i(-m)$ denotes the state frequency of trait $i$ at $m$ units of time before the present. In this study, we set $m = 1$, which has been verified in both simulated and empirical validations (Supplementary Figs. 2, 3, and 10). Once the occurrence of the past trait state can be dated, this dimensionless unit of time can be converted into an exact period.

## Establish the velocity field

We established a high-dimensional velocity field to quantify the diachronic evolutionary trajectories of linguistic traits. This velocity field is composed of a collection of velocity vectors. The velocity vector for language $l$ ($\mathbf{V}_l$) is approximated as the difference between past and present state frequencies of its linguistic traits divided by the reconstruction time as shown in Eq. (4) (Fig. 1d2; see details in Supplementary Methods section 1.1.3).

$$\mathbf{V}_l = \frac{1}{m}[\mathbf{X}_l(0) - \mathbf{X}_l(-m)] \tag{4}$$

Here, $\mathbf{X}_l(0) = [x_{l1}^1(0), x_{l1}^2(0), \ldots, x_{l1}^p(0)]^T$ and $\mathbf{X}_l(-m) = [x_{l1}^1(-m), x_{l1}^2(-m), \ldots, x_{l1}^p(-m)]^T$. $x_{l1}^i(0)$ denotes the state frequency (frequency of state 1) of trait $i$ for language $l$ at present. $x_{l1}^i(-m)$ denotes the state frequency of trait $i$ for language $l$ at $m$ units of time before the present. For each language family or group, the velocity vectors of $n$ language samples can compose a high-dimensional velocity field denoted as matrix $\mathbf{V}$ as shown in Eq. (5).

$$\mathbf{V} = [\mathbf{V_1}, \mathbf{V_2}, \ldots, \mathbf{V_n}]^T \tag{5}$$

## PCA projection of the velocity field

We projected the high-dimensional velocity field $\mathbf{V}$ into PC space to depict the diachronic evolutionary trajectories of linguistic traits that shape observed linguistic relatedness. First, we conducted PCA[76] on the binary-coded linguistic data to rearrange the binary-coded linguistic traits into two optimal new traits (i.e., PC1 and PC2) using Eq. (6) (Fig. 1b).

$$[\mathbf{PC1}, \mathbf{PC2}] = \mathbf{PC} = \begin{bmatrix} \mathbf{PC_1}^T \\ \mathbf{PC_2}^T \\ \vdots \\ \mathbf{PC_n}^T \end{bmatrix} = \mathbf{DA_2} \tag{6}$$

Here, $\mathbf{D}$ signifies a matrix containing $n$ language samples and $p$ binary-coded linguistic traits. $\mathbf{PC1}$ and $\mathbf{PC2}$ are the PC values of $n$ language samples. $\mathbf{PC}_l$ denotes the PC values of language $l$. $\mathbf{A_2}$ is the matrix containing the first two columns of the eigenvector matrix of the covariance matrix of $\mathbf{D}$ (see details in Supplementary Methods section 1.2.1). Second, the high-dimensional velocity field $\mathbf{V}$ is projected into the two-dimensional PC space using Eq. (7) (Fig. 1e1; see details in Supplementary Methods section 1.2.1). This projection can be regarded as mapping the present and past state frequencies of linguistic traits in each language sample simultaneously into PC space followed by taking their difference divided by reconstruction time.

$$\mathbf{V}^{PC} = \begin{bmatrix} (\mathbf{V_1^{PC}})^T \\ (\mathbf{V_2^{PC}})^T \\ \vdots \\ (\mathbf{V_n^{PC}})^T \end{bmatrix} = \mathbf{VA_2} = \begin{bmatrix} \mathbf{V_1^T A_2} \\ \mathbf{V_2^T A_2} \\ \vdots \\ \mathbf{V_n^T A_2} \end{bmatrix} = \frac{1}{m}\left( \begin{bmatrix} \mathbf{X_1^T}(0) \\ \mathbf{X_2^T}(0) \\ \vdots \\ \mathbf{X_n^T}(0) \end{bmatrix} \mathbf{A_2} - \begin{bmatrix} \mathbf{X_1^T}(-m) \\ \mathbf{X_2^T}(-m) \\ \vdots \\ \mathbf{X_n^T}(-m) \end{bmatrix} \mathbf{A_2} \right) \tag{7}$$

Here, $\mathbf{V}^{PC}$ denotes the velocity field within the PC space. $\mathbf{V}_l^{PC}$ signifies the velocity vector of language $l$ within PC space. Based on the above steps, we could derive a velocity field within the PC space that can visualise the diachronic evolutionary trajectories of linguistic traits that shape observed linguistic relatedness (Fig. 1e2 and Supplementary Fig. 4). The PCA algorithm was performed by the prcomp function in R (4.3.1).

## Geographic projection of the velocity field in the PC space

According to the observed correlation between linguistic relatedness and language geography, the kernel projection proposed by La Manno et al.[40] is conducted to project the velocity field from the PC space into the geographic space (Figs. 1e, 1f, and Supplementary Figs. 4–5). The kernel projection seeks each velocity vector in the geographic space,

ensuring that its correlation with language distribution in the PC space aligns closely with the one in the geographic space (see details in Supplementary Methods section 1.2.2). With kernel projection, the velocity vector of language $l$ within the geographic space ($\mathbf{V}_l^{Geo}$) can be calculated based on Eq. (8) proposed by La Manno et al.[40] (see details in Supplementary Methods section 1.2.2). The direction of $\mathbf{V}_l^{Geo}$ reflects from where language $l$ diffused into its current geographic location (Fig. 1f1). The velocity field within the geographic space is noted as matrix $\mathbf{V}^{Geo} = [\mathbf{V}_1^{Geo},\dots,\mathbf{V}_n^{Geo}]^T$.

$$\mathbf{V}_l^{Geo} = \sum_{j=1}^{s}\left(P_{lj} - \frac{1}{s}\right)\frac{\mathbf{C}_j - \mathbf{C}_l}{||\mathbf{C}_j - \mathbf{C}_l||} \tag{8}$$

Here, $\mathbf{C}_l$ and $\mathbf{C}_j$ represent the geographic coordinates of languages $l$ and $j$, respectively. $P_{lj}$ demonstrates the correlation between the velocity vector of language $l$ within PC space ($\mathbf{V}_l^{PC}$) and the distinction between the PC values of languages $l$ and $j$ ($\mathbf{PC}_j$ - $\mathbf{PC}_l$). It measures the correlation between the $\mathbf{V}_l^{PC}$ and the distribution of languages $l$ and $j$ within PC space (see details in Supplementary Methods section 1.2.2). Particularly, language $j$ is one of the $s$ language samples closest to language $l$ in the PC space (see details in Supplementary Methods section 1.2.2).

## Spatial and grid smoothing for the velocity field

We employed the spatial and grid smoothing approaches to better visualise the velocity field $\mathbf{V}^{Geo}$ within geographic space. These smoothing approaches contribute to better visualising the velocity field while preserving the original language dispersal pattern as reflected in $\mathbf{V}^{Geo}$. For spatial smoothing, we first solely scaled the length of each velocity vector to be the same as the one in the PC space using Eq. (9). Second, we further adjusted its length by weighting the lengths of velocity vectors of other language samples using Eq. (10) (Fig. 1f2; see details in Supplementary Methods section 1.2.3).

$$\mathbf{V}_l^{Geo-scale} = \frac{\mathbf{V}_l^{Geo}}{||\mathbf{V}_l^{Geo}||}||\mathbf{V}_l^{PC}|| \tag{9}$$

$$\mathbf{V}_l^{Geo-scale-smooth} = \frac{\mathbf{V}_l^{Geo-scale}}{||\mathbf{V}_l^{Geo-scale}||}\sum_{j=1}^{n}K_\sigma(\mathbf{C}_l,\mathbf{C}_j)||\mathbf{V}_j^{Geo-scale}|| \tag{10}$$

Here, $K_\sigma(\mathbf{C}_l,\mathbf{C}_j)$ is the Gaussian kernel measuring the closeness between the geographic locations of languages $l$ and $j$ (see details in Supplementary Methods section 1.2.3). It is noted that the language dispersal pattern is determined by the directions of the velocity vectors. Accordingly, the spatial smoothing procedure, which exclusively adjusts the vector lengths rather than the vector directions, would not alter the language dispersal pattern as reflected in the original velocity field $\mathbf{V}^{Geo}$.

Grid smoothing aims to better visualise a velocity field on regular grid points. With grid smoothing, the velocity vectors could exhibit a uniform distribution across the geographic space (Fig. 1g). Moreover, the grid smoothing can also estimate the velocity vectors within the geographic area which lacks available language samples. This ensures that the grid-smoothed velocity field can effectively illustrate the continuous dispersal pattern of language samples throughout their entire geographic span. The velocity vector at grid $g$ ($\mathbf{V}_g^{Grid}$) is calculated using Eq. (11) (Fig. 1g; see details in Supplementary Methods section 1.2.4).

$$\mathbf{V}_g^{Grid} = \sum_{l=1}^{s}K_\sigma(\mathbf{C}_g^{Grid},\mathbf{C}_l)\mathbf{V}_l^{Geo-scale-smooth} \tag{11}$$

Here, $\mathbf{C}_g^{Grid}$ denotes the geographic coordinate of grid $g$. $\mathbf{V}_l^{Geo-scale-smooth}$ signifies the spatial-smoothed velocity vector of

language $l$ which is one of the $s$ language samples geographically closest to the grid $g$ (see details in Supplementary Methods section 1.2.4). Accordingly, we denote the grid-smoothed velocity field defined at $M$ grid points as the matrix $\mathbf{V}^{Grid} = [\mathbf{V}_1^{Grid},\dots,\mathbf{V}_M^{Grid}]^T$.

## Dispersal centre inference

To infer the language dispersal centre, we designed a simple strategy relying on the grid-smoothed velocity field $\mathbf{V}^{Grid}$ within the geographic space. Given that the velocity vectors within geographic space depict the language dispersal directions, we postulated that the velocity vectors around the dispersal centre should exhibit an outwards radiative pattern (see details in Supplementary Notes section 1.3.1). According to this postulation, we measured the degree of the outwards radiative pattern of grid-smoothed velocity vectors around each grid point. The degree of such a pattern is measured by calculating the average for the variance (average variance) of these velocity vectors in each dimension as shown in Eqs. (12–13) (see details in Supplementary Methods section 1.3.1). The grid point that exhibits the highest average variance, which indicates the strongest outwards radiative pattern of the neighbouring velocity vectors, is regarded as the language dispersal centre (Fig. 1g).

$$\mathbf{V}_g^{Grid-scale} = \frac{\mathbf{V}_g^{Grid}}{||\mathbf{V}_g^{Grid}||} \tag{12}$$

$$\sigma_g^2 = \frac{1}{2(s-1)}tr\left[\left(\mathbf{V}^{Grid-scale}\right)^T\left(\mathbf{E}_s - \frac{\mathbf{1}\mathbf{1}^T}{s}\right)\left(\mathbf{V}^{Grid-scale}\right)\right] \tag{13}$$

Here, $\mathbf{V}_g^{Grid-scale}$ signifies the normalised grid-smoothed velocity vector of grid $g$. $\sigma_g^2$ denotes the average variance of grid $g$. $\mathbf{E}_s$ is the identity matrix with $s$ rows and $s$ columns. $\mathbf{1} = [1,1,\dots 1]^T$. $\mathbf{V}^{Grid-scale} = [\mathbf{V}_1^{Grid-scale},\dots,\mathbf{V}_s^{Grid-scale}]^T$ represents the normalised grid-smoothed velocity vectors of $s$ grid points geographically closest to grid $g$. Using the traditional jackknife resampling approach[77], we also estimated the standard deviation (SD) of the estimated geographic coordinate of the language dispersal centre for each language case (Supplementary Table 2 and Supplementary Fig. 11; see details in Supplementary Methods section 1.3.2). For the sampling criteria of the traditional jackknife approach[77], the number of jackknife samples for each language case equals the number of its language samples.

## Simulated validation for LVF

To validate the effectiveness and robustness of the LVF, we applied it to the 1000 simulated linguistic datasets obtained from Wichmann and Rama (2021)[44]. To evaluate the effectiveness of the LVF, we adopted the two-sided Wilcoxon rank-sum test to examine the difference between the given and inferred coordinates of the dispersal centres under specific parametric settings ($k = 10$, $\lambda = 1$, and $m = 1$). $k$ denotes the $k$-nearest neighbours, $\lambda$ denotes the mutation rate of the Poisson process, and $m$ denotes the reconstruction time. The results showed that under the parametric setting of $k = 10$, $\lambda = 1$, and $m = 1$, the dispersal centre estimated by LVF was not significantly different from the given dispersal centre (Supplementary Fig. 2). This indicates the high effectiveness of LVF with these parametric settings. Furthermore, we also conducted the two-sided Wilcoxon rank-sum test to examine the differences between the given and inferred coordinates of the dispersal centres under different parametric settings. Specifically, we varied across the values of $k$ ($k = 2, 4, 6, \dots, 18$), $\lambda$ ($\lambda = 0.1, 0.5, 1, 5, 10$), and $m$ ($m = 1, 3, 5, 7, 9$) for LVF when applying it to the simulated datasets. The results showed that the inferred coordinates of the language dispersal centres under different parametric settings were not significantly different from the given one. This indicates that the LVF remains effective under different parametric settings (Supplementary Fig. 2). Accordingly, we set $k = 10$, $\lambda = 1$, and $m = 1$ as default parametric values for LVF.

To validate the robustness of the LVF against the different parametric settings, we examined the cosine similarity among the velocity fields estimated from different parametric settings in either high-dimensional or two-dimensional PC spaces (see details in Supplementary Methods section 1.4.2, section 1.4.3, and section 1.4.4). Specifically, we varied across the values of $k$ ($k = 2, 4, 6, …, 18$), $\lambda$ ($\lambda = 0.1, 0.5, 1, 5, 10$), and $m$ ($m = 1, 3, 5, 7, 9$) for LVF when applying it to the simulated datasets. The results showed that the velocity fields exhibited no significant difference from each other under different parametric settings. This indicates that the LVF is highly robust against the different parametric settings (Supplementary Fig. 3). According to the simulation results, we also provided suggested ranges of the parametric settings for the empirical application of the LVF (Supplementary Fig. 12; see details in Supplementary Notes section 2.2.3).

### Empirical validation for LVF

We assessed the robustness of the LVF in empirical applications against different parametric settings. Specifically, we applied the LVF to estimate the velocity fields of four empirical cases with different values of $k$ ($k = 5, 10, 15, 20$), $\lambda$ ($\lambda = 0.1, 0.5, 1, 5, 10$), and $m$ ($m = 1, 3, 5, 7, 9$). Subsequently, we conducted the cosine similarity to examine the similarity among the velocity fields under these different settings (see details in Supplementary Methods section 1.4.2, section 1.4.3, and section 1.4.4). The results showed that the velocity fields estimated under different parametric settings exhibited no significant difference from each other in either high-dimensional or two-dimensional PC spaces (Supplementary Figs. 8–10). This confirms the robustness of the LVF against the different parametric settings.

### Three types of relatedness matrixes and delta scores

The overall relatedness matrix is constructed by quantifying the Manhattan distance between the binary-coded linguistic traits of each language sample pair. The PCA-based relatedness matrix is derived by calculating the Euclidean distance between two optimal principal components that are rearranged from binary-coded linguistic traits of each language sample pair. The tree-based relatedness matrix is generated by measuring the length of the branch linking each language sample pair (i.e., pairwise phylogenetic distance) on the given phylogenetic tree. The pairwise phylogenetic distance can be estimated using the cophenetic.phylo function of the ape package[78] in R (4.3.1). The correlations among these relatedness matrixes are estimated and examined by the Mantel test using the mantel function of the vegan package[79] in R (4.3.1). The delta score serves as a metric to quantify the tree-likeness of language phylogenetic topology. It is calculated using the delta.plot function of the ape package in R (4.3.1)[78].

### Diversity, Centroid, and Minimal distance approaches

For the diversity approach, the linguistic diversity of a certain area was measured by the information entropy in this study as Eq. (14)[80]. The diversity approach posits that the geographic location of the language sample that exhibits the highest diversity is the dispersal centre, as shown in Eq. (15) (see details in Supplementary Methods section 1.3.3).

$$Div_l = -\mathbf{X}_l^T(0) \log\left[\mathbf{X}_l(0)\right] - [\mathbf{1} - \mathbf{X}_l^T(0)] \log[\mathbf{1} - \mathbf{X}_l(0)] \quad (14)$$

$$C_{centre} = C_{\underset{l}{\mathrm{argmax}(Div_l)}} \quad (15)$$

Here, $\mathbf{X}_l(0) = [x_{l1}^1(0), x_{l1}^2(0), …, x_{l1}^p(0)]^T$ and $\mathbf{1} = [1,1,…,1]^T$. $x_{l1}^i(0)$ denotes the state frequency (frequency of state 1) of trait $i$ for language $l$ at present. $Div_l$ signifies the diversity of language $l$. $C_l$ represents the geographic coordinate of language $l$. $C_{centre}$ denotes the geographic coordinate of the language dispersal centre. It is noted that each $x_{l1}^i(0)$ is the composite value of the binary values of trait $i$ for language $l$ and its $k$ geographically nearest language samples. Consequently, $Div_l$ is a measurement of the linguistic diversity in the area covering language $l$ and its $k$-nearest language samples. For the centroid approach, we calculated the centroid of the polygon represented by the extension of the geographic locations of the observed language samples using the centroid function of the geosphere package in R (4.3.1). The geographic location of this centroid is regarded as the language dispersal centre. For the minimal distance approach, we computed the average Euclidean distance from each language sample to all the other language samples according to their geographic coordinates. The geographic location of the language sample that has the smallest average Euclidean distance to other language samples is regarded as the dispersal centre.

### Reporting summary

Further information on research design is available in the Nature Portfolio Reporting Summary linked to this article.

## Data availability

All the datasets used in this study are available on GitHub (https://github.com/Stan-Sizhe-Yang/Inferring-language-dispersal-patterns-with-velocity-field-estimation) and Zendo (https://doi.org/10.5281/zenodo.10223872). The Source data for generating Figs. 1–3 and Supplementary Figs. 2-12 are available in GitHub and Zendo.

## Code availability

All analyses were performed in R (4.3.1). For the convenience of utilising LVF, we built an R package named LVF and provided a comprehensive tutorial for its application. This R package with its tutorial and other R codes for the implementation, validation, and comparison of the LVF are all available on GitHub (https://github.com/Stan-Sizhe-Yang/Inferring-language-dispersal-patterns-with-velocity-field-estimation) and Zendo (https://doi.org/10.5281/zenodo.10223872). The Source Codes for generating Figs. 1–3 and Supplementary Figs. 2-12 are available in GitHub and Zendo.

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

## Acknowledgements

This research is supported by the National Natural Science Foundation of China (T2122007 and 32070577), National key research and development programme (2020YFE0201600), National Social Science Foundation (20&ZD301), Shanghai Municipal Science and Technology Major Project (2017SHZDZX01), and the European Research Council (ERC) under the European Union's Horizon 2020 research and innovation programme (Grant Agreement No. 883700 TRAM). This work is also sponsored by "Shuguang Programme" supported by Shanghai Education Development Foundation and Shanghai Municipal Education Commission (20SG06) and supported by Fundamental Research Funds for the Central Universities(2022ECNU-XWK-XK005).

## Author contributions

S.Y., L.J. and M.Z. designed the research; S.Y. and M.Z. performed the research; S.Y. and M.Z. developed the computational approach; S.Y., X.S. and M.Z. evaluated the computational approach; S.Y., X.S., L.J. and M.Z. discussed the results; S.Y., X.S., L.J. and M.Z. wrote the paper.

## Competing interests

The authors declare no competing interests.
