## [Peer Review File · Nature Communications]

Inferring language dispersal patterns with velocity field estimationReviewers' Comments:

Reviewer #1:

Remarks to the Author:

The authors propose a vectorial framework to reconstruct the spatial dispersal of four language families around the world. The authors use a very wide range of methods that are borrowed from data science, physics and others from linguistics. I do not have the expertise to cover all of these methods, however the authors could help the reader understand if these methods are clustering algorithms, prediction methods, accuracy tests, etc. Some methods are called in the main text without further description, while some others are wrongly described, e.g. PCA is described in the main text as a similarity or clustering algorithm, actually PCA helps filtering out the least important features in order to describe a target variable in a space defined by superposition of few important features.

The methods section is a repetition of the vague description of the tools made in the main text and no further information is provided. The reader needs to get to Supplementary Information #3 to finally get a technical description of the methods that should actually appear in the Methods section.

However, here the technical details are not clearly expressed and the physical meaning of the vector is unclear. Due to this, all the following results are unclear.

The text is hard to read, mostly due the presence of many typos and other grammar issues. Long sentences are used for speculative purposes, while key methodological descriptions are narrowed down to few vague sentences.

I realize that the authors did a very hard work and that the storytelling is not easy to unroll in a linear way. Still, I feel that the authors should make an effort to simplify, correct and make the text clearer in order to be readable by an interdisciplinary audience.

Here is a list of concerns:

- typo in the abstract, the sentence "And its effectiveness and robustness have been carefully verified by both simulated and empirical validations" starts with "and".
- line 87: again the sentence starts with "and"
- "And such relatedness could vary with time when languages continuously dispersal into new regions." sentence starts with "and" + dispersal is a noun, the verb is disperse. The same is repeated in many other sentences, please correct.
- line 106: "The Principal Component Analysis (PCA) is implemented to exhibit the linguistic relatedness of present languages." it is not clear on what variables the PCA is implemented. PCA identifies the most important variables to explain the variance of a target variable (in this case, I guess, the target variable is the languages relatedness?). Clustering classification is a forthcoming step.
- subsection "Simulated validations for language velocity field estimation". I really struggle here to understand what data did the authors use to validate their results. The dataset that is supposedly used as ground truth is also simulated by a phylogeographic algorithm. The authors claimed in the introduction that this method only captures vertical dependency of languages and not horizontal contacts and borrowings. I am confused about what is the contribution of this validation. Maybe the authors could add this discussion in the limitations of the study.
- I would avoid the usage of the word "true", unless there are striking evidences of the coordinates of the language dispersal origin.
- what is the delta score of tree-likeness?
- the authors do not describe the data they used accurately. For instance, what is a trait? What is a cognate? It is never stated.
- "Third, the changes in the state frequencies of linguistic traits are proportional to their sociolinguistic prestige in a certain area.". I don't get the logic of this sentence. What is the meaning of prestige here? The definition of prestige is expressed only in the next paragraph, it should be introduced before going into interpretations.
- "It is noted that the larger length of the velocity vector of a language denotes the more rapid change of this language during its evolution". The reader is provided with no tools to understand this

sentence. A schematic representation of a vector could really help. E.g. what are the elements of a vector?

- Does the PCA find only two components, or the authors found that more components did not lead to more variance explainability? Again, PCA here is presented as a tool to find similarities among datapoints, actually it is a rearrangements of the predictors of the model that tells what are the most important features in the model. The authors say nothing about all this. Projecting the points in to the PC space allows to visualize clusters, but actual clustering is performed by other tools, such as k-nearest-neighbors.

- it is not clear how the vectors are formed in the PC space. Up to my understanding the PCA describes the datapoint with two components, hence I expect to observe a single point with coordinates (PC1,PC2) in the PC space. By the way, we cannot build a vector with one point. I understand from SI-3 that the vectors are computed as the difference in the PC space of $X(0) - X(-m)$, where $t=0$ represents now and $t=-m$ represents a moment in the past. What is this moment in the past? Then I read "Therefore, VI describes the change of the state frequencies of language I in a unit of time.". what is the unit of time? Years, centuries?

- what is the delta score and how is it computed? It is never stated in the text, nor in the SI

- Later on I read "In this study, we set $m = 1$.", but no reason is given, nor the unit of time is stated. One year? One century? Again, this is very opaque. I do not understand the physical meaning of this vectorial framework because no clear explanation is provided.

- the authors said that they study the spatial dispersal of languages along 10,000 years, to my understanding the vector field describes the change of the language between one exact moment of the past and $t=0$, which is supposed to be today.

Reviewer #2:

Remarks to the Author:

As I stated in my previous reviews of this paper, it is interesting, convincing, and historically significant in its conclusions. I am pleased to see that the authors have cut down the paper to deal with the four clearest examples, these being Indo-European, Sino-Tibetan, Bantu, and Arawak. The more troublesome Austroasiatic, Japonic and Oceanic examples have been removed, and I think this decision has added greatly to the clarity of the paper. It deserves to be published in Nature Communications.

My first comment is that the paper still needs a light level of English editing. I do not have time to do this on behalf of the authors, but perhaps I can use the abstract as an example of how some light editing might increase its clarity:

Here is the original abstract:

Reconstructing the spatial evolution of worldwide languages could shed light on understanding the global demic diffusions and cultural spreads. The phylogeographic approaches have been frequently used to infer the dispersal patterns of languages. However, they have shown some limitations primarily because the phylogenetic tree cannot properly capture the complex socio-cultural scenarios like contact-induced borrowings and areal diffusions of languages. Here, we introduced the language velocity field, which could be estimated directly from linguistic data without phylogenetic reconstruction, to enable the inference of the dispersal routes and centers of language families and groups in the geographic space. And its effectiveness and robustness have been carefully verified by both simulated and empirical validations. With the language velocity field estimation, we made inferences on the dispersal patterns of four language families and groups worldwide including around 700 languages. Our results showed that the dispersal routes of these languages were primarily compatible with the population activities inferred from ancient DNA and archaeological materials, and their dispersal centers were geographically proximate to the ancient homelands of agricultural or Neolithic cultures. Our findings highlight that the agricultural languages

dispersed along with demic diffusions and cultural spreads globally in the past 10,000 years. We expect that language velocity field estimation could greatly aid the spatial analysis of language evolution, and many more studies of demographic and cultural dynamics.

And here is how I would edit it:

Reconstructing the spatial evolution of languages worldwide can shed light on understanding global demic diffusions and cultural spreads. The phylogeographic approaches that have been frequently used to infer the dispersal patterns of languages show limitations, primarily because a phylogenetic tree cannot properly capture complex socio-cultural scenarios that involved contact-induced borrowing and areal diffusion of languages. Here, we introduce the language velocity field, which can be estimated directly from linguistic data without phylogenetic reconstruction, as a resource that can enable the inference of the dispersal routes and centers of language families and groups in geographic space. Its effectiveness and robustness have been carefully verified by both simulated and empirical validations. Using language velocity field estimations, we infer the dispersal patterns of four language families and groups worldwide, covering around 700 languages. Our results show that the dispersal routes of these languages were primarily compatible with human population spreads inferred from ancient DNA and archaeological materials, and their dispersal centers were geographically proximate to ancient homelands of agricultural (or Neolithic) cultures. Our findings highlight that agricultural languages dispersed with demic diffusions and cultural spreads on a global scale during the past 10,000 years. We expect that language velocity field estimation will aid greatly the spatial analysis of language evolution, with implications for studies of demographic and cultural dynamics.

Back to my commentary:

Figure 2 shows the proposed agricultural homeland in northern Amazonia for Arawak. This conflicts with text lines 184-186, where it is stated that "In addition, the language velocity field posited the dispersal of Arawak languages originated from the border of Peru, Brazil, and Bolivia in Western Amazonia, which was geographically close to the known ancient agricultural homeland of South America in the Andes". This statement implies a homeland much further to the south than shown on the map, which is what the archaeology would suggest. The map shows an area too far north. I note in Supplementary Notes 1 Table S2 that the Arawak homeland is put in the northern lowlands of Bolivia (upper Madeira River), which is precisely where I would expect it to be!

Likewise, lines 187-189 state "Moreover, in the case of Sino-Tibetan languages, their dispersal center was inferred in the Gansu province of China (Figure 2b). It was approximate to the geographic ranges of the Yangshao (7,000-5,000 years BP) and/or Majiayao (5,500-4,000 years BP) Neolithic cultures, although it was far from the ancient agricultural homelands known in the Yangzi and Yellow River Basins of China." Surely, Yangshao and Majiayao were centrally located in the Yellow River homeland of millet and pig agriculture? I cannot understand what is meant here, although, of course, the Yangzi is a different matter.

The discussion from lines 197 to 298 is highly technical, and I have no observations on it. Much the same applies to the materials and methods section. I can understand from lines 301-9 that the basic data come from a geographical plotting of cognate presences and absences, but I was puzzled by the statement (lines 304-6) "Lexical cognates of these language samples in each language family or group were binary-coded traits..." This sentence seems to confuse the concepts of cognate and language. How many cognate terms were used in the analysis, and from which proto-language levels were these cognates derived? In other words, how was a cognate defined? This might be explained in the supplementary data, but I think it should be clearer here in the main text.

Lines 449-40 state: "The diversity approach is an alternative phylogenetic tree-free approach and simply infers the location of the language homeland to the areas with the highest linguistic diversity." What is meant here by linguistic diversity? Does it relate to relative times of splitting from an inferred

phylogenetic family tree? (i.e., deeper-splitting subgroups are older)? I presume it is not simply related to number of languages.

I noticed in Supplementary Note 1 that phylogenetic discussions of Austroasiatic, Japonic and Oceanic are still mentioned, even through these groupings are no longer discussed in the main text.

Supplementary Notes 2: it is not clear to me that Supplementary sections 2 and 3 are really necessary (The interdisciplinary alignment of Genetics, Archaeology, and Linguistics; The Age-Area Hypothesis for inferring the language homeland). I think the observations made in this paper can stand quite well without them.

Peter Bellwood

Reviewer #3:

Remarks to the Author:

I find this study generally quite interesting, since the authors claim that they have developed a new method that allows to represent historical dynamics of individual languages in comparison with neighboring languages by multidimensional vectors, which can then be projected in lower-dimensional space in order to even infer the original locations from which the language family as a whole dispersed.

While interesting, I see some general problems with the study, mainly its fit with the journal where it was submitted to, and as a result, I recommend it to be rejected -- not because it is too low in quality, but rather because it is not a good fit with the journal, as I'll explain below.

Apart from this, I see some major and minor flaws, which I'll discuss below.

First, regarding the fit of the approach: What the authors propose is a methodological study, a new methodology of which they claim it outperforms established -- albeit controversial -- methods. In such a case, the journal where they submitted their study to, does not really qualify as a good fit, since we do not deal with new findings (they cannot be made until the method has been thoroughly evaluated) but rather with a new method that needs to be shown to work. For this reason, I think some journal like "Nature Methods" would be a much better fit here.

Second, if the authors accept that they need to convince us first that their method is useful and will enlarge our future knowledge about the spread of language families over time, they should please provide their method in a way that it can be replicated. As of now, we have a bunch of unrelated, badly documented R-scripts in a folder of 600 MB, that are hard to read and even harder to understand. Where is the vector estimation happening, what is the k you choose for the k -means languages that you select as neighbors, what is the impact of k on your results, etc. It makes me extremely nervous to see such a huge bunch of barely commented R-scripts that often do the same, but bear another name of another language family. This is definitely not how you make a new method successful. The least we would expect is a package in R with a tutorial that runs us through your code, for one language family, and then an extended tutorial with all four language families.

Third, speaking of four, I hate to say this, but I was reviewing this study before, not negatively, but pointing to the code, and to other issues. Interestingly, the number of language families has now dropped from 7 to 4. How the heck did that happen? How do the authors explain that they discard three language families now? I know having the same reviewers for the same paper across journals is annoying, but please, good scientific practice requires you to be transparent and tell us what happened here. Did you discard them, because they did not bring the results you hoped for?

Fourth, the claim of the method not using phylogenetic information is a bit exaggerated: we know geography correlates often with language relatedness (see for example here: <https://doi.org/10.1371/journal.pone.0265460>), so if geography explains the tree, you cannot say you do not use the tree if you use geography as a proxy for the construction of your vectors.

Fifth, the question of homeland has always been problematic, but if you already use data by Wichmann and Rama, you should also check the much simpler baseline published in Glottolog by now (www.pyglottolog.readthedocs.io/en/latest/homelands.html#module-pyglottolog.homelands). This method seems to work as well as the one by Wichmann and Rama, but it is even simpler, so I would say there's one more baseline to be tested. And when speaking of testing: why restrict your study to four datasets (or seven), if there are many more available in terms of phylogenies now, which are all with nicely coded cognate sets in standardized data formats (see e.g., <https://doi.org/10.1038/s41597-022-01432-0> for a very large collection of standardized data)? It seems the data has been cherry-picked to yield good results. Taking ten of the datasets in the Lexibank collection should not be difficult and would tell us much more clearly where we are with this new method.

Sixth, the method has the rather infelicitous name "language velocity field estimation", and I could not find any explanation why the authors chose to call it like that, since the name is very confusion and difficult to parse, and it does not really help to understand what the method could be about. I think in general it would be useful to 1) change the name to something that explains the method in a better way (dynamic trait vectors? I am not sure) and 2) to explain the method in much, much more detail. For this, figures would be needed that show how vectors for some of the traits are estimated, and the authors would need to also check the resulting vectors on an individual basis in order to see if they make sense.

Seventh, the authors praise their method for not needing trees, but at the same time, they do not tell the readers why trees are so useful: they tell us various scenarios of character evolution in a very transparent way, in which we have scenario and can plot how the trait evolved. Of course, this is not always done, but they should tell the readers to which the method they propose allows us to get some insights into the black box, since a simple black box, even if it works, is not satisfying from a scientific viewpoint, and we talk about scientific approaches here.

Eighth, and final point, the paper is not nice to read, the authors should check their wordings, which are often hard to follow, at times with flaws in grammar, and it would really profit from a complete overhaul and a thorough checking by a proof reader.

Due to all these reservations, I recommend that the paper be rejected, but I emphasize that it is not for poor quality, but for lack of fit. I look forward to see a new methods paper emerging from this, in which the authors work hard to share a useful new approach with the scientific world that they also evaluate rigorously against existing approaches. I am convinced they have the potential to turn their paper into such a study, and I am also very confident that this would be the right way to go, instead of trying to sell this as some study with new insights, or a study with a method that beats all existing approaches, since this is obviously not the case.

Response Letter to Reviewers

Replies to Reviewer 1:

Q1: The authors propose a vectorial framework to reconstruct the spatial dispersal of four language families around the world. The authors use a very wide range of methods that are borrowed from data science, physics and others from linguistics. I do not have the expertise to cover all of these methods, however the authors could help the reader understand if these methods are clustering algorithms, prediction methods, accuracy tests, etc. Some methods are called in the main text without further description, while some others are wrongly described, e.g. PCA is described in the main text as a similarity or clustering algorithm, actually PCA helps filtering out the least important features in order to describe a target variable in a space defined by superposition of few important features.

Replies to Q1:

We sincerely appreciate the invaluable suggestions provided by the reviewer. Our computational approach can be characterized as a kind of spatial reconstruction method that primarily encompasses other three distinct methods. The first one is the Principal Component Analysis (PCA) which is an unsupervised dimensionality reduction technique for rearranging linguistic traits into fewer more important new traits. The second one is the dynamic model consisting of ordinary differential equations for reconstructing the past states of linguistic traits. The third one is the geographic projection technique utilized for mapping the velocity vectors from the PC space into the geographic space. In the revised manuscript, we have modified unclear and problematic descriptions of our approaches and added more corresponding comprehensive explanations (*Lines 110-151* of the revised main text).

The reviewer has pointed out: “*PCA is described in the main text as a similarity or clustering algorithm, actually PCA helps filtering out the least important features in order to describe a target variable in a space defined by superposition of few important features*”. We are sorry for the imprecise descriptions of the PCA algorithm in the previous version of our manuscript. In this study, the PCA algorithm is not implemented to cluster language samples. Instead, it is used to reduce the dimension of linguistic traits by reassembling them into two important new traits (i.e., PC1 and PC2). Accordingly, each language sample can be visualized in the two-dimensional

PC space based on its PC1 and PC2 values. The Euclidean distances among pair-wise
language samples in the PC space (i.e., PCA-based distance) represent their linguistic
relatedness with each other. To be specific, the language samples sharing closer
linguistic relatedness tend to distribute closer in the PC space. Therefore, the
linguistic relatedness can be shown through the Euclidean distances among the
language samples in the PC space.

It is noted that utilizing the PCA-based distance metric to assess sample
relatedness is a prevailing practice in many studies within the fields of genetics and
linguistics [1-3]. Accordingly, we employ the PCA-based distance to quantify the
linguistic relatedness among language samples in this study. Following the reviewer's
comments, we have revised all the contents related to the PCA algorithm (*Lines*
*114-122* of the main text).

**Reference**

[1] Wang, Chuan-Chao, et al. "Genomic insights into the formation of human
populations in East Asia." *Nature* 591.7850 (2021): 413-419.

[2] Haak, Wolfgang, et al. "Massive migration from the steppe was a source for
Indo-European languages in Europe." *Nature* 522.7555 (2015): 207-211.

[3] Norvik, Miina, et al. "Uralic typology in the light of a new comprehensive
dataset." *Journal of Uralic Linguistics* 1.1 (2022): 4-42.

*Q2: The methods section is a repetition of the vague description of the tools made in*
*the main text and no further information is provided. The reader needs to get to*
*Supplementary Information #3 to finally get a technical description of the methods*
*that should actually appear in the Methods section. However, here the technical*
*details are not clearly expressed and the physical meaning of the vector is unclear.*
*Due to this, all the following results are unclear.*

**Replies to Q2:**

We appreciate these comments. Following the reviewer's comments, we have
rephrased some vague descriptions of our approach and added more technical

descriptions and key mathematical formulas in the Materials and Methods section.
 Considering the readability of the manuscript, detailed mathematical derivations and
 professional mathematical terminology descriptions have still been retained in
 Supplementary Note 3. Moreover, we have also provided a new schematic diagram
 (Figure 1 in the revised manuscript) to illustrate the rationale and procedure of our
 approach comprehensively. For the convenience of the reviewer, this figure is
 attached below namely Figure to Q2.

**Figure to Q2. Schematic diagram of language velocity field estimation (LVF) for**
 **inferring the dispersal trajectories and centers of languages.** The computational
 procedures of the LVF comprise two major steps. Subfigures (a) to (e) illustrate the
 first step which is to estimate a velocity field on the PC space to outline the diachronic
 evolutionary trajectories of linguistic traits that shape the observed linguistic
 relatedness. Subfigures (f) to (g) illustrate the second step, which is to project the
 velocity field from PC space into geographic space. Within the velocity field in
 geographic space, the directions of the velocity vectors compose a set of continuously
 changing trajectories that delineate from where these languages diffuse to their current
 locations. These procedures are exemplified using the Bantu language family.

Comprehensive insights into the underlying principles and computational steps can be
found in the Materials and Methods section, as well as Supplementary Note 1.

*Q3: The text is hard to read, mostly due the presence of many typos and other*
*grammar issues. Long sentences are used for speculative purposes, while key*
*methodological descriptions are narrowed down to few vague sentences.*

**Replies to Q3:**

In the revised manuscript, we corrected the typos and grammar errors and
modified several long and vague sentences. Furthermore, we engaged the AJE
language editing service to thoroughly polish our manuscript (ID: Q2K9ZRSF). To
make our methodological description clearer, we rephrased some vague sentences and
added detailed mathematical formulas and explanations for our approach in the
Materials and Methods section.

*Q4: I realize that the authors did a very hard work and that the storytelling is not*
*easy to unroll in a linear way. Still, I feel that the authors should make an effort to*
*simplify, correct and make the text clearer in order to be readable by an*
*interdisciplinary audience.*

**Replies to Q4:**

We sincerely appreciate the reviewer's comments. Considering the readability of
the interdisciplinary audience, we have rephrased the sentences in the manuscript to
enhance the clarity and comprehensibility of the narrative. Moreover, we have
rearranged the structure of our whole manuscript to improve its clarity and readability.

*Q5: typo in the abstract, the sentence “And its effectiveness and robustness have*
*been carefully verified by both simulated and empirical validations” starts with*
*“and”.*

**Replies to Q5:**

We greatly thank the reviewer for pointing this out. We have corrected this typo
in the abstract as shown in the *Line 35* of the revised main text.

*Q6: line 87: again the sentence starts with “and”*

**Replies to Q6**

This typo has been corrected in the revision.

*Q7: “And such relatedness could vary with time when languages continuously*
*dispersal into new regions.” sentence starts with “and” + dispersal is a noun, the*
*verb is disperse. The same is repeated in many other sentences, please correct.*

**Replies to Q7**

These grammatical errors have been corrected in the revision.

*Q8: line 106: “The Principal Component Analysis (PCA) is implemented to exhibit*
*the linguistic relatedness of present languages.” it is not clear on what variables the*
*PCA is implemented. PCA identifies the most important variables to explain the*
*variance of a target variable (in this case, I guess, the target variable is the*
*languages relatedness?). Clustering classification is a forthcoming step.*

**Replies to Q8:**

We appreciate these important comments. In our study, Principal Component
Analysis (PCA) has been applied to the binary-coded lexical trait, where the value 1

indicates the presence of the lexical trait in a language, while 0 signifies its absence.
 More specifically, our dataset is organized in the form of a matrix comprising binary
 values. The rows of this matrix correspond to diverse language samples, while the
 columns denote distinct binary-coded lexical traits, as illustrated in Table to Q8. In
 this study, both the empirical and simulated datasets adhere to this form.

The target variables derived from the PCA process are not referred to as linguistic
 relatedness. Instead, linguistic relatedness among language samples is represented by
 their Euclidean distances within the PC space. Specifically, in this study, PCA is
 employed to linearly transform lexical traits into two critical variables designated as
 PC1 and PC2. These PC1 and PC2 variables are the target variables extracted by the
 PCA algorithm. They represent the two most significant dimensions capable of
 capturing the primary variations within the original linguistic traits. Consequently, we
 can visually represent language samples based on their coordinates (PC1, PC2) within
 a two-dimensional PC space. In this space, language samples with closer linguistic
 relatedness are naturally distributed together. In such instances, the Euclidean
 distances among language samples within the PC space serve as a manifestation of
 their linguistic relatedness.

**Table to Q8.** The format of the linguistic dataset utilized in this study.

	Trait 1	Trait 2	Trait 3	...	Trait k
Language 1	0	1	0	...	1
Language 2	1	0	1	...	1
...
Language n	1	1	0	...	0

*Q9: subsection “Simulated validations for language velocity field estimation”. I*
 *really struggle here to understand what data did the authors use to validate their*
 *results. The dataset that is supposedly used as ground truth is also simulated by a*
 *phylogeographic algorithm. The authors claimed in the introduction that this*
 *method only captures vertical dependency of languages and not horizontal contacts*
 *and borrowings. I am confused about what is the contribution of this validation.*
 *Maybe the authors could add this discussion in the limitations of the study.*

**Replies to Q9:**

We are grateful for these comments. The reasons for utilizing the simulated
datasets in this study are given below:

**1. Simulated datasets with known dispersal centers can be used for model**
**validations**

The optimal validation for our methodology should be implemented relying on
benchmark datasets where the actual language dispersal centers are already
documented. These datasets enable us to validate our approach by comparing the
estimated dispersal center locations with the documented ones. Since empirical
datasets often lack precise information on the actual dispersal center locations,
validating our approach using empirical datasets is challenging due to the credibility
of the estimated dispersal center is hard to verify. Fortunately, a viable solution is
provided by simulated datasets from Wichmann et al. (2021) [1]. These simulated
datasets include known locations of true language dispersal centers, as they are
generated through a random walk model applied to a phylogenetic tree assigned with
given dispersal centers. Given the locations of the language dispersal centers are
known in these simulated datasets, they can serve as robust benchmarks for validating
our approach. In the previous manuscript, we extensively demonstrated the
effectiveness and robustness of our approach based on these simulated datasets.

**2. Simulated datasets are not generated by the phylogeographic approach**

We would like to clarify that the simulated datasets are not generated through the
phylogeographic approach but the random walk model. We understand that the
unclear descriptions in the previous manuscript may have led the reviewer to consider
these two approaches are the same. However, the phylogeographic approach is just a
specific application of the random walk model in the phylogenetic domain [2-3]. The
phylogeographic approach aims to backwardly reconstruct the language dispersal
center based on the locations of observed language samples assigned to a
phylogenetic tree. In contrast, the random walk model utilized in Wichmann et al.
(2021) [1] is employed to forwardly generate the locations of observed language
samples based on a phylogenetic tree assigned with a given language dispersal center.
As mentioned in Wichmann et al. (2021), the generation of the simulated datasets
follows below procedures:

*“...The simulation process can be summarized as follows. Movements are*

*constrained to any populated place on Earth, i.e. a place included in the*
*geonames.org database. A starting point is found by randomly choosing from this set*
*of populated places. At each time step there is a preset probability of moving to a new*
*place within a square containing at least ch populated places.....The kind of*
*movement we simulate here may be called a semi-random walk, since it is a kind of*
*random walk constrained to populated places.....Maps of all 1000 cases, showing*
*the homeland, intermediate stations, locations of current languages, and inferred*
*homelands similarly to Figure 2 below, as well as the script that produced the maps,*
*are provided in the electronic supplementary material (SI-11)....”*

Therefore, it is important to note that the simulated datasets are not produced
through the phylogeographic approach, even though the simulation process
incorporates the phylogenetic tree and random walk model.

**3. Simulated datasets as benchmarks for model comparisons**

**(i) Our approach and the phylogeographic approach share a common theoretical**
**foundation but employ distinct implementation strategies.** Both our approach and
phylogeographic approach involve two key steps in inferring language dispersal
through the diachronic evolution of linguistic traits (Figure 1 to Q9). The first step
entails delineating the diachronic evolutionary trajectories of linguistic traits that
contribute to linguistic relatedness among observed language samples. The second
step involves transforming these trajectories into language dispersal trajectories based
on the correlation between linguistic relatedness and language geography [2, 4].

However, these two approaches differ in their detailed strategies for implementing
these steps (Figure 1 to Q9). The primary distinctions revolve around how linguistic
relatedness is represented. Specifically, in the phylogeographic approach, linguistic
relatedness is represented by the phylogenetic tree, which captures only vertical
language divergence. In contrast, our approach measures linguistic relatedness
through the Euclidean distances among language samples in the two-dimensional PC
space (PCA-based distance). This method can capture both vertical divergence and
horizontal contact. We anticipate that our approach would perform similarly to the
phylogeographic approach when linguistic relatedness can be explained by the tree
model (Table to Q9). However, when linguistic relatedness cannot be fully explained
by the tree model, there is a notable difference between the two approaches (Table to

Q9).

To illustrate this, we conducted comprehensive simulated and empirical
comparisons between our approach and the phylogeographic approach. The results of
the comparisons are summarized in Figure 3 in the revised main text. For the
reviewer's convenience, we have attached this figure to this reply as Figure 2 to Q9. It
is important to note that Figure 2 to Q9 outlines the comparison results not only
between our approach and the phylogeographic approach but also against four other
spatial reconstruction approaches: the diversity approach, the minimal distance
approach, and the centroid approach. However, in this response, we focus solely on
the comparison between our approach and the phylogeographic approach to highlight
their similarities and differences (Figure 1 to Q9).

**(ii) Simulated comparisons when linguistic relatedness can be explained by the**
**tree model.** The simulated datasets can serve as benchmarks to compare the
performance between our approach and the phylogeographic approach when the
linguistic relatedness can be explained by the tree model. Due to simulated datasets
being generated based on a specific phylogenetic tree, the linguistic relatedness of the
simulated language samples is solely raised by the vertical divergence. In other words,
the linguistic relatedness among simulated language samples can be well captured by
the tree model. Therefore, based on the simulated datasets, the dispersal centers
inferred by the phylogeographic approach and our approach should be the same as
each other.

Fortunately, the simulated results indeed showed the same performance between
the phylogeographic approach and our approach (p -value > 0.05 ; Figures 2b1 to Q9).
More importantly, under the circumstance of the linguistic relatedness being solely
raised by vertical divergence, the phylogenetic tree and PCA-based distance
estimation can both adequately explain the linguistic relatedness (p -value < 0.05 ;
Figure 2b6 to Q9). It evidences that our approach and phylogeographic approach
indeed share the same theoretical foundation but with different implementations.

**(iii) Empirical comparisons using simulated results as baselines when linguistic**
**relatedness cannot be explained by the tree model.** The four empirical datasets can
be utilized for comparisons between our approach and the phylogeographic approach
when the linguistic relatedness cannot be explained by the tree model. Based on the

phylogenetic topology of simulated language samples as baseline (Figure 2b2 to Q9),
the phylogenetic topology of language samples in four empirical datasets utilized in
this study significantly deviates from the tree topology in this study (p -value < 0.05 ;
Figure 2b6 to Q9). It indicates that both vertical divergence and horizontal contact
could have contributed to the linguistic relatedness among these empirical language
samples. Accordingly, the phylogenetic tree cannot be able to adequately interpret the
linguistic relatedness within these four empirical cases. Under this circumstance, we
would anticipate different dispersal centers estimated by our approach and the
phylogeographic approach in empirical applications.

With the estimated difference in simulated comparisons as the baseline, the
empirical comparisons demonstrated a significant difference in performances between
our approach and the phylogeographic approach in Sino-Tibetan and Arawak (p -value
< 0.05 ; Figure 2a to Q9) languages. However, such difference was not observed in the
Bantu and Indo-European languages (p -value > 0.05 ; Figure 2a to Q9). The reason is
that for Bantu and Indo-European languages, PCA-based distance and phylogenetic
tree can both explain the linguistic relatedness among language samples (p -value $<$
0.05 ; Figure 2b6 to Q9). It indicates that the phylogenetic tree can explain the
linguistic relatedness under the influence of a certain degree of horizontal contact. In
contrast to Bantu and Indo-European languages, the comparison results showed that
PCA-based distance (Sino-Tibetan: p -value < 0.05 ; Arawak: p -value < 0.05 ; Figure
2b6 to Q9) could well explain the linguistic relatedness of Sino-Tibetan and Arawak
languages, while the phylogenetic tree cannot (Sino-Tibetan: p -value = 0.115; Arawak:
p -value = 0.121; Figure 2b6 to Q9). These empirical comparisons confirm that the
difference between our approach and the phylogeographic approach can be attributed
to the distinct strategies for representing linguistic relatedness.

**In summary, the simulated and empirical comparisons confirm that the**
**distinction between our approach and the phylogeographic approach is raised by**
**their different explanatory power for linguistic relatedness. To be specific, when**
**linguistic relatedness can be explained by the family-tree model, the performance**
**between the phylogeographic approach and our approach is identical. However,**
**when linguistic relatedness cannot be explained by the family-tree model, a**
**notable distinction would emerge between the phylogeographic approach and**
**our approach.** In the revision, all the aforementioned contents have been added to the
revised main text as shown in the *Lines 153-172 and Lines 210-303*.

**Table to Q9.** Expected performance between the phylogeographic approach and our
 approach utilizing simulated and empirical datasets.

		Simulated dataset	Empirical dataset	
Linguistic relatedness attribution		Vertical divergence	Vertical divergence	Horizontal contact
Whether the approaches can capture the divergence or contact	Phylogeographic approach	✓	✓	✗
	Language velocity field	✓	✓	✓
Equality of two approaches		=	≠	

**Figure 1 to Q9. Language velocity field estimation (LVF) shares the same**
 **foundation as the phylogeographic approach but with different implementation**
 **strategies.** Both LVF and phylogeographic approach entails two major steps to infer
 language dispersal pattern. The first is to depict the diachronic evolutionary
 trajectories of linguistic traits that shape the observed linguistic relatedness. The
 second is to transform these diachronic evolutionary trajectories of linguistic traits
 into language dispersal trajectories. In the phylogenetic tree, each language is
 determined by k linguistic traits. In the velocity field within PC space, each language
 is determined by PC1 and PC2 which are rearranged from the k linguistic traits
 through the PCA algorithm. The red number denotes a language. The black arrow
 signifies the evolutionary direction of linguistic traits in a language. The blue arrow
 represents the dispersal direction of a language. The red star denotes the estimated
 dispersal center.

**Figure 2 to Q9. The comparison between LVF and other spatial reconstruction**
 **approaches.** a) The dispersal centres of four empirical language families and groups
 inferred by five different approaches: language velocity field estimation (LVF),

phylogeographic approach (PhyloG), diversity approach (DIV), centroid approach
(Centr), and minimal distance approach (MD). b1) The density plot for the
distribution of differences between the coordinates of dispersal centres in the aspects
of longitude and latitude inferred from LVF and PhyloG based on 1,000 simulated
datasets. The p-value is calculated based on the Wilcoxon rank-sum test, where < 0.05
indicates that the difference between the inferred coordinates is significantly different
from zero. b2) The density plot for the average delta score of the languages whose
linguistic relatedness can be well-explained by the tree model. It was estimated from
200 bootstrap replicates on the simulated languages. b3) The density plot for the
distribution of the absolute differences in the aspects of longitude and latitude
between the coordinates of dispersal centres inferred from LVF and PhyloG based on
1,000 simulated datasets. b4) The linear relation between the average delta score and
the absolute difference of the longitude estimated from LVF and PhyloG. The orange
ribbon denotes the 95% confidence interval. b5) The linear relation between the
average delta score and the absolute difference of the latitude estimated from LVF and
PhyloG. The blue ribbon denotes the 95% confidence interval. b6) The table of the
delta score, estimated difference between LVF and PhyloG, and linguistic relatedness
explanatory power of PCA-based distance estimation and phylogenetic tree. The
p-value is calculated by the Wilcoxon rank-sum test where < 0.05 indicates the
significance of the delta score, estimated difference, and linguistic relatedness
explanatory power.

**Reference**

- [1] Wichmann, Søren, and Taraka Rama. "Testing methods of linguistic homeland
detection using synthetic data." *Philosophical Transactions of the Royal Society B*
376.1824 (2021): 20200202.
- [2] Bouckaert, Remco, et al. "Mapping the origins and expansion of the
Indo-European language family." *Science* 337.6097 (2012): 957-960.
- [3] Grollemund, Rebecca, et al. "Bantu expansion shows that habitat alters the route
and pace of human dispersals." *Proceedings of the National Academy of Sciences*
112.43 (2015): 13296-13301.
- [4] Koile, Ezequiel, et al. "Geography and language divergence: The case of Andic
languages." *Plos one* 17.5 (2022): e0265460.

*Q10: I would avoid the usage of the word “true”, unless there are striking evidences*
*of the coordinates of the language dispersal origin.*

**Replies to Q10:**

We appreciate this comment. As elucidated in our **Replies to Q9**, the most
significant characteristic of the simulated datasets is that they are generated based on
the given language dispersal centers. In other words, the actual locations of the
dispersal centers are already known within the simulated datasets. Following the
reviewer’s suggestion, we have corrected the word “*true*” as “*given*” in the revised
manuscript.

*Q11: what is the delta score of tree-likeness?*

**Replies to Q11:**

In the revision, we have added a comprehensive explanation of the delta score in
*Lines 253-257* of the revised main text. Here, we provide a brief description. The
delta score, denoted as δ score, serves as a widely used metric for quantifying the
likeness between the language phylogenetic topology and the tree topology in the
phylo-linguistics [1-3]. In other words, the delta score quantifies the degree of
linguistic relatedness of languages that can be explained by the tree model. The delta
score is calculated based on the distance among the languages, with a value ranging
from 0 to 1. A larger value of the delta score denotes that the language phylogenetic
topology is more compatible with the tree topology [4]. In other words, a larger value
of the delta score signifies that the linguistic relatedness is less affected by the
horizontal contacts and can be better explained by the tree model.

**Reference**

[1] Greenhill, Simon J., et al. "Evolutionary dynamics of language systems."
Proceedings of the National Academy of Sciences 114.42 (2017): E8822-E8829.

[2] Kolipakam, Vishnupriya, et al. "A Bayesian phylogenetic study of the Dravidian

language family." Royal Society open science 5.3 (2018): 171504.

[3] Birchall, Joshua, Michael Dunn, and Simon J. Greenhill. "A combined
comparative and phylogenetic analysis of the Chapacuran language family."
International Journal of American Linguistics 82.3 (2016): 255-284.

[4] Holland, Barbara R., et al. "δ plots: a tool for analyzing phylogenetic distance
data." Molecular biology and evolution 19.12 (2002): 2051-2059.

*Q12: the authors do not describe the data they used accurately. For instance, what*
*is a trait? What is a cognate? It is never stated.*

**Replies to Q12:**

We thank the reviewer for pointing this out. In this study, our datasets contain the
Indo-European, Sino-Tibetan, Bantu, and Arawak lexical cognate datasets derived
from the previous publications respectively [1-4]. These datasets contain several
lexical words following a specific wordlist such as Swadesh 100 or 200 wordlist.
Each word (item) contains different lexical cognates identified by linguistic experts,
which manifest the same meaning and similar sounds. Furthermore, each cognate has
been transformed into a binary-coded lexical trait where the value of 1 denotes the
presence of this cognate in the language, while 0 indicates its absence (an example of
cognate coding is shown in Table to Q12). Accordingly, the Indo-European dataset
contains 5,995 binary lexical cognates across 103 language samples; the Sino-Tibetan
dataset encompasses 949 binary lexical cognates across 109 Sino-Tibetan language
samples; the Bantu dataset comprises 3,859 binary lexical cognates across 420
language samples; Arawak dataset involves 694 binary lexical cognates across 60
language samples. The detailed cognate coding process for each case is described as
follows.

For the Indo-European lexical dataset, Bouckaert et al. compiled 207 lexical
items [1]. According to these lexical items, they identified 5,995 cognates across 103
Indo-European languages, which were further recoded as 5,995 binary-coded lexical
traits. Bouckaert et al. described their cognate coding process as follows: "We
recorded word forms and cognacy judgments across 207 meanings in 103

contemporary and ancient languages.... Cognate data were coded as binary
characters showing the presence or absence of a cognate set in a language. There
were 5995 cognate sets in total, with most meanings represented by several different
cognate sets. All cognate coding decisions were checked with published historical
linguistic sources (Table S1). The database contained 25908 cognate coded lexemes.
Of these, 67% came originally from ref. (17), 14% from ref. (16), and 19% were
newly compiled from published sources. Ref. (17) required considerable correction,
and changes were made to approximately 26% of coding decisions on individual
lexemes. Ref. (16) required corrections to only 0.5% of lexemes.”.

For the Sino-Tibetan lexical dataset, Zhang et al. compiled 90 lexical items from
the *Sino-Tibetan Etymological Dictionary and Thesaurus* (STEDT) project [5]. These
lexical items also appear in *Swadesh’s 100-word list* [6]. These selected lexical items
facilitated the identification of 949 cognates across 109 Sino-Tibetan languages,
which were then encoded as 949 binary-coded lexical traits. Zhang et al. described
their cognate coding process as below: “*The lexical root-meanings used in this study*
*came from the Sino-Tibetan Etymological Dictionary and Thesaurus (STEDT)*
*project1, which was developed by a number of experienced historical linguists led by*
*James A. Matisoff over a 30-year period (URL: <http://stedt.berkeley.edu/>).....To*
*minimize the word lateral transfers, in this study we chose only the words with*
*meaning inside the Swadesh 100-word list, since they are relatively resistant to*
*borrowing2.....In order to make sure that all the languages were comparable to each*
*other, we filtered only those languages with at least 90 lexical meanings of Swadesh*
*100-word list recorded (no matter whether an RM exists) and 30 – 120*
*RMs.....Finally, we retained 109 ST language samples with 949 binary-coded lexical*
*RMs for further phylogenetic analyses.”*

For the Bantu lexical dataset, Grollemund et al. compiled 100 lexical items from
the *Atlas Linguistique du GABon list* [7], of which 68 lexical items overlap with
*Swadesh’s 100-word list*. According to these lexical items, they recognized 3,859
cognates across 420 Bantu languages. These 3,859 cognates were further transformed
into 3,859 binary-coded lexical traits. Grollemund described their cognate coding
process as: “*For phylogenetic inference, we used a selection of 100 meanings*
*comprising a modified version of the Atlas Linguistique du GABon list (52). The Atlas*
*includes 159 meanings, and our sample of 100 meanings are those that are best*
*documented for the languages we studied.....We identified 3,859 cognate sets across*

*the n = 100 meanings. These were coded as binary characters for purposes of*
 *phylogenetic analysis.”*

For the Arawak lexical dataset, Walker et al. compiled *Swadesh’s 100-word list*
 and identified 694 cognates across 60 Arawak languages. Subsequently, these
 cognates were then recoded as 694 binary-coded lexical traits. Walker et al. described
 their cognate coding process as below: “*We compiled Swadesh [20] lists of 100*
 *common vocabulary items and scored cognate sets across 60 Arawak languages and*
 *dialects representing all the major branches of the Arawak language family (see*
 *electronic supplementary material, table S1).....We transformed coded cognates into*
 *binary codes for each variant with sites representing whether any particular cognate*
 *set is present (‘1’) or absent (‘0’) in that language..... The method yields 694 sites of*
 *which 88 per cent are complete.”*

According to the reviewer’s suggestions, we have incorporated the
 aforementioned contents about the cognate and binary-coded lexical trait in *Lines*
 *373-382* of the revised main text.

**Tabel to Q12.** Example of cognate coding using two lexical items (Mouth and Bone)
 for four languages: Apurina, Bare, Yavitero, and Palikur. Lexical lists (left table) are
 transformed into binary codes for each cognate variant with sites representing whether
 any particular cognate is present ("1") or absent ("0") in that language (right table).

	Lexical item	
	Mouth	Bone
Apurina	nama	api
Bare	numa	bani
Yavitero	numa	ihiu
Palikur	by	api

Transform
data into
binary
codings
→

Lexical trait	Mouth		Bone	
	A	B	A	B
Apurina	1	0	1	0
Bare	1	0	1	0
Yavitero	1	0	0	1
Palikur	0	1	1	0

**Reference**

- [1] Bouckaert, Remco, et al. "Mapping the origins and expansion of the
Indo-European language family." *Science* 337.6097 (2012): 957-960.
- [2] Zhang, Menghan, et al. "Phylogenetic evidence for Sino-Tibetan origin in
northern China in the Late Neolithic." *Nature* 569.7754 (2019): 112-115.
- [3] Grollemund, Rebecca, et al. "Bantu expansion shows that habitat alters the route
and pace of human dispersals." *Proceedings of the National Academy of Sciences*
112.43 (2015): 13296-13301.
- [4] Walker, Robert S., and Lincoln A. Ribeiro. "Bayesian phylogeography of the
Arawak expansion in lowland South America." *Proceedings of the Royal Society B:
Biological Sciences* 278.1718 (2011): 2562-2567.
- [5] Matisoff, James A. "Sino-Tibetan etymological dictionary and thesaurus
(STEDT)." Berkeley: Sino-Tibetan Etymological Dictionary and Thesaurus
Project.(stedt.berkeley.edu/dissemination/STEDT.pdf)[accessed on 18 October 2020]
(2015).
- [6] Swadesh, Morris. "Towards greater accuracy in lexicostatistic dating."
*International journal of American linguistics* 21.2 (1955): 121-137.
- [7] Hombert, Jean-Marie. "Atlas linguistique du Gabon." *Revue gabonaise des
Sciences de l'homme* 2 (1990): 37-42.

*Q13: "Third, the changes in the state frequencies of linguistic traits are*
*proportional to their sociolinguistic prestige in a certain area." I don't get the logic*
*of this sentence. What is the meaning of prestige here? The definition of prestige is*
*expressed only in the next paragraph, it should be introduced before going into*
*interpretations.*

**Replies to Q13:**

We thank the reviewer for pointing this out. This prestige parameter reflects the
social opportunities or convenience for individuals who speak a specific language
containing a particular trait state [1]. States of linguistic traits with higher prestige

would be more prevalent in future generations, while those with lower prestige would
be less prevalent. Accordingly, the prestige of a specific state in a linguistic trait can
be mathematically defined as the probability of this linguistic trait remaining in that
state after a unit of time. According to the reviewer's comment, we have modified the
corresponding section and rearranged the sequence of the paragraph related to the
prestige parameter as shown in *Lines 408-412* of the revised main text.

**Reference:**

[1] Abrams, Daniel M., and Steven H. Strogatz. "Modelling the dynamics of
language death." *Nature* 424.6951 (2003): 900-900.

*Q14: "It is noted that the larger length of the velocity vector of a language denotes*
*the more rapid change of this language during its evolution". The reader is*
*provided with no tools to understand this sentence. A schematic representation of a*
*vector could really help. E.g. what are the elements of a vector?*

**Replies to Q14:**

We appreciate your comment. In the revision, we have added a more
comprehensive schematic representation for the velocity vector in Figure 1d of the
revised main text. We also attach this subfigure related to the calculation of the
velocity vector at the end of this **Replies to Q14** (Figure to Q14).

As shown in Figure to Q14, we can see that each velocity vector contains two
aspects: direction and length. Each vector is calculated as the difference between the
past reconstructed and current trait states divided by the reconstruction time.
Accordingly, the direction of each vector signifies the direction of the diachronic
change of the linguistic traits in each language in the high-dimensional space and
low-dimensional PC space (i.e., 2-D PC plot). In short, the direction of each vector
depicts how the linguistic traits evolve into their current states. Moreover, when the
linguistic traits of a language undergo rapid evolution, its trait states should change
significantly over a given time period. Such change can be represented by the length
of the velocity vector visualized as an arrow in the high-dimensional space and
low-dimensional PC space. However, our study exclusively concentrates on the

language dispersal pattern which can be reflected solely by the directions of the
 velocity vectors. Accordingly, the lengths of the velocity vectors are actually not
 utilized in this study. Noting these, we have removed the descriptions about the
 lengths of velocity vectors in the revised manuscript.

**Figure to Q14.** Schematic diagram of the calculation of velocity vector.

*Q15: Does the PCA find only two components, or the authors found that more*
 *components did not lead to more variance explainability? Again, PCA here is*
 *presented as a tool to find similarities among datapoints, actually it is a*
 *rearrangement of the predictors of the model that tells what are the most important*
 *features in the model. The authors say nothing about all this. Projecting the points*
 *in to the PC space allows to visualize clusters, but actual clustering is performed by*
 *other tools, such as k-nearest-neighbors.*

**Replies to Q15:**

Thank you for your comments. We have three specific reasons for selecting only
 two principal components in this study which are explained below:

Firstly, visualizing the samples in a two-dimensional plane using two principal
components is a common and effective practice [1-3]. It enables a clear visualization
of the distribution of the data points in a two-dimensional plane. Accordingly, we also
selected two principal components and visualized the language samples in the
two-dimensional space.

Secondly, in the subsequent step of our approach, the language velocity field will
be projected from the PC space into the two-dimensional (i.e., longitude and latitude)
geographic space. By selecting the first two principal components, we ensure that the
PC space and the geographic space share an identical dimension, thereby preventing
the loss of information during geographic mapping. For instance, attempting to map a
three-dimensional language velocity field to a two-dimensional geographic space
would result in the loss of one crucial dimension of information regarding the
language velocity field. Nevertheless, the reviewer provided a novel insight into our
approach. It is that when the geographic coordinates of language samples have a
higher dimension, it would be prudent to retain more principal components for the
geographic mapping of the language velocity field.

Thirdly, according to the simulated validations, we found that relying on two
principal components was sufficient to estimate a reliable language velocity field in
the geographic map. Based on this language velocity field, we could accurately reflect
the language dispersal trajectories and centers. Consequently, we only selected two
principal components for the construction of the velocity field in this study.

Fourthly, we do not conduct the PCA algorithm to cluster or find similarities
among language samples. Actually, in this study, the PCA algorithm is only conducted
to recombine the original traits into two important traits. we plot each language
sample according its coordinate (PC1, PC2) in the 2-dimensional PC space. The
shorter Euclidean distances among language samples in PC space embody their higher
linguistic relatedness. However, if we aim to further identify which language samples
should be clustered together, we will need to employ other clustering approaches.
According to the reviewer's comments, we have revised the descriptions about the
PCA algorithm as shown in the *Lines 114-122* of the revised main text.

**Reference**

[1] Wang, Chuan-Chao, et al. "Genomic insights into the formation of human

populations in East Asia." Nature 591.7850 (2021): 413-419.

[2] Haak, Wolfgang, et al. "Massive migration from the steppe was a source for
Indo-European languages in Europe." Nature 522.7555 (2015): 207-211.

[3] Norvik, Miina, et al. "Uralic typology in the light of a new comprehensive
dataset." Journal of Uralic Linguistics 1.1 (2022): 4-42.

*Q16: it is not clear how the vectors are formed in the PC space. Up to my*
*understanding the PCA describes the datapoint with two components, hence I*
*expect to observe a single point with coordinates (PC1,PC2) in the PC space. By the*
*way, we cannot build a vector with one point. I understand from SI-3 that the*
*vectors are computed as the difference in the PC space of $X(t) - X(-m)$, where $t=0$*
*represents now and $t=-m$ represents a moment in the past. What is this moment in*
*the past? Then I read "Therefore, VI describes the change of the state frequencies*
*of language l in a unit of time.". what is the unit of time? Years, centuries?*

**Replies to Q16:**

We sincerely thank the reviewer for bringing up these important points. We
address the reviewer's concern as below.

**1. The derivation of the velocity vectors in PC space**

We agree with the reviewer that PCA can describe each current language sample
with two components PC1 and PC2. The PC1 and PC2 are derived by applying a
matrix $A_{2 \times n}$ (2 rows and n columns) to each current language sample $l_{current} = [x_1, \dots,$
$x_n]^T$ (n linguistic traits): $[PC1_{current}, PC2_{current}]^T = A l_{current} = A[x_1, \dots, x_n]^T$. It can be
regarded as projecting a n -dimensional vector into a 2-dimensional PC space as a
2-dimensional vector. Therefore, we can only observe a single language point with a
coordinate $(PC1_{current}, PC2_{current})$ in the PC space.

However, given a dynamic model [1-3], our approach can reconstruct the past
trait states for each language sample according to its current observed trait states
noted as $l_{past} = [y_1, \dots, y_n]^T$. When projecting current trait states for each language
sample into the PC space, we simultaneously project its past trait states into this PC

space as well: $[PC1_{past}, PC2_{past}]^T = Al_{past} = A[y_1, \dots, y_n]^T$. Therefore, we can observe
two points noted as $(PC1_{current}, PC2_{current})$ and $(PC1_{past}, PC2_{past})$ in the PC space,
where one represents the current trait states of this language sample, and another
represents its past trait states. By taking the difference between these two points
divided by the reconstruction time, we can derive a vector that describes the rate and
direction of the changes in the trait states of this language sample.

According to the reviewer's suggestions, the calculations of the vectors are
illustrated by a schematic diagram in Figure 1 in the revised main text. For the
convenience of the reviewer, we have attached the subfigure of Figure 1 related to the
calculation of velocity vectors below as Figure 1 to Q16.

**2. The definition of a unit of time**

**(i) The definition of a unit of time is identical to the one in the phylogenetic study.**

The velocity vector is calculated as the difference in the PC space of $X(0) - X(-m)$
divided by reconstruction time m , where $t = 0$ represents the present time, and $t = -m$
represents a moment in the past. Here, m denotes m units of times, and $-m$ thus
represents m units of times before the present time. Given that we often have limited
knowledge regarding the precise origin time of past languages, we thus define a unit
of time as one generation. It serves as a dimensionless time indicator representing the
period during which the linguistic traits in language accumulate one mutation. This
definition of the unit of time in our study is identical to the definition in the
phylogenetic tree where no exact time calibrations have been made (hereafter
non-time-calibrated phylogenetic tree).

To be specific, in a non-time-calibrated phylogenetic tree, the branch length
between a parent node and a child node (where the language is referred to as a node
for convenience hereafter) represents the time during which the child language has
evolved from its parent language. This branch length is typically represented by the
number of mutations that occurred in linguistic traits during the evolution of the child
language from its parent language. Because the longer evolutionary time of a
language results in more mutations being accumulated in linguistic traits (see Figure 2
to Q16 attached below) [4-5]. Under this circumstance, a unit of time is defined as the
period in which the linguistic traits of language undergo one mutation.

**(ii) A unit of time can be calibrated based on prior origin time.** This dimensionless

unit of time can be further converted into the exact period once given the precise
origin time of the parent language (see Figure 2 to Q16 attached below). For instance,
we assume that one branch length between a parent language $L5$ and a child language
$L4$ within a non-time-calibrated phylogenetic tree corresponds to 100 mutations (see
Figure 2 to Q16 attached below). Moreover, we assume that we also possess prior
knowledge about the precise origin time of that parent language, said 500 years ago.
Accordingly, we can calibrate the unit of time as $500/100 = 5$ years using the
commonly utilized strict molecular clock model in linguistics which assumes the
mutation rate is constant [3, 6]. According to this unit of time with exact time
calibration, we can calibrate all the branch lengths with exact periods in the
630 non-time-calibrated phylogenetic tree according to the times of mutations (see [Figure](#)
2 to Q16 attached below).

Similarly, the unit of time defined in our approach can also be converted to an
exact period in our approach, once we have prior knowledge about the precise origin
634 times of the past language samples. Nevertheless, the calibration of the unit of time in
our approach is not essential, since our approach is not designed to estimate the
divergence time of languages. It is just like the application of the phylogeographic
approach to a non-time-calibrated phylogenetic tree to solely infer the geographical
dispersal center of languages [7]. We have added the definition of unit of time into the
*Lines 441-443* of the revised main text.

**Figure 1 to Q16.** The calculation of velocity vectors in the PC space.

**Figure 2 to Q16.** Calibrating each branch length of the non-time-calibrated tree based
 on the mutation times and prior knowledge about language divergence times.

**Reference**

[1] Yang, Ziheng. "Maximum-likelihood estimation of phylogeny from DNA
 sequences when substitution rates differ over sites." *Molecular biology and evolution*
 10.6 (1993): 1396-1401.

[2] Penny, David, et al. "Mathematical elegance with biochemical realism: the
covarion model of molecular evolution." *Journal of Molecular Evolution* 53 (2001):
711-723.

[3] Zhang, Menghan, et al. "Phylogenetic evidence for Sino-Tibetan origin in
northern China in the Late Neolithic." *Nature* 569.7754 (2019): 112-115.

[4] Choudhuri, Supratim. *Bioinformatics for beginners: genes, genomes, molecular
evolution, databases and analytical tools*. Elsevier, 2014.

[5] Lewis, Paul O. "A genetic algorithm for maximum-likelihood phylogeny
inference using nucleotide sequence data." *Molecular biology and evolution* 15.3
(1998): 277-283.

[6] Chang, Will, et al. "Ancestry-constrained phylogenetic analysis supports the
Indo-European steppe hypothesis." *Language* (2015): 194-244.

[7] Walker, Robert S., and Lincoln A. Ribeiro. "Bayesian phylogeography of the
Arawak expansion in lowland South America." *Proceedings of the Royal Society B:
Biological Sciences* 278.1718 (2011): 2562-2567.

*Q17: what is the delta score and how is it computed? It is never stated in the text,*
*nor in the SI*

**Replies to Q17:**

We thank the reviewer for pointing this out. The rationale for the delta score has
been introduced in the **Replies to Q11**. Here, we offer a brief description of its
calculation procedure.

For any quarter of four elements $x, y, u,$ and $v,$ we denote $d_{xy|uv} = d_{xy} - d_{uv}$.
Then, the delta score is defined as the ratio $\delta_q = \frac{d_{xv|yu} - d_{xu|yv}}{d_{xv|yu} - d_{xy|uv}}$ [1]. This ratio
measures the tree-likeness of the quartet q that $\delta_q = 0$ if $d_{xv|yu} = d_{xu|yv} = d_{xy|uv}$
hold. The larger the value of δ_q indicates the less tree-like of q . The average value of
δ_q of the all-possible quarter of the language samples thus can serve as the metric to

quantify the overall tree-likeness of the language topology. In this study, the delta
score is calculated using the “*delta.plot*” function of the “*ape*” package [2]. The
corresponding contents have been included in *Lines 579-580* of the Materials and
Method section of the revised main text.

**Reference**

[1] Holland, Barbara R., et al. "δ plots: a tool for analyzing phylogenetic distance
data." *Molecular biology and evolution* 19.12 (2002): 2051-2059.

[2] Paradis, Emmanuel, and Klaus Schliep. "ape 5.0: an environment for modern
phylogenetics and evolutionary analyses in R." *Bioinformatics* 35.3 (2019): 526-528.

*Q18: Later on I read “In this study, we set $m = 1$.”, but no reason is given, nor the*
*unit of time is stated. One year? One century? Again, this is very opaque.*

**Replies to Q18:**

We appreciate the reviewer for pointing these out. As mentioned in the **Replies to**
**Q16**, a unit of time in this study is defined as one generation, which serves as a
dimensionless time indicator representing a period during which the linguistic traits in
language accumulate one mutation. This dimensionless unit of time can be converted
into an exact time once the precise divergence time of the past language sample is
given. However, the exact time calibration of the unit of time is not necessary in our
approach, since our approach is designed to infer the dispersal pattern of languages
rather than their origin time.

In this study, the setting of $m = 1$ is chosen based on the results of both empirical
and simulated validations. To be specific, in simulated validations, we demonstrated
that relying on the setting $m = 1$ could estimate a reliable language velocity field in
the geographic space. Based on this language velocity field, the estimated language
dispersal center shows no significant difference from the prior given dispersal center
(Figure 1 to Q18).

Without a loss of generality, we also tested the robustness of the language
velocity field estimated through different settings of m in simulated validations. The

results indicate that there are no significant differences among the language velocity
 fields estimated through different settings of m (Figure 2 to Q18). These results
 indicate that the rate of change of linguistic traits can remain relatively constant
 during different evolutionary periods. It is compatible with the rate assumption of the
 widely-used molecular clock model in linguistics that postulates the evolutionary rate
 of linguistic traits is constant [1-2]. In other words, the velocity vector is almost
 unchanged either setting $m = 1$ or setting m as other different reconstruction times.
 Therefore, it is feasible to estimate the velocity vector for representing the diachronic
 change in linguistic traits by setting $m = 1$.

According to the simulated validations, we further set $m = 1$ in the empirical
 applications. Without a loss of generality, we also tried different parametric settings of
 716 m in the empirical applications. The results also suggested that the language velocity
 field was robust under different settings of m (Figure 3 to Q18), and all could identify
 the language dispersal centers that can be supported by genetic and archaeological
 evidence. Based on all these empirical and simulated validations, we ultimately set m
 $= 1$ as the default parametric value in our approach.

**Figure 1 to Q18. The simulated validation for the effectiveness of the language**
 **velocity field estimation (LVF) under different parametric settings.** The
 probability density plot demonstrates the distributions of the errors of the longitude
 and latitude respectively between the true and inferred language dispersal center
 estimated from 1,000 simulated datasets under different parametric settings. These

parameters are the number of the grid points $n.grid$ ($n.grid = 50, 100, 200, 300, 400,$
 and 500); the number of the nearest neighbors k ($k = 2, 4, 6, \dots,$ and 18); mutation rate
 of Poisson process λ ($\lambda = 0.1, 0.5, 1, 5,$ and 10); reconstruction time m ($m = 1, 3, 5, 7,$
 and 9). We set the default parametric values as $n.grid = 300, k = 4, \lambda = 1,$ and $m = 1$
 when varying across the settings of these parameters respectively. The black texts are
 the p -value of the statistical significance of the error derived from the Wilcoxon
 rank-sum test. p -value > 0.05 denotes the statistical non-significance of the error
 (significantly equal to 0).

**Figure 2 to Q18. The simulated validation for the robustness of the language**
 **velocity field estimation (LVF) under different parametric settings.** The
 probability density plot demonstrates the distribution of the average cosine similarity
 between language velocity fields estimated from 1,000 simulated datasets under
 different parametric settings. The parameters are the number of the nearest neighbors
 k ($k = 2, 4, 6, \dots,$ and 18); mutation rate of Poisson process λ ($\lambda = 0.1, 0.5, 1, 5,$ and
 10); reconstruction time m ($m = 1, 3, 5, 7,$ and 9). We set the default parametric values
 as $k = 4, \lambda = 1,$ and $m = 1$ when varying across the settings of these parameters
 respectively. The black texts are the p -value of the statistical significance of this

average similarity derived from the Wilcoxon rank-sum test. p -value < 0.05 denotes
 the statistical significance of this average similarity (significantly not equal to 0).

**Figure 3 to Q18. The empirical validation for the robustness of the language**
 **velocity field estimation (LVF) against the setting of the reconstruction time.** The
 probability density plot demonstrates the distribution of the cosine similarity among
 the language velocity vectors calculated under different settings of reconstruction time
 754 m ($m = 1, 3, 5, 7, \text{ and } 9$) before the current time in four language families and groups.
 We set the default parametric values as $k = 10$ and $\lambda = 1$ when varying across the
 settings of m . The black texts are the average similarity of the distribution of
 similarity and the p -value of the statistical significance of this average similarity
 derived from the permutation test (Permutation Times = 500). The average similarity
 ranges from 0 to 1, where 1 denotes that these two velocity fields are most similar and
 0 is dissimilar. p -value < 0.05 denotes the statistical significance of the average
 similarity.

**Reference**

[1] Zhang, Menghan, et al. "Phylogenetic evidence for Sino-Tibetan origin in
 northern China in the Late Neolithic." Nature 569.7754 (2019): 112-115.

[2] Chang, Will, et al. "Ancestry-constrained phylogenetic analysis supports the
Indo-European steppe hypothesis." *Language* (2015): 194-244.

*Q19. I do not understand the physical meaning of this vectorial framework because*
*no clear explanation is provided.*

**Replies to Q19:**

To provide a clearer explanation of our approach, we have added more detailed
explanations of our approach in the revised main text (*Lines 109-151*). Moreover, we
have also added detailed mathematical formulas of our approach in the Materials and
Methods section. As the supplementary, we have also redrawn the schematic diagram
presented as Figure 1 to Q19 (also referred to as Figure 1 in the revised main text) to
visually elucidate the rationale and calculation procedure of our approach.

**Our approach shares the same theoretical foundation as the phylogeographic**
**approach but with different implementation strategies.** As the most prevailing
approach, the phylogeographic approach performs two major steps to infer language
dispersal patterns. The first is to obtain a phylogenetic tree to delineate the
evolutionary trajectories of linguistic traits that shape the observed linguistic
relatedness (Figure 2 to Q19) [1-3]. The second is to project the phylogenetic tree into
the geographic space based on the correlation between linguistic relatedness and
language geography (Figure 2 to Q19) [1-4]. With the projection, evolutionary
trajectories of linguistic traits can be transformed into language dispersal trajectories.
Our approach shares the similar two major steps as the phylogeographic approach that
infers language dispersal through the diachronic evolution of linguistic traits (Figure 2
to Q19). However, our approach employs different strategies to carry out these two
steps compared to the phylogeographic approach.

**The velocity field in PC space delineates the diachronic evolutionary**
**trajectories of linguistic traits that shape the observed linguistic relatedness.** Our
approach conducts the PCA-based distance rather than a phylogenetic tree to
represent linguistic relatedness. Specifically, the PCA algorithm is conducted to
rearrange the lexical traits into two principal components namely PC1 and PC2.
According to PC1 and PC2, the distribution of language samples can be visualized in

the PC space. The shorter distances among language samples in the PC space imply
their higher linguistic relatedness. In parallel, the language velocity vector is
estimated to demonstrate the direction of the average change of trait states for each
language sample in a unit of time. With the past trait states reconstructed by the
dynamic model, the velocity vector can be calculated by dividing the diachronic
changes in trait states of each language sample by the m unit of time. This velocity
vector depicts how the linguistic traits in a language sample evolve into their current
states. By mapping these velocity vectors into the PC space, a language velocity field
can be derived on the PC space to delineate the diachronic evolutionary trajectories of
linguistic traits that shape the observed linguistic relatedness. This velocity field in PC
space functions similarly to the phylogenetic tree in the phylogeographic approach.

**Projecting velocity field into geographic space to transform the evolutionary**
**trajectories of linguistic traits into the language dispersal trajectories.** Based on
the correlation between observed linguistic relatedness and language geography, we
further project each velocity vector from PC space into geographic space utilizing
kernel projection [3]. The rationale of this projection is to search for the velocity
vector in the geographic space ensuring that its correlation with language geography
closely matches with its correlation with linguistic relatedness. With the kernel
projection, the vector directions in the geographic space, which compose a set of
trajectories, render from where the observed language samples diffuse into their
current locations. This geographic projection of the velocity field is similar to the
projection of the phylogenetic tree into the geographic space to outline the dispersal
trajectories in the phylogeographic approach.

**Figure 1 to Q19. Schematic overview of the language velocity field estimation**
 **(LVF) for inferring the dispersal trajectories and centers of languages.** The
 computational procedures of the LVF comprise two major steps. Subfigures (a) to (e)
 illustrate the first step which is to estimate a velocity field on the PC space to outline
 the diachronic evolutionary trajectories of linguistic traits that shape the observed
 linguistic relatedness. Subfigures (f) to (g) illustrate the second step, which is to
 project the velocity field from PC space into geographic space. Within the velocity
 field in geographic space, the directions of the velocity vectors compose a set of
 continuously changing trajectories that delineate from where these languages diffuse
 to their current locations. These procedures are exemplified using the Bantu language
 family. Comprehensive insights into the underlying principles and computational
 steps can be found in the Materials and Methods section, as well as Supplementary
 Note 1.

**Figure 2 to Q19. Language velocity field estimation (LVF) shares the same**
**foundation as the phylogeographic approach but with different implementation**
**strategies.** Both LVF and phylogeographic approach entails two major steps to infer
language dispersal pattern. The first is to depict the diachronic evolutionary
trajectories of linguistic traits that shape the observed linguistic relatedness. The
second is to transform these diachronic evolutionary trajectories of linguistic traits
into language dispersal trajectories. In the phylogenetic tree, each language is
determined by k linguistic traits. In the velocity field within PC space, each language
is determined by PC1 and PC2 which are rearranged from the k linguistic traits
through the PCA algorithm. The red number denotes a language. The black arrow
signifies the evolutionary direction of linguistic traits in a language. The blue arrow
represents the dispersal direction of a language. The red star denotes the estimated
dispersal center.

**Reference**

[1] Bouckaert, Remco, et al. "Mapping the origins and expansion of the
Indo-European language family." *Science* 337.6097 (2012): 957-960.

[2] Grollemund, Rebecca, et al. "Bantu expansion shows that habitat alters the route
and pace of human dispersals." *Proceedings of the National Academy of Sciences*
112.43 (2015): 13296-13301.

[3] Currie, Thomas E., et al. "Cultural phylogeography of the Bantu Languages of
sub-Saharan Africa." *Proceedings of the Royal Society B: Biological Sciences*
280.1762 (2013): 20130695.

[4] Koile, Ezequiel, et al. "Geography and language divergence: The case of Andic
languages." *Plos one* 17.5 (2022): e0265460.

[5] La Manno, Gioele, et al. "RNA velocity of single cells." *Nature* 560.7719 (2018):
494-498.

*Q20: the authors said that they study the spatial dispersal of languages along*
*10,000 years, to my understanding the vector field describes the change of the*
*language between one exact moment of the past and $t=0$, which is supposed to be*
*today.*

**Replies to Q20:**

We are grateful for the reviewer's comments. The reason why we mentioned
10,000 years in the main text is that all four language families and groups utilized in
this study originated within the last 10,000 years. For the Indo-European languages,
different phylogenetic studies have reported that their origin time could be either
approximately 8,000 to 9,500 years ago [1] or approximately 6,000 years ago [2]. For
the Sino-Tibetan languages, its initial divergence has been estimated to occur between
4,000 to 8,000 years ago [3-5]. The origin of the Bantu languages has been traced
back to roughly 5,000 years ago [6]. Although the detailed origin time of the Arawak
languages remains unclear, its origin is interlinked with the agricultural advancement
in lowland South America around 5,000 years ago [7-8]. Consequently, the origin of
Arawak languages should have dated at most 5,000 years ago. Overall, 10,000 years

is the upper limit of the origin time for these four language families and groups.

**Reference**

[1] Bouckaert, Remco, et al. "Mapping the origins and expansion of the
Indo-European language family." *Science* 337.6097 (2012): 957-960.

[2] Chang, Will, et al. "Ancestry-constrained phylogenetic analysis supports the
Indo-European steppe hypothesis." *Language* (2015): 194-244.

[3] Zhang, Menghan, et al. "Phylogenetic evidence for Sino-Tibetan origin in
northern China in the Late Neolithic." *Nature* 569.7754 (2019): 112-115.

[4] Zhang, Hanzhi, et al. "Dated phylogeny suggests early Neolithic origin of
Sino-Tibetan languages." *Scientific Reports* 10.1 (2020): 20792.

[5] Sagart, Laurent, et al. "Dated language phylogenies shed light on the ancestry of
Sino-Tibetan." *Proceedings of the National Academy of Sciences* 116.21 (2019):
10317-10322.

[6] Grollemund, Rebecca, et al. "Bantu expansion shows that habitat alters the route
and pace of human dispersals." *Proceedings of the National Academy of Sciences*
112.43 (2015): 13296-13301.

[7] Diamond, Jared, and Peter Bellwood. "Farmers and their languages: the first
expansions." *Science* 300.5619 (2003): 597-603.

[8] Clement, Charles Roland, et al. "Crop domestication in the upper Madeira River
basin." *Boletim do Museu Paraense Emílio Goeldi. Ciências Humanas* 11 (2016):
193-205.

**Replies to Reviewer 2:**

*Q1: As I stated in my previous reviews of this paper, it is interesting, convincing,*
*and historically significant in its conclusions. I am pleased to see that the authors*
*have cut down the paper to deal with the four clearest examples, these being*
*Indo-European, Sino-Tibetan, Bantu, and Arawak. The more troublesome*
*Austroasiatic, Japonic and Oceanic examples have been removed, and I think this*
*decision has added greatly to the clarity of the paper. It deserves to be published in*
*Nature Communications. My first comment is that the paper still needs a light level*
*of English editing. I do not have time to do this on behalf of the authors, but*
*perhaps I can use the abstract as an example of how some light editing might*
*increase its clarity:*

**Replies to Q1:**

We are deeply grateful for the reviewer's great support and affirmation of our
work. Moreover, we also would like to express our sincere appreciation for the
reviewer personally revising our abstract. According to this valuable example of
revision, we have carefully revised our manuscript. This revision involves correcting
many typos and grammatical errors, and rephrasing some lengthy and vague sentences.
Moreover, we also engaged the AJE language editing service to thoroughly edit the
language of our manuscript (ID: Q2K9ZRSF). We expect that our revisions could
enhance the readability and clarity of our manuscript for native English speakers.

*Q2: Figure 2 shows the proposed agricultural homeland in northern Amazonia for*
*Arawak. This conflicts with text lines 184-186, where it is stated that " In addition,*
*the language velocity field posited the dispersal of Arawak languages originated*
*from the border of Peru, Brazil, and Bolivia in Western Amazonia, which was*
*geographically close to the known ancient agricultural homeland of South America*
*in the Andes". This statement implies a homeland much further to the south than*
*shown on the map, which is what the archaeology would suggest. The map shows*
*an area too far north. I note in Supplementary Notes 1 Table S2 that the Arawak*
*homeland is put in the northern lowlands of Bolivia (upper Madeira River), which*
*is precisely where I would expect it to be!*

**Replies to Q2:**

We are sincerely grateful for the reviewer to point these out. According to the
reviewer's suggestions, we found inaccuracies in our descriptions regarding the origin
of Arawak languages near the Andes, since their estimated dispersal center was indeed
located too far from the Andes foothills. As mentioned by the reviewer, the dispersal
center of Arawak languages estimated by our approach is located in the upper
Madeira River basin within the northern lowlands of Bolivia. Accordingly, we further
made some literature investigations about the upper Madeira River basin.

To our knowledge, the Madeira River rises from the Andes and flows through a
larger part of the Southwestern Amazonian [1]. The upper Madeira River basin, which
has raised numerous complex Neolithic Societies, has long been regarded as an
important homeland of ancient agriculture in lowland South America [2]. In this area,
plenty of crops have been domesticated, such as manioc, peanuts, peach palms, coca,
and tobacco. It is noted that the estimated dispersal center of the Arawak language is
located in the upper Madeira River basin. This estimation implies that the Arawak
language origin is associated with the agricultural origin in Southwestern Amazonian.
Accordingly, we revise the sentences of *Lines 184-186* in the original main text into:
*"In addition, the LVF showed the dispersal of Arawak languages originating from*
*the northern lowlands of Bolivia in the upper Madeira River basin, which is an*
*important homeland of ancient agriculture in lowland South America."* as shown in
*Lines 202-204* of the revised main text.

**Reference**

[1] Clement, Charles Roland, et al. "Crop domestication in the upper Madeira River
basin." *Boletim do Museu Paraense Emílio Goeldi. Ciências Humanas* 11 (2016):
193-205.

[2] Piperno, Dolores R. "The origins of plant cultivation and domestication in the
New World tropics: patterns, process, and new developments." *Current anthropology*
52.S4 (2011): S453-S470.

*Q3: Likewise, lines 187-189 state " Moreover, in the case of Sino-Tibetan*
*languages, their dispersal center was inferred in the Gansu province of China*
*(Figure 2b). It was approximate to the geographic ranges of the Yangshao*
*(7,000-5,000 years BP) and/or Majiayao (5,500-4,000 years BP) Neolithic cultures,*
*although it was far from the ancient agricultural homelands known in the Yangzi*
*and Yellow River Basins of China." Surely, Yangshao and Majiayao were centrally*
*located in the Yellow River homeland of millet and pig agriculture? I cannot*
*understand what is meant here, although, of course, the Yangzi is a different matter.*

**Replies to Q3:**

We greatly appreciate the reviewer for bringing these points out. The original
intention of our statement was to express that the dispersal and origin of Sino-Tibetan
languages appear to have stronger connections with the agriculture that originated in
the Yellow River basin rather than the Yangzi River basin.

Early farming in China can be divided into two distinct attributes. One originated
in the Yellow River basin with a focus on millet cultivation, while another one was
developed in the Yangzi River basin with a focus on rice cultivation [1].
Geographically located in the center of the Yellow River basin, Yangshao, and
Majiayao Neolithic cultures were predominantly engaged in millet cultivation, as
evidenced by the archaeological materials [2-3]. Therefore, the estimated
Sino-Tibetan language dispersal center located in the geographic ranges of Yangshao
and Majiayao Neolithic cultures indicates that the Sino-Tibetan languages could have
dispersed with the spread of millet from the Yellow River basin rather than the Yangzi
River basin.

However, according to the reviewer’s suggestion, we think that it is not necessary
to mention the agriculture in the Yangzi River basin in this study which is not relevant
to the case of Sino-Tibetan languages. The agriculture in Yangzi River should be
another story in another research. Accordingly, we have revised the sentences in *Lines*
*187-189* of the original main text as: “***Moreover, in the case of Sino-Tibetan***
***languages, their dispersal centre was inferred to be located in the Gansu Province***
***of China (Figure 2b). This centre is situated within the geographic ranges of the***
***Yangshao (7,000-5,000 years BP) and/or Majiayao (5,500-4,000 years BP) Neolithic***
***cultures 6 in the ancient agricultural homeland of China, the Yellow River plains.***”
in the *Lines 195-198* of the revised main text.

**Reference**

[1] Deng, Zhenhua, et al. "From early domesticated rice of the middle Yangtze Basin
to millet, rice and wheat agriculture: Archaeobotanical macro-remains from Baligang,
Nanyang Basin, Central China (6700–500 BC)." *PLoS One* 10.10 (2015): e0139885.

[2] Sagart, Laurent, et al. "Dated language phylogenies shed light on the ancestry of
Sino-Tibetan." *Proceedings of the National Academy of Sciences* 116.21 (2019):
10317-10322.

[3] Zhang, Menghan, et al. "Phylogenetic evidence for Sino-Tibetan origin in
northern China in the Late Neolithic." *Nature* 569.7754 (2019): 112-115.

*Q4: The discussion from lines 197 to 298 is highly technical, and I have no*
*observations on it. Much the same applies to the materials and methods section. I*
*can understand from lines 301-9 that the basic data come from a geographical*
*plotting of cognate presences and absences, but I was puzzled by the statement*
*(lines 304-6) "Lexical cognates of these language samples in each language family*
*or group were binary-coded traits..." This sentence seems to confuse the concepts of*
*cognate and language. How many cognate terms were used in the analysis, and*
*from which proto-language levels were these cognates derived? In other words, how*
*was a cognate defined? This might be explained in the supplementary data, but I*
*think it should be clearer here in the main text.*

**Replies to Q4:**

We appreciate these valuable comments. In this study, we have used four lexical
datasets encompassing 103 Indo-European, 109 Sino-Tibetan, 420 Bantu, and 60
Arawak languages, respectively, which were derived from previously published works
[1-4]. These lexical datasets are constructed upon the foundation of cognates (also
referred to as cognate sets) which are varied word expressions for a particular lexical
item (meaning) across diverse languages. These linguistic expressions (cognates) for
the same lexical item have been identified as being inherited from a common ancestor.
Within each lexical dataset, every linguistic expression (cognate) has been recorded as
a binary lexical trait, where a value of 1 indicates its presence in a language, while 0
indicates its absence.

To be specific, for the Indo-European lexical dataset, Bouckaert et al. compiled
207 lexical items [1] which facilitated the identification of 5,995 lexical cognates
across 103 Indo-European languages. These cognates were further recoded into 5,995
binary-coded lexical traits. Bouckaert et al. described their cognate coding process as
follows: *"We recorded word forms and cognacy judgments across 207 meanings in*
*103 contemporary and ancient languages.... Cognate data were coded as binary*
*characters showing the presence or absence of a cognate set in a language. There*
*were 5995 cognate sets in total, with most meanings represented by several different*
*cognate sets. All cognate coding decisions were checked with published historical*
*linguistic sources (Table S1). The database contained 25908 cognate-coded lexemes.*
*Of these, 67% came originally from ref. (17), 14% from ref. (16), and 19% were*
*newly compiled from published sources. Ref. (17) required considerable correction,*

*and changes were made to approximately 26% of coding decisions on individual*
*lexemes. Ref. (16) required corrections to only 0.5% of lexemes.”.*

For the Sino-Tibetan lexical dataset, Zhang et al. compiled 90 lexical items from
the *Sino-Tibetan Etymological Dictionary and Thesaurus* (STEDT) project [5]. These
lexical items can be also found in *Swadesh’s 100-word list* [6]. These chosen lexical
items led to the detection of 949 cognates across 109 Sino-Tibetan languages, which
were then encoded as 949 binary-coded lexical traits. Zhang et al. described their
cognate coding process as below: “*The lexical root-meanings used in this study came*
*from the Sino-Tibetan Etymological Dictionary and Thesaurus (STEDT) project1,*
*which was developed by a number of experienced historical linguists led by James A.*
*Matisoff over a 30-year period (URL: <http://stedt.berkeley.edu/>).....To minimize the*
*word lateral transfers, in this study we chose only the words with meaning inside the*
*Swadesh 100-word list since they are relatively resistant to borrowing2.....In order*
*to make sure that all the languages were comparable to each other, we filtered only*
*those languages with at least 90 lexical meanings of the Swadesh 100-word list*
*recorded (no matter whether an RM exists) and 30 – 120 RMs.....Finally, we*
*retained 109 ST language samples with 949 binary-coded lexical RMs for further*
*phylogenetic analyses.”*

For the Bantu lexical dataset, Grollemund et al. selected 100 lexical items from
the *Atlas Linguistique du GABon list* [7], of which 68 lexical items overlap with
*Swadesh’s 100-word list*. According to these lexical items, they recognized 3,859
cognates across 420 Bantu languages. These cognates were further transformed into
3,859 binary-coded lexical traits. Grollemund described their cognate coding process
as: “*For phylogenetic inference, we used a selection of 100 meanings comprising a*
*modified version of the Atlas Linguistique du GABon list (52). The Atlas includes 159*
*meanings, and our sample of 100 meanings are those that are best documented for the*
*languages we studied.....We identified 3,859 cognate sets across the n = 100*
*meanings. These were coded as binary characters for purposes of phylogenetic*
*analysis.”*

For the Arawak lexical dataset, Walker et al. compiled *Swadesh’s 100-word list*
and identified 694 cognates across 60 Arawak languages. Subsequently, these
cognates were then recoded as 694 binary-coded lexical traits. Walker et al. described
their cognate coding process as below: “*We compiled Swadesh [20] lists of 100*

*common vocabulary items and scored cognate sets across 60 Arawak languages and*
*dialects representing all the major branches of the Arawak language family (see*
*electronic supplementary material, table S1).....We transformed coded cognates into*
*binary codes for each variant with sites representing whether any particular cognate*
*set is present ('1') or absent ('0') in that language..... The method yields 694 sites of*
*which 88 per cent are complete.”*

According to the reviewer’s suggestions, we have revised the corresponding
contents as shown in the *Lines 373-382* of the revised main text.

**Reference**

[1] Bouckaert, Remco, et al. "Mapping the origins and expansion of the
Indo-European language family." *Science* 337.6097 (2012): 957-960.

[2] Zhang, Menghan, et al. "Phylogenetic evidence for Sino-Tibetan origin in
northern China in the Late Neolithic." *Nature* 569.7754 (2019): 112-115.

[3] Grollemund, Rebecca, et al. "Bantu expansion shows that habitat alters the route
and pace of human dispersals." *Proceedings of the National Academy of Sciences*
112.43 (2015): 13296-13301.

[4] Walker, Robert S., and Lincoln A. Ribeiro. "Bayesian phylogeography of the
Arawak expansion in lowland South America." *Proceedings of the Royal Society B:*
*Biological Sciences* 278.1718 (2011): 2562-2567.

[5] Matisoff, James A. "Sino-Tibetan etymological dictionary and thesaurus
(STEDT)." Berkeley: Sino-Tibetan Etymological Dictionary and Thesaurus
Project.(stedt.berkeley.edu/dissemination/STEDT.pdf)[accessed on 18 October 2020]
(2015).

[6] Swadesh, Morris. "Towards greater accuracy in lexicostatistic dating."
*International journal of American linguistics* 21.2 (1955): 121-137.

[7] Hombert, Jean-Marie. "Atlas linguistique du Gabon." *Revue gabonaise des*
*Sciences de l'homme* 2 (1990): 37-42.

*Q5: Lines 449-40 state: "The diversity approach is an alternative phylogenetic*
*tree-free approach and simply infers the location of the language homeland to the*
*areas with the highest linguistic diversity." What is meant here by linguistic*
*diversity? Does it relate to relative times of splitting from an inferred phylogenetic*
*family tree? (i.e., deeper-splitting subgroups are older)? I presume it is not simply*
*related to number of languages.*

**Replies to Q5:**

We are sorry for the lack of clarity regarding the definition of linguistic diversity.
As the reviewer correctly mentioned, linguistic diversity is not determined solely by
the number of languages. As described by Wichmann and Sapir [1-3], the level of
linguistic diversity is determined by the degree of differentiation among languages
within a specific geographical area. Higher linguistic diversity indicates greater
dissimilarities among the languages within that region. Consequently, even if there is
a large number of languages in a particular geographic area, the linguistic diversity
might still be low if those languages do not exhibit significant distinctions with each
other.

The traditional diversity approach does not directly involve the divergence time
provided by the phylogenetic tree for calculation. It simply measures the degree of
distinctions among the observed languages (i.e., linguistic diversity) and assumes that
the homeland of languages should be located in the area possessing the largest
linguistic diversities [3]. Nevertheless, the theoretical foundation of this approach is
somewhat related to the divergence time as the reviewer mentioned. In short, the
diversity approach assumes that early divergence exhibits a higher divergence rate,
which subsequently leads to the birth of an extraordinary number of distinct languages
around the language homeland [3]. However, this theoretical underpinning has always
been criticized because no solid evidence has been proposed to link divergence rate
and homeland location. Additionally, other population activities, such as the migration
of native speakers out of their original homeland, could also alter the linguistic
diversity of the language homeland [4].

Following the reviewer's suggestions, we have added more detailed descriptions
of the linguistic diversity approach in *Lines 308-311* of the revised main text.
Moreover, a more comprehensive discussion of the linguistic diversity approach can

be found in Supplementary Note 2: Section 2.

**Reference**

[1] Sapir, Edward. Time perspective in aboriginal American culture: a study in
method. No. 13. Government Printing Bureau, 1916.

[2] Wichmann, Søren, and Taraka Rama. "Testing methods of linguistic homeland
detection using synthetic data." *Philosophical Transactions of the Royal Society B*
376.1824 (2021): 20200202.

[3] Wichmann, Søren, André Müller, and Viveka Velupillai. "Homelands of the
world's language families: A quantitative approach." *Diachronica* 27.2 (2010):
247-276.

[4] Neureiter, Nico, et al. "Can Bayesian phylogeography reconstruct migrations and
expansions in linguistic evolution?." *Royal Society open science* 8.1 (2021): 201079.

*Q6: I noticed in Supplementary Note 1 that phylogenetic discussions of*
*Austroasiatic, Japonic and Oceanic are still mentioned, even through these*
*groupings are no longer discussed in the main text.*

**Replies to Q6:**

We express our appreciation to the reviewer for bringing these points to our
attention. In the revision, we have deleted the discussions related to the Austroasiatic,
Japonic, and Oceanic languages in Supplementary Note 1.

*Q7: Supplementary Notes 2: it is not clear to me that Supplementary sections 2 and*
*3 are really necessary (The interdisciplinary alignment of Genetics, Archaeology,*
*and Linguistics; The Age-Area Hypothesis for inferring the language homeland). I*
*think the observations made in this paper can stand quite well without them.*

**Replies to Q7:**

We sincerely appreciate the reviewer's suggestions. Following these suggestions,
we have made several revisions to the main text and Supplementary Note 2. To be
specific, we have excluded Section 3 (i.e., The Interdisciplinary Alignment of
Genetics, Archaeology, and Linguistics) from Supplementary Note 2. After careful
consideration, we have decided to retain Section 2 within Supplementary Note 2. This
decision is motivated by the fact that the diversity approach is another famous
phylogeny-free approach for identifying the language dispersal center. In our study,
we have undertaken empirical comparisons between our approach and this
methodology. As a result, Section 2 of Supplementary Note 2 offers an invaluable
complement to the main text, providing readers with a more comprehensive grasp of
the underlying rationale and limitations of the diversity approach.

**Replies to Reviewer 3:**

*Q1: I find this study generally quite interesting, since the authors claim that they*
*have developed a new method that allows to represent historical dynamics of*
*individual languages in comparison with neighboring languages by*
*multidimensional vectors, which can then be projected in lower-dimensional space*
*in order to even infer the original locations from which the language family as a*
*whole dispersed. While interesting, I see some general problems with the study,*
*mainly its fit with the journal where it was submitted to, and as a result, I*
*recommend it to be rejected -- not because it is too low in quality, but rather because*
*it is not a good fit with the journal, as I'll explain below. Apart from this, I see some*
*major and minor flaws, which I'll discuss below. First, regarding the fit of the*
*approach: What the authors propose is a methodological study, a new methodology*
*of which they claim it outperforms established -- albeit controversial -- methods. In*
*such a case, the journal where they submitted their study to, does not really qualify*
*as a good fit, since we do not deal with new findings (they cannot be made until the*
*method has been thoroughly evaluated) but rather with a new method that needs to*
*be shown to work. For this reason, I think some journal like "Nature Methods"*
*would be a much better fit here.*

**Replies to Q1:**

We are genuinely grateful for the reviewer's recommendation regarding the
potential fit of our manuscript with *Nature Methods*, which is another outstanding
Nature-branded journal renowned for its specialization in novel methods. Nonetheless,
we firmly maintain our conviction that our work is ideally suited for *Nature*
*Communications*.

Firstly, *Nature Communications* stands as a top-rank multidisciplinary journal
that is devoted to publishing high-quality research in all interdisciplinary areas. Apart
from reporting novel discoveries, it also has published many papers that propose
novel methods to address interesting scientific questions. Specifically, diverse
velocity field-based methods applicable to various research fields have been published
in *Nature Communications*. These velocity fields have contributed to inferring the
trajectories of dynamic changes in natural and social systems such as single-cell
differentiation [1-2], human mobility [3], and atmospheric circulation [4].

Accordingly, we think that our paper, which proposes a novel velocity field-based
method to infer the language dispersal trajectory, is also suitable to the aim and scope
of *Nature Communications*.

Secondly, although our paper presents a new computational approach, its essence
remains firmly rooted in multidisciplinary exploration. Our study seeks to investigate
the spatial alignment of linguistic, genetic, and archaeological evidence in
reconstructing prehistoric population activities worldwide. We believe that this topic
could spark broad interest among researchers devoted to the interdisciplinary studies
of human prehistory. It should also meet the aim and scope of *Nature*
*Communications*.

**Reference:**

[1] Gao, Mingze, Chen Qiao, and Yuanhua Huang. "UniTVelo: temporally unified
RNA velocity reinforces single-cell trajectory inference." *Nature Communications*
13.1 (2022): 6586.

[2] Riba, Andrea, et al. "Cell cycle gene regulation dynamics revealed by RNA
velocity and deep-learning." *Nature Communications* 13.1 (2022): 2865.

[3] Mazzoli, Mattia, et al. "Field theory for recurrent mobility." *Nature*
*communications* 10.1 (2019): 3895.

[4] Sohn, Byung-Ju, et al. "Regulation of atmospheric circulation controlling the
tropical Pacific precipitation change in response to CO2 increases." *Nature*
*communications* 10.1 (2019): 1108.

*Q2: Second, if the authors accept that they need to convince us first that their*
*method is useful and will enlarge our future knowledge about the spread of*
*language families over time, they should please provide their method in a way that*
*it can be replicated. As of now, we have a bunch of unrelated, badly documented*
*R-scripts in a folder of 600 MB, that are hard to read and even harder to*
*understand. Where is the vector estimation happening, what is the k you choose for*
*the k-means languages that you select as neighbors, what is the impact of k on your*
*results, etc. It makes me extremely nervous to see such a huge bunch of barely*
*commented R-scripts that often do the same, but bear another name of another*
*language family. This is definitely not how you make a new method successful. The*
*least we would expect is a package in R with a tutorial that runs us through your*
*code, for one language family, and then an extended tutorial with all four language*
*families.*

**Replies to Q2:**

We are grateful for the reviewer's suggestion. It greatly enhances the readability
of our R codes and the convenience of the replications and utilizations of our
approach by other users. Following the reviewer's suggestions, we have built an R
package and provided some detailed tutorials on this package. Please see
[https://github.com/Stan-Sizhe-Yang/Language-velocity-field-estimation-for-language](https://github.com/Stan-Sizhe-Yang/Language-velocity-field-estimation-for-language-dispersal-pattern-inference)
[-dispersal-pattern-inference.](https://github.com/Stan-Sizhe-Yang/Language-velocity-field-estimation-for-language-dispersal-pattern-inference)

*Q3: Third, speaking of four, I hate to say this, but I was reviewing this study before,*
*not negatively, but pointing to the code, and to other issues. Interestingly, the*
*number of language families has now dropped from 7 to 4. How the heck did that*
*happen? How do the authors explain that they discard three language families now?*
*I know having the same reviewers for the same paper across journals is annoying,*
*but please, good scientific practice requires you to be transparent and tell us what*
*happened here. Did you discard them, because they did not bring the results you*
*hoped for?*

**Replies to Q3:**

We appreciate the reviewer for pointing this out. Moreover, we are deeply

grateful for the reviewer to dedicate valuable personal time to review our manuscript
again. As mentioned by the reviewer, the previous version of our manuscript
contained seven cases: Sino-Tibetan, Indo-European, Bantu, Arawak, Japonic,
Austroasiatic, and Oceanic languages. However, for this version submitted to *Nature*
*Communications*, we have excluded three language cases: Japonic, Austroasiatic, and
Oceanic languages.

**The primary reason for dropping three language cases.** We dropped these
three language cases due to the lack of language samples around their suggested
language homelands. To be specific, the proposed homelands of Indo-European,
Sino-Tibetan, Bantu, and Arawak languages are situated in geographic ranges where
sufficient language samples can be found [1-4]. However, there lack of sufficient
language samples within the geographic areas covering the suggested homelands of
Japonic (West Liao River of China [5-7]), Oceanic (Taiwan of China [6-8]), and
Austroasiatic (Southern China [6]) languages respectively. Due to the lack of
available language samples, it is nearly possible to determine the homelands of these
three language cases in China solely based on the geographic coordinates of their
language samples observed today. Accordingly, we can solely reconstruct the parts of
their complete dispersal histories. The estimated results of these three language cases
are described as follows.

**The estimated results of three dropped language cases.** (i) The Japonic
languages are regarded as the branch of the Trans-Eurasian languages [5]. Our
approach traced their dispersal originating from the Honshu, followed by spread
northward and southward across Japan. This dispersal pattern is in accordance with
the expansion of the Trans-Eurasian languages from the Korean peninsula into Japan
archipelago [5-7]. (ii) The Oceanic languages are a branch of the Austronesian
languages [8]. We estimated their dispersal from the region near Southern Halmahera
Island with subsequent eastward expansion across the Pacific settlement. The
Southern Halmahera Island region is located at the easternmost edge of the
geographic range of Oceanic language samples. Therefore, the estimated Oceanic
dispersal pattern is compatible with the expansion of the Oceanic branch of the
Austronesian language in the Pacific settlement [6-8]. (iii) For the Austroasiatic
languages, our approach inferred their dispersal from the Mekong River region (one
of the agricultural homelands in Mainland Southeast Asia), with subsequent
expansion throughout Mainland Southeast Asia. This result favors the “Riverine

hypothesis” proposed by Sidwell [9].

Overall, due to a lack of sufficient language samples, the inferred dispersal
patterns of Japonic, Oceanic, and Austroasiatic languages can only reflect a portion of
their complete dispersal histories respectively. And, the estimated dispersal centers of
these languages may be the secondary centers that are formed after they diffused into
their current observed geographic ranges. Therefore, these three cases are unable to
depict the full picture of their corresponding language dispersal patterns and illustrate
the full power of our approach. More importantly, retaining these three language cases
in our manuscript would make our narrative less clear which would potentially
confuse the readers. In the version submitted to *Nature Communications*, we therefore
decided to drop these three more troublesome cases.

**Reference:**

[1] Bouckaert, Remco, et al. "Mapping the origins and expansion of the
Indo-European language family." *Science* 337.6097 (2012): 957-960.

[2] Zhang, Menghan, et al. "Phylogenetic evidence for Sino-Tibetan origin in
northern China in the Late Neolithic." *Nature* 569.7754 (2019): 112-115.

[3] Grollemund, Rebecca, et al. "Bantu expansion shows that habitat alters the route
and pace of human dispersals." *Proceedings of the National Academy of Sciences*
112.43 (2015): 13296-13301.

[4] Walker, Robert S., and Lincoln A. Ribeiro. "Bayesian phylogeography of the
Arawak expansion in lowland South America." *Proceedings of the Royal Society B:
Biological Sciences* 278.1718 (2011): 2562-2567.

[5] Robbeets, Martine, et al. "Triangulation supports agricultural spread of the
Transeurasian languages." *Nature* 599.7886 (2021): 616-621.

[6] Diamond, Jared, and Peter Bellwood. "Farmers and their languages: the first
expansions." *science* 300.5619 (2003): 597-603.

[7] Skoglund, Pontus, and Iain Mathieson. "Ancient genomics of modern humans:
the first decade." *Annual review of genomics and human genetics* 19 (2018): 381-404.

[8] Gray, Russell D., Alexei J. Drummond, and Simon J. Greenhill. "Language
phylogenies reveal expansion pulses and pauses in Pacific settlement." *science*
323.5913 (2009): 479-483.

[9] Paul, Sidwell. "The Austroasiatic central riverine hypothesis." *Вопросы*
*языкового родства* 16 (59) (2010): 117-134.

*Q4: Fourth, the claim of the method not using phylogenetic information is a bit*
*exaggerated: we know geography correlates often with language relatedness (see*
*for example here: <https://doi.org/10.1371/journal.pone.0265460>), so if geography*
*explains the tree, you cannot say you do not use the tree if you use geography as a*
*proxy for the construction of your vectors.*

**Replies to Q4:**

Thank you for your comments. At first, we fully agree with the reviewer that
language geography usually strongly correlates with linguistic relatedness. The
languages with closer geographic locations often possess higher relatedness due to
either vertical divergence or horizontal contact. This connection guarantees the
viability of various methods to reconstruct the dispersal pattern of languages based on
linguistic relatedness, such as the phylogeographic approach [1] and our language
velocity field estimation approach. To be specific, both the phylogeographic approach
and our approach initially delineate the diachronic evolutionary trajectories of
linguistic traits that shape the observed linguistic relatedness. Subsequently, based on
the correlation between linguistic relatedness and language geography, these
evolutionary diachronic evolutionary trajectories are transformed into language
dispersal trajectories.

Secondly, we would like to emphasize that our approach necessitates the
phylogenetic information, but this phylogenetic information is not represented by the
phylogenetic tree. To be specific, phylogenetic information or linguistic relatedness is
not identical to the phylogenetic tree. It is noted that linguistic relatedness can be
shaped by both vertical divergence and horizontal contact. The phylogenetic tree is
just one of the models utilized to extract and represent the part of the linguistic
relatedness of languages solely resulting from vertical divergence [2]. In our approach,

we do not utilize the phylogenetic tree but a more general approach—the PCA
algorithm to measure the linguistic relatedness through the distances among languages
in a two-dimensional PC space (PCA-based distance). In the PC space, the languages
exhibiting more linguistic relatedness resulting from either vertical divergence or
horizontal contacts are intended to be distributed closer. Accordingly, if the linguistic
relatedness is solely attributed to vertical divergence, the PCA-based distance should
be able to capture the phylogenetic information similar to that of the phylogeographic
tree.

Thirdly, in our approach, we first depict the diachronic evolutionary trajectories
of linguistic traits that shape the observed linguistic relatedness within the PC space.
Based on the correlation between linguistic relatedness and language geography, we
subsequently transform these diachronic evolutionary trajectories into language
dispersal trajectories. Accordingly, we actually utilize language geography to
approximate the linguistic relatedness for constructing the velocity field. Although the
linguistic relatedness can be partially captured by the phylogenetic tree, it does not
mean that our approach adopts the topological structure of the phylogenetic tree as
input data used in our computational approach. **However, if the linguistic
relatedness can be adequately captured by the phylogenetic tree, the
phylogenetic information distilled by our approach should be similar to that
distilled by the phylogenetic tree. Under this circumstance, our approach can be
somehow regarded as utilizing the phylogenetic tree as well. In contrast, if
linguistic relatedness bears more influence from horizontal contacts, our
approach cannot be regarded as utilizing the phylogenetic tree. This conclusion
has been verified in the revised main text (*Lines 210-303*).**

Reference

[1] Bouckaert, Remco, et al. "Mapping the origins and expansion of the
Indo-European language family." *Science* 337.6097 (2012): 957-960.

[2] François, Alexandre. "Trees, waves and linkages: Models of language
diversification." *The Routledge handbook of historical linguistics*. Routledge, 2015.
161-189.

*Q5: Fifth, the question of homeland has always been problematic, but if you*
*already use data by Wichmann and Rama, you should also check the much simpler*
*baseline published in Glottolog by now*
*(www.pyglottolog.readthedocs.io/en/latest/homelands.html#module-pyglottolog.homelands).*
*This method seems to work as well as the one by Wichmann and Rama, but*
*it is even simpler, so I would say there's one more baseline to be tested.*

**Replies to Q5:**

We highly value these insightful suggestions. Therefore, we compared our
approach—language velocity field estimation (LVF) to two other baseline approaches
suggested by the reviewer. These comparisons were achieved based on 1,000
simulated datasets and 4 empirical datasets. These two baseline approaches are
referred to as “centroid (Centr)” and “minimal distance (MD)” approaches. The Centr
approach postulates that the center of the polygon formed by the extension of current
language geographic locations should be the dispersal center. The MD approach posits
that the location of the language that exhibits the smallest average geographic distance
to the other languages should be the dispersal center.

**1. Simulated validations for baseline approaches.**

It is noted that the simulated datasets are generated by applying a random walk
model to the phylogenetic tree given a set of predefined dispersal centers. Accordingly,
we have already known the true dispersal centers in these simulated datasets. Utilizing
these simulated datasets provided by Wichmann et al., we first verified whether Centr
and MD approaches can effectively estimate the predefined dispersal center. By
applying Centr and MD approaches to the simulated datasets, we computed the errors
in terms of longitude and latitude respectively between the true and estimated
dispersal centers (Figure 1a to Q5). For either Centr or MD approaches, the outcomes
of the Wilcoxon rank-sum test demonstrated that the errors between true and
estimated dispersal centers were not significantly different from zero in both terms of
longitude and latitude (p -value > 0.05; Figure 1a to Q5). It indicates that there is no
difference between the dispersal centers estimated by either Centr or MD approaches
and the true ones, thus affirming the high effectiveness of both Centr and MD
approaches.

**2. Simulated comparisons between LVF and baseline approaches.**

After justifying the effectiveness of the Centr and MD approach, we further
compared the performance of LVF within these two approaches respectively based on
1,000 simulated datasets. It is noted that the effectiveness of the LVF had already been
verified using these simulated datasets in our previous manuscript. Therefore, we
anticipated that LVF should exhibit the same performance as the Centr and MD
approaches in simulated applications. Noting these, we calculated the differences in
terms of longitude and latitude between the dispersal centers estimated by LVF and
these two approaches respectively (Figure 1b to Q5). According to the Wilcoxon
rank-sum test, we indeed found no significant differences in terms of longitude and
latitude between the dispersal center estimated by LVF and those estimated by these
two approaches respectively (p -value > 0.05; Figure 1b to Q5). This result confirms
that LVF exhibits identical performance as these two baseline approaches in simulated
applications.

**3. Empirical comparisons between LVF and baseline approaches.**

We proceeded to compare the performance between LVF and baseline approaches
in empirical applications. However, we found significant differences between the
dispersal centers estimated by LVF and those estimated by these two baseline
approaches (Figure 2 to Q5). Moreover, it appeared that the estimated dispersal
centers of Centr and MD approaches seemed to lack support from the genetic and
archeological evidence and were well less aligned with linguistics' conventional
intuitions. In contrast, the estimated results of LVF can be more favored by the
archaeological and genetic evidence, implying the better performance of LVF in
empirical applications as compared to Centr and MD approaches.

**4. The possible reasons why two baseline approaches are useful in simulated** 1431 **validations but not in empirical applications.**

Given the distinctions between the theoretical foundations of LVF and these two
baseline approaches (i.e., Centr and MD), it is not surprising to see such obvious
differences between the estimated result of LVF and those of the two baseline
approaches in empirical applications. The LVF reconstructs language dispersal by
transforming the diachronic evolutionary trajectories of linguistic traits that shape the
observed linguistic relatedness into the language dispersal trajectories. In contrast,
these two baseline approaches rely solely on the geographic locations of language

samples, making their estimated results more susceptible to the biased geographic
distribution of language samples. Nevertheless, these two baseline approaches exhibit
high effectiveness in simulated validations probably owing to that simulated datasets
are generated by the random walk model. The random walk model simulates that
languages diffuse evenly as an outward radiating pattern from a given center.
Accordingly, such simulation may display two characteristics:

(a) The simulated language samples tend to be evenly distributed around this
given dispersal center in the geographic space.

(b) Due to (a), the simulated language samples located closer to the center of
their geographic distribution would have a shorter average geographic
distance to other languages.

Due to these two characteristics, both Centr and MD approaches can exhibit good
performance in identifying the language dispersal center within simulated applications.
Nevertheless, the empirical language samples may be not geographically distributed
around the dispersal center uniformly, due to numerous reasons such as sampling bias,
environmental constraints (i.e., mountain, desert, and river), and population
movement (carrying languages out of the dispersal center) [1-2]. Consequently, Centr
and MD approaches solely relying on the geographic locations of language samples
may not perform as effectively in empirical applications.

**Reference**

[1] Grollemund, Rebecca, et al. "Bantu expansion shows that habitat alters the route
and pace of human dispersals." *Proceedings of the National Academy of Sciences*
112.43 (2015): 13296-13301.

[2] Neureiter, Nico, et al. "Can Bayesian phylogeography reconstruct migrations and
expansions in linguistic evolution?" *Royal Society open science* 8.1 (2021): 201079.

**Figure 1 to Q5. Simulated validations of two baseline approaches and simulated**
 **comparisons between LVF and baseline approaches.** a) density plot shows the
 distribution of the error between the true and estimated dispersal center in terms of
 longitude and latitude. The p-value is calculated based on the Wilcoxon rank-sum test.
 b) density plot shows the distribution of the difference between the dispersal center
 estimated by LVF and baseline approaches in terms of longitude and latitude. The
 p-value is calculated based on the Wilcoxon rank-sum test.

**Figure 2 to Q5. The dispersal centers estimated by LVF, Centr, and MD**

**approaches for four language families and groups.**

*Q6: And when speaking of testing: why restrict your study to four datasets (or*
*seven), if there are many more available in terms of phylogenies now, which are all*
*with nicely coded cognate sets in standardized data formats (see e.g.,*
*<https://doi.org/10.1038/s41597-022-01432-0> for a very large collection of*
*standardized data)? It seems the data has been cherry-picked to yield good results.*
*Taking ten of the datasets in the Lexibank collection should not be difficult and*
*would tell us much more clearly where we are with this new method.*

**Replies to Q6:**

We express our sincere gratitude to the reviewer for introducing the *Lexibank*
which is an important lexical dataset to us. The *Lexibank* covers nearly 3,000
language samples of around 300 language families and groups around the world. This
lexical dataset could provide comprehensive insights into the origins and dispersals of
various language families and groups around the world.

The primary objective of our paper is to examine the alignment of language
dispersal, demic diffusion, and Neolithic/Agricultural cultures spread in human
prehistory. Therefore, the language cases utilized in our paper are expected to fulfill
the following criteria. Firstly, the language case should have a possible association
with the origin and development of ancient agriculture. Secondly, the demic or
cultural diffusions in the specific geographic areas where these languages are spoken
should be supported by corresponding genetic or archaeological evidence. Thirdly, the
language cases are preferably renowned cases with sufficient language samples that
have been rigorously investigated in previous phylogenetic research. More
importantly, the lexical items in these language cases should have been carefully
collated and well coded into cognate sets that meet the standard of computational
linguistics. With these criteria, we hope that the empirical cases can better serve our
paper's primary objective and make our estimated results more acceptable to the
broad range of audiences.

According to these criteria, four language cases which are Indo-European,
Sino-Tibetan, Bantu, and Arawak languages are included in our study. These

languages are hypothesized to be closely associated with agricultural development in
this area [1-2]. Moreover, they are widely spoken in their corresponding geographic
area and have all been rigorously studied by former phylogenetic studies [3-6]. More
importantly, the lexical items utilized in these four cases have undergone careful
selections and validations. In the geographic areas where the languages are spoken,
the demic diffusion and cultural spread have been delineated based on sufficient
genetic or archaeological evidence [1-2].

Following these criteria, within the *Lexibank*, we filtered out the language cases
with a sample size lower than 20, ultimately leaving us with 17 language cases. These
cases are *Afro-Asiatic*, *Arawak*, *Atlantic-Congo*, *Austroasiatic*, *Austronesian*,
*Hmong-Mien*, *Indo-European*, *Nuclear Trans New Guinea*, *Pama-Nyungan*,
*Quechuan*, *Sino-Tibetan*, *Dravidian*, *Tucanoan*, *Tupian*, *Turkic*, *Uralic*, and
*Uto-Aztecan languages*. Among them, the *Indo-European*, *Sino-Tibetan*, *Austroasiatic*,
and *Arawak languages* have been incorporated into our study and *Afro-Asiatic* and
*Pama-Nyungan languages* are the hunter-gatherer languages. Additionally, there lack
of sufficient Austronesian language samples within their suggested homeland in China.
Therefore, we ultimately selected 10 language cases: *Uralic*, *Trans-New-Guinea*,
*Quechuan*, *Turkic*, *Tukanoan*, *Tupian*, *Uto-Aztecan*, *Hmong-Mien*, *Atlantic-Congo*,
and *Dravidian languages*.

However, either the evolution or dispersals of these 10 language cases has not
been well investigated and remains highly controversial in the previous computational
linguistic studies. Therefore, investigating their dispersal patterns seems worthy of
being pursued as separate research endeavors for publication. Moreover,
corresponding genetic and archaeological evidence is also hard to find to support the
demic diffusion and cultural spread within the area where these languages are spoken.
Given these constraints, we hold the view that including these 10 language cases in
our study may not align with our primary research objective and make the narrative of
our manuscript less clear. Therefore, we still hope to retain the original well-attested
four language cases (i.e., Indo-European, Sino-Tibetan, Bantu, and Arawak languages)
in our manuscript.

Although we have decided not to include these language cases in our revision, we
still have applied our approach—language velocity field estimation (LVF) to these
language cases to infer their dispersal patterns. In this reply, we present the results

regarding the dispersal patterns of these 10 language cases to the reviewer below
(Table 1 to Q6 and Figure 1 to Q6). The datasets of these 10 language cases and the R
codes for replicating the results of these 10 language cases can be downloaded from
[https://github.com/Stan-Sizhe-Yang/Language-velocity-field-estimation-for-language](https://github.com/Stan-Sizhe-Yang/Language-velocity-field-estimation-for-language-dispersal-pattern-inference)
[-dispersal-pattern-inference](https://github.com/Stan-Sizhe-Yang/Language-velocity-field-estimation-for-language-dispersal-pattern-inference).

**1. Uralic languages**

Uralic languages are widely distributed across northeastern Europe and Northern
Asia. The lexical dataset of Uralic languages was sourced from Honkola et al. (2013)
[7]. The LVF inferred that the dispersal center of Uralic languages is situated in the
steppe region in the southeast of the Ural Mountains (Lon: 64.6, Lat: 54.9) (Figure 1b
to Q6). From this dispersal center, Uralic languages dispersed westward crossing the
Ural Mountains into Europe and eastward into the Far East region. It advocates the
“east of the southern Urals origin hypothesis” of Uralic languages, which is proposed
according to the historical contact between Uralic and Indo-Iranian languages [8].

**2. Trans-New-Guinea languages**

Trans–New Guinea languages are widely spoken on the island of New Guinea
and neighboring islands. The Trans–New Guinea lexical dataset was obtained from
Greenhill (2015) [9]. The LVF depicted the dispersal of Trans-New-Guinea languages
originating from the center in central Papua New Guinea (Lon: 144.3, Lat: -6.4),
which used to be the ancient agricultural homeland of New Guinea island (Figure 1c
to Q6). This result is compatible with the conclusion drawn from recent linguistic
studies and corroborated by the archaeological evidence [10-11]. It suggests that the
Trans–New Guinea dispersal could be closely associated with the development and
spread of agriculture across the New Guinea island.

**3. Quechuan languages**

The Quechuan languages are widely spoken by the native peoples in South
America. We collected the Quechuan lexical dataset from the Blum et al. (2023) [12].
The dispersal center of Quechuan languages (Lon: -75.5, Lat: -9.8) was inferred more
adjacent to the Lima near the Andes which is the ancient agricultural homeland in
South America [13] (Figure 1d to Q6). From this dispersal center, Quechuan
languages spread northward and southward along the Andes. These results are

compatible with the evidence drawn from the Quechua dialectology [14].

**4. Turkic languages**

Turkic languages span the vast expanse of the Eurasian continent, stretching from
the northwest of China to the west of Eastern Europe, and from the north of Siberia to
the south of Iran. The precise homeland of Turkic languages remains a subject of
intense debate. The expansive geographic area encompassing the Transcaspian steppe
to the far northeastern reaches of Manchuria in Asia is regarded as a potential
homeland for these languages [15]. We applied LVF to the Turkic lexical dataset
structured by Savelyev et al. (2020) [16]. The spatial reconstruction showed that
Turkic languages spread westward into Europe and eastward into the Far East region
from the dispersal center inferred in Kazakhstan near Mongolia and Southern Siberia
(Lon: 77.1, Lat: 54.4) (Figure 1e to Q6). This result can be advocated by the genetic
evidence that suggests the potential origin of Turkic-speaking populations in the area
near Mongolia and Southern Siberia [15]. However, we noticed that the Turkic
language samples manifested an exceedingly sparse geographic distribution across the
Eurasian continent. Such sparse geographic distribution may introduce more
uncertainties into the LVF estimation. Therefore, collecting more Turkic language
samples may enable LVF to yield a more precise depiction of the Turkic dispersal
pattern.

**5. Tukanoan languages**

Tukanoan, also referred to as Tucanoan, is a language family of Colombia, Brazil,
Ecuador, and Peru in South America. We applied the LVF to the Tucanoan dataset
derived from Chacon et al. (2017) [17]. The dispersal center of Tucanoan languages
was inferred in the region of the Japurá River (Lon: -70.0, Lat: -0.9) (Figure 1f to Q6).
The location of this dispersal center is compatible with the conclusion drawn from
previous linguistic studies and can be advocated by the archaeological evidence
[17-18].

**6. Tupian languages**

The Tupian language family is one of the largest linguistic groups in South
America. The dataset of the Tupian language was sourced from Galucio et al. (2015).
We applied LVF to this dataset for inferring the dispersal pattern of Tupian languages.

The result showed that Tupian languages dispersed from the center located in the
regions of Rondônia in Brazil within the Madeira River basin (Lon: -62.3, Lat: -11.6)
across South America. This result is compatible with previous linguistic studies [19]
(Figure 1g to Q6).

**7. Uto-Aztecan languages**

The Uto-Aztecan languages are the mother tongue of native Americans, which are
primarily spoken in the Great Basin region, including states such as California,
Nevada, and Arizona, and extending into Mexico. The Uto-Aztecan lexical dataset
was derived from the Greenhill (2023) [20]. The LVF identified the dispersal center of
Uto-Aztecan languages in Southern Arizona (Lon: -113.5, Lat: 33.9) near the border
between Arizona and Mexico (Figure 1h to Q6). This location was compatible with
the one inferred by the phylogeographic approach as reported in Greenhill (2023)
(Lon: -116.7, Lat: 34.8). From this dispersal center, the Uto-Aztecan languages spread
southeastward and northwestward along the coastline, and northeast into South
America. These results favor the “Northern origin hypothesis” supported by the
reconstruction of flora and fauna terms [20-21]. This hypothesis postulates that
Uto-Aztecan languages originated in the area between Southern California’s Mojave
Desert and the Sonoran and Chihuahuan desert regions of Arizona and northern
Mexico.

**8. Hmong-Mien languages**

The Hmong-Mien languages are primarily spoken by various ethnic groups in
southern China, northern Vietnam, Laos, Thailand, and Myanmar. Linguistic
reconstructions focusing on ancient terminology related to flora and fauna have
suggested that the origins of Hmong-Mien languages might be found in the provinces
to the south of the Yangzi River [22]. In our investigation, we applied the LVF to the
Hmong-Mien lexical dataset derived from Chen (2013) [23] (Figure 1i to Q6). The
results consistently indicated that the dispersal center of Hmong-Mien languages is
indeed located within Guizhou province, situated to the south of the Yangzi River
(Longitude: 107.7, Latitude: 27.0).

**9. Atlantic-Congo languages**

The Atlantic-Congo languages, which constitute a prominent subgroup of the

Niger-Congo language family, have a significant presence across the African
 continent. The Atlantic-Congo lexical dataset was collected from the public dataset
 compiled by Koelle (1853) [24]. Utilizing the LVF, we traced the dispersal of
 Atlantic-Congo languages initiating from Nigeria near Cameroon (Lon: 5.6, Lat: 6.4),
 which used to be the ancient agricultural homeland in Africa [25] (Figure 1j to Q6). It
 suggests that the Atlantic-Congo dispersal could be associated with agricultural
 expansion in Africa.

10. Dravidian languages

The Dravidian languages are widely scattered across southern and central India
 and surrounding countries. The dispersal of Dravidian languages has been a
 long-standing debate. The genetic evidence indicates the potential origin of Dravidian
 languages in the Indus Valley, with subsequent southward and eastward expansion
 across the Indian subcontinent [26]. The linguistic evidence drawn from the term
 reconstruction suggests that Dravidian languages might originate somewhere in South
 India (i.e., Peninsular India) [26]. Archaeological evidence yields the connection
 between the origin of the Dravidian language and the development of the Southern
 Neolithic complex in Karnataka and Andhra Pradesh [27, 28]. Based on the Dravidian
 lexical dataset derived from Kolipakam et al. (2018) [29], LVF inferred the dispersal
 of Dravidian languages originating from the center located in the range of Andhra
 Pradesh (Lon: 80.6, Lat: 13.6) (Figure 1k to Q6). This result can be supported by the
 archaeological evidence that implies the close association between Dravidian
 dispersal and Neolithic culture spread in India.

Table and Figure

**Table 1 to Q6.** The coordinates of dispersal centers inferred by LVF for ten language
 families and groups.

Language	Longitude	Latitude
Uralic	64.6	54.9
Trans-New-Guinea	144.3	-6.4
Quechuan	-75.5	-9.8
Turkic	77.1	54.4
Tukanoan	-70.0	-0.9

Tupian	-62.3	-11.6
Uto-Aztecan	-113.5	33.9
Hmong-Mien	107.7	27.0
Atlantic-Congo	5.6	6.4
Dravidian	80.6	13.6

**Figure 1 to Q6.** The Language velocity fields reveal the dispersal patterns of 10

language families and groups worldwide. The red dot denotes the dispersal center
inferred by LVF. The pink dot signifies the language sample. The black arrow
represents the grid-smoothed velocity vector.

**Reference**

[1] Diamond, Jared, and Peter Bellwood. "Farmers and their languages: the first
expansions." *science* 300.5619 (2003): 597-603.

[2] Skoglund, Pontus, and Iain Mathieson. "Ancient genomics of modern humans:
the first decade." *Annual review of genomics and human genetics* 19 (2018): 381-404.

[3] Bouckaert, Remco, et al. "Mapping the origins and expansion of the
Indo-European language family." *Science* 337.6097 (2012): 957-960.

[4] Zhang, Menghan, et al. "Phylogenetic evidence for Sino-Tibetan origin in
northern China in the Late Neolithic." *Nature* 569.7754 (2019): 112-115.

[5] Grollemund, Rebecca, et al. "Bantu expansion shows that habitat alters the route
and pace of human dispersals." *Proceedings of the National Academy of Sciences*
112.43 (2015): 13296-13301.

[6] Walker, Robert S., and Lincoln A. Ribeiro. "Bayesian phylogeography of the
Arawak expansion in lowland South America." *Proceedings of the Royal Society B:
Biological Sciences* 278.1718 (2011): 2562-2567.

[7] Honkola, Terhi, et al. "Cultural and climatic changes shape the evolutionary
history of the Uralic languages." *Journal of Evolutionary Biology* 26.6 (2013):
1244-1253.

[8] Nichols, Johanna. "The origin and dispersal of Uralic: Distributional typological
view." *Annual Review of Linguistics* 7 (2021): 351-369.

[9] Greenhill, Simon J. "TransNewGuinea. org: An online database of New Guinea
languages." *PLoS One* 10.10 (2015): e0141563.

[10]Diamond, Jared, and Peter Bellwood. "Farmers and their languages: the first
expansions." *science* 300.5619 (2003): 597-603.

- [11]Schapper, Antoinette. "Farming and the Trans-New Guinea family." *Language*
dispersal beyond farming (2017): 155-181.
- [12]Blum, Frederic, et al. "A phylolinguistic classification of the Quechua language
family." (2023).
- [13]Diamond, Jared, and Peter Bellwood. "Farmers and their languages: the first
expansions." *science* 300.5619 (2003): 597-603.
- [14]King, Kendall A., and Nancy H. Hornberger. "Quechua as a lingua franca."
*Annual Review of Applied Linguistics* 26 (2006): 177-196.
- [15]Yunusbayev, Bayazit, et al. "The genetic legacy of the expansion of
Turkic-speaking nomads across Eurasia." *PLoS Genetics* 11.4 (2015): e1005068.
- [16]Savelyev, Alexander, and Martine Robbeets. "Bayesian phylolinguistics infers the
internal structure and the time-depth of the Turkic language family." *Journal of*
*Language Evolution* 5.1 (2020): 39-53.
- [17]Chacon, Thiago. "Arawakan and Tukanoan contacts in Northwest Amazonia
prehistory." *PAPIA Rev. Bras. Estud. Crioulos E Similares* 27 (2017): 237-265.
- [18]Chacon, Thiago. "On Proto-Languages and Archaeological Cultures: pre-history
and material culture in the Tukanoan Family." *Revista Brasileira de Linguística*
*Antropológica* 5.1 (2013): 217-245.
- [19]Galucio, Ana Vilacy, et al. "Genealogical relations and lexical distances within
the Tupian linguistic family." *Boletim do Museu Paraense Emílio Goeldi. Ciências*
*Humanas* 10 (2015): 229-274.
- [20]Greenhill, Simon J., et al. "A recent northern origin for the Uto-Aztecan family."
*Language* (2023).
- [21]Campbell, Lyle. "What drives linguistic diversification and language spread."
*Examining the farming/language dispersal hypothesis* (2002): 49-63.
- [22]Ratliff, Martha. *Hmong-Mien language history*. Pacific Linguistics, Research
School of Pacific and Asian Studies, The Australian National University, 2010.

[23]Qiguang, Chen. "Miao-Yao yuwen [Miao and Yao Language]." Beijing:
Zhongyang Minzu Daxue Chubanshe (2013).

[24]Koelle, Sigismund Wilhelm. "Polyglotta Africana; or a comparative vocabulary
of nearly three hundred words and phrases in more than one hundred distinct African
languages." (No Title) (1853).

[25]Diamond, Jared, and Peter Bellwood. "Farmers and their languages: the first
expansions." science 300.5619 (2003): 597-603.

[26] Narasimhan, Vagheesh M., et al. "The formation of human populations in South
and Central Asia." Science 365.6457 (2019): eaat7487.

[27]Southworth, Franklin. Linguistic archaeology of south Asia. Routledge, 2004.

[28]Kolipakam, Vishnupriya, et al. "A Bayesian phylogenetic study of the Dravidian
language family." Royal Society open science 5.3 (2018): 171504.

[29]Kolipakam, Vishnupriya, et al. "DravLex: A Dravidian lexical database:(Version
v1. 0.0)[Data set]." (2018).

*Q7: Sixth, the method has the rather infelicitous name "language velocity field*
*estimation", and I could not find any explanation why the authors chose to call it*
*like that, since the name is very confusion and difficult to parse, and it does not*
*really help to understand what the method could be about. I think in general it*
*would be useful to 1) change the name to something that explains the method in a*
*better way (dynamic trait vectors? I am not sure) and 2) to explain the method in*
*much, much more detail. For this, figures would be needed that show how vectors*
*for some of the traits are estimated, and the authors would need to also check the*
*resulting vectors on an individual basis in order to see if they make sense.*

**Replies to Q7:**

We are sorry for not being clear about the rationale of our approach. After careful
consideration, we have decided to retain the original name "language velocity field"
of our approach. Because this name can intuitively reflect the characteristics of our

approach. Following the valuable *suggestion 2*) offered by the reviewer, we have
redrawn our original schematic diagrams for the rationale and calculation procedure
of our approach with greater detail and accuracy as shown in Figure 1 of the revised
main text. For the convenience of the reviewer, we attach Figure 1 of the revised main
text to the end of this reply as Figure 1 to Q7. Additionally, we have added more
detailed descriptions of our approach into the *Lines 109-151* of the revised main text.
Considering the word limit in the main text, more detailed explanations of our
approach can be found in Supplementary Note 1. Here, we provide a concise
explanation of the rationale of our approach.

**The inspiration for proposing language velocity field estimation.** The velocity
field can be visualized as a collection of arrows with given magnitudes and directions
estimated by a specific dynamic model, which demonstrates the directions of the
spatiotemporal changes of individuals [1]. The directions of the vectors in the velocity
field compose sets of continuously changing paths that visualize the dynamic
trajectories of natural phenomena such as atmospheric circulation [2] (e.g., water
vapor transport), and cell differentiation [3] (e.g., RNA transcription). Furthermore,
this approach has now extended to infer the trajectories of the spatial-temporal
changes of social phenomena such as demic diffusion [4] (e.g., human mobility), and
cultural spread [5] (e.g., Neolithic culture propagation). Given that humans are the
carriers of languages which are also the carriers of cultures, we believe that the
velocity field could also contribute to the inference of the language dispersal.
Accordingly, our approach is designed to establish a language velocity field on the
geographic map to depict language dispersal patterns. By visualizing the language
velocity field on the geographic map, the directions of velocity vectors can intuitively
show how and from where (i.e., dispersal trajectory and center) these languages have
dispersed into their current locations.

**Our approach shares the same theoretical foundation as the phylogeographic**
**approach but with different implementation strategies.** As the most prevailing
approach, the phylogeographic approach implements two major steps to infer
language dispersal from the diachronic evolution of linguistic traits [6]. The first is to
establish a phylogenetic tree to depict the diachronic evolutionary trajectories of
linguistic traits that shape the observed linguistic relatedness (Figure 2 to Q7). The
second is to project the phylogenetic tree into the geographic space to transform these
diachronic evolutionary trajectories into dispersal trajectories, based on the correlation

between linguistic relatedness and geography (Figure 2 to Q7). Akin to the
phylogeographic approach, our approach also infers language dispersal through the
diachronic evolution of linguistic traits with two major steps (Figure 2 to Q7). The
first is to establish a velocity field to depict the diachronic evolutionary trajectories of
linguistic traits that shape the observed linguistic relatedness. The second is to project
this velocity field into the geographic space to outline the language dispersal
trajectories. These two steps are described as follows.

**The velocity field in PC space delineates diachronic evolutionary trajectories**
**of linguistic traits that shape the observed linguistic relatedness.** Our approach
conducts the PCA-based distance rather than a phylogenetic tree to represent
linguistic relatedness. To be specific, we employ the PCA algorithm to extract two
optimal principal components (i.e., PC1 and PC2) from the linguistic traits. According
to PC1 and PC2, we represent the linguistic relatedness among language samples as
the distances among them in the PC space that can be shaped by both divergence and
contact (Figure 1b to Q7). In parallel, we use a dynamic model, similar to the
widely-used covarion model for linguistic trait evolution [7-9], to reconstruct the past
states of linguistic traits for each language sample (Figure 1d1 to Q7). Given the
differences between the past and current trait states of each language sample, we can
obtain a velocity vector that reflects the direction of diachronic changes in its
linguistic traits (Figure 1d2 to Q7). In other words, the velocity vector depicts how the
linguistic traits in each language sample evolve into their current states. Finally, we
project this language velocity field into the PC space formed by the aforementioned
two principal components (Figure 1e to Q7). For convenience, we can interpret the
language velocity field in the PC space as the collection of arrows connecting the past
and current states of linguistic traits within language samples in the PC space (Figure
1e1 to Q7). Accordingly, the past and current states of linguistic traits within language
samples can simultaneously be visualized in the PC space. Each arrow connecting the
past and current states of linguistic traits for each language sample outlines the
diachronic change of the linguistic traits in this language. Therefore, the arrows in the
PC space compose a set of trajectories to depict the diachronic evolution of the
linguistic traits that shape the observed linguistic relatedness (Figure 1e2 to Q7).

**Transforming the diachronic evolutionary trajectories of the linguistic traits**
**into language dispersal trajectories.** We project the language velocity field from the
PC space to the geographic space based on the correlation between linguistic

relatedness and geography [6-8] (Figure 1f to Q7). To achieve this, we utilize the
kernel projection approach proposed by La Manno et al. [3] to project the language
velocity field from the PC space into the two-dimensional geographic space. The
rationale behind this kernel projection is to estimate the velocity vectors of language
samples in the geographic space, ensuring that their correlation with language
distributions in the PC space can be best preserved within the geographic space
(Figure 1f1 to Q7). This projection is similar to the projection of the phylogenetic tree
to the geographic space in the phylogeographic approach. Accordingly, the directions
of these vectors compose a set of trajectories that depict from where the observed
language samples have diffused into their current locations (Figure 1f2 to Q7). We
hope these contents supplemented by Figure 1 to Q7 can provide the reviewer with a
clearer understanding of our approach.

**Validation of velocity field.** The direction of the ultimate velocity vector of a
language sample we estimated within the geographic space manifests the direction
from where this language sample diffuses into its current locations. However, it is
important to highlight that the power of any spatial reconstruction method is
inevitably affected by the heterogeneity of the spatial distribution of samples.
Therefore, each estimated velocity vector cannot signify exactly the diffusion
direction of each language sample. However, our approach aims to reconstruct the
general dispersal pattern of the entire language family or group rather than the exact
dispersal direction of just one language sample. Moreover, relying solely on a single
velocity vector is insufficient to ascertain the dispersal pattern of the entire language
family. And, the overall dispersal pattern of the entire language family is deduced by
the continuously changing trajectories formed by a collection of velocity vectors.
Consequently, it appears less critical to validate the effectiveness of a solitary velocity
vector on the individual level. Accordingly, we consider that the effectiveness of our
approach should be validated on the global level of the language velocity field rather
than the individual level of a single language velocity vector. Under this circumstance,
simulated validations of our approach have confirmed its ability to reconstruct
accurate language dispersal patterns based on the language velocity field in our
previous manuscript. Therefore, with these simulated validations, we believe that the
velocity vectors can indeed contribute to reconstructing the language dispersal
pattern.

**Reference**

- [1] Galbis, Antonio, and Manuel Maestre. Vector analysis versus vector calculus.
Springer Science & Business Media, 2012.
- [2] Sohn, Byung-Ju, et al. "Regulation of atmospheric circulation controlling the
tropical Pacific precipitation change in response to CO2 increases." Nature
communications 10.1 (2019): 1108.
- [3] La Manno, Gioele, et al. "RNA velocity of single cells." Nature 560.7719 (2018):
494-498.
- [4] Mazzoli, Mattia, et al. "Field theory for recurrent mobility." Nature
communications 10.1 (2019): 3895.
- [5] Fort, Joaquim. "Demic and cultural diffusion propagated the Neolithic transition
across different regions of Europe." Journal of the Royal Society interface 12.106
(2015): 20150166.
- [6] Bouckaert, Remco, et al. "Mapping the origins and expansion of the
Indo-European language family." Science 337.6097 (2012): 957-960.
- [7] Yang, Ziheng. "Maximum-likelihood estimation of phylogeny from DNA
sequences when substitution rates differ over sites." Molecular biology and evolution
10.6 (1993): 1396-1401.
- [8] Penny, David, et al. "Mathematical elegance with biochemical realism: the
covarion model of molecular evolution." Journal of Molecular Evolution 53 (2001):
711-723.
- [9] Zhang, Menghan, et al. "Phylogenetic evidence for Sino-Tibetan origin in
northern China in the Late Neolithic." Nature 569.7754 (2019): 112-115.

**Figure**

**Figure 1 to Q7. Schematic diagram of language velocity field estimation (LVF)**
 **for inferring the dispersal trajectories and centers of languages.** The
 computational procedures of the LVF comprise two major steps. Subfigures (a) to (e)
 illustrate the first step which is to estimate a velocity field on the PC space to outline
 the diachronic evolutionary trajectories of linguistic traits that shape the observed
 linguistic relatedness. Subfigures (f) to (g) illustrate the second step, which is to
 project the velocity field from PC space into geographic space. Within the velocity
 field in geographic space, the directions of the velocity vectors compose a set of
 continuously changing trajectories that delineate from where these languages diffuse
 to their current locations. These procedures are exemplified using the Bantu language
 family. Comprehensive insights into the underlying principles and computational
 steps can be found in the Materials and Methods section, as well as Supplementary
 Note 1.

**Figure 2 to Q7. Language velocity field estimation (LVF) shares the same**
**foundation as the phylogeographic approach but with different implementation**
**strategies.** Both LVF and phylogeographic approach entails two major steps to infer
language dispersal pattern. The first is to depict the diachronic evolutionary
trajectories of linguistic traits that shape the observed linguistic relatedness. The
second is to transform these diachronic evolutionary trajectories of linguistic traits
into language dispersal trajectories. In the phylogenetic tree, each language is
determined by k linguistic traits. In the velocity field within PC space, each language
is determined by PC1 and PC2 which are rearranged from the k linguistic traits
through the PCA algorithm. The red number denotes a language. The black arrow
signifies the evolutionary direction of linguistic traits in a language. The blue arrow
represents the dispersal direction of a language. The red star denotes the estimated
dispersal center.

*Q8: Seventh, the authors praise their method for not needing trees, but at the same*
*time, they do not tell the readers why trees are so useful: they tell us various*
*scenarios of character evolution in a very transparent way, in which we have*
*scenario and can plot how the trait evolved. Of course, this is not always done, but*
*they should tell the readers to which the method they propose allows us to get some*
*insights into the black box, since a simple black box, even if it works, is not*
*satisfying from a scientific viewpoint, and we talk about scientific approaches here.*

**Replies to Q8:**

We really appreciate the reviewer for raising this crucial point. To improve the
credibility and interpretability of our approach, we have added more comprehensive
descriptions and explanations of our approach to the revised main text (*Lines*
*109-151*). Here, we offer a brief answer.

**1. The phylogenetic tree visualizes the diachronic evolutionary trajectories of**
**the linguistic traits that shape the observed linguistic relatedness.**

The phylogeographic approach infers the language dispersal through the
diachronic evolution of linguistic traits. As the reviewer mentioned, the phylogenetic
tree plays an important role in the phylogeographic approach. To be specific, the
phylogenetic tree is a power representation for the diachronic evolutionary trajectories
of the linguistic traits that shape the observed linguistic relatedness (Figure 1 to Q8).
This representation relies on the branching pattern within the phylogenetic tree. This
branching pattern visualizes the diachronic evolution of linguistic traits in languages
after diverging from their ancestors [1]. The shorter branch linking two languages
indicates fewer diachronic changes occurring between their traits, resulting in a higher
linguistic relatedness between them. This phylogenetic tree can be projected into the
geographic space based on the correlation between linguistic relatedness and language
geography (Figure 1 to Q8) [1-2]. To be specific, each branch within the phylogenetic
tree, that has been projected into the geographic space, is regarded as a segment of the
dispersal trajectories (Figure 1 to Q8). With this projection, the evolutionary
trajectories of linguistic traits can thus be transformed into language dispersal
trajectories.

**2. The theoretical foundation and interpretability of our approach.**

Akin to the phylogeographic approach, our approach also aims to reconstruct the
language dispersal pattern through the diachronic evolution of linguistic traits. Our
approach and phylogeographic approach actually share the same theoretical
foundation but with different implementation strategies (Figure 1 to Q8).

**The velocity field in PC space depicts the diachronic evolutionary**
**trajectories of the linguistic traits that shape the observed linguistic relatedness.**

Our approach represents the linguistic relatedness of observed language samples
through the distances among them in a two-dimensional PC space instead of a
phylogenetic tree. This PC space is determined by two optimal axes (PC1 and PC2)
estimated through the PCA algorithm (Figure 2b to Q7). In this PC space, the
language samples with higher relatedness, due to both divergence and contact, would
be distributed closer. In parallel, we reconstruct the past states of linguistic traits for
each language sample using a dynamic model that is derived from the widely-used
covarion model for linguistic trait evolution [3-5] (Figure 2d to Q7). Subsequently, we
also project these past trait states onto the PC space. Accordingly, both past and
current states of linguistic traits for each language sample can be visualized in the PC
space. By computing the differences between the current and past trait states divided
by the reconstruction time for each language sample in the PC space, we can derive a
velocity vector representing the diachronic changes of its linguistic traits (Figure 2e1
to Q7). In other words, this velocity vector illustrates how the linguistic traits in this
language sample evolve into their current states. Accordingly, these velocity vectors
consist of a velocity field in the PC space. And, this velocity field outlines a set of
trajectories that represent the diachronic change of linguistic traits that shape the
observed linguistic relatedness (Figure 2e2 to Q7).

**Transforming the diachronic evolutionary trajectories of the linguistic traits**
**into language dispersal trajectories.**

Subsequently, we adopt the kernel projection
proposed by La Manno et al. to map the velocity field from PC space into the
geographic space. This projection seeks the velocity vector in the geographic space
ensuring that its correlation with language geography aligns closely with its
correlation with linguistic relatedness (Figure 2f1 to Q7). This projection is similar to
the projection of each branch within the phylogenetic into the geographic space as a
segment of dispersal trajectories (Figure 1 to Q8). With the kernel projection, the
velocity vectors compose a set of trajectories in geographic space that depict from
where the observed language samples have diffused into their current locations

(Figure 2f2 to Q7).

**The relationship between the phylogeographic approach and our approach.**

It is noted that if linguistic relatedness can be adequately demonstrated by the
phylogenetic tree, our approach and phylogenetic tree can capture similar linguistic
relatedness. Accordingly, our approach and phylogeographic approach would exhibit
the same performance. In contrast, if linguistic relatedness cannot be adequately
demonstrated by the phylogenetic tree, our approach can capture additional
phylogenetic information from linguistic relatedness due to horizontal contacts as
compared to the phylogeographic approach. Accordingly, our approach may derive a
more reliable result than the phylogeographic approach. In summary, our approach
can be seen as an extension of the phylogeographic approach by relaxing its tree
topology assumption of the phylogeographic approach. This conclusion has been
verified in the revised main text (*Lines 210-303*). Therefore, our approach does not
stand as the opposite of the phylogeographic approach but as its extension.

**Figure**

**Figure 1 to Q8. Language velocity field estimation (LVF) shares the same**
 **foundation as the phylogeographic approach but with different implementation**
 **strategies.** Both LVF and phylogeographic approach entails two major steps to infer
 language dispersal pattern. The first is to depict the diachronic evolutionary
 trajectories of linguistic traits that shape the observed linguistic relatedness. The
 second is to transform these diachronic evolutionary trajectories of linguistic traits
 into language dispersal trajectories. In the phylogenetic tree, each language is
 determined by k linguistic traits. In the velocity field within PC space, each language
 is determined by PC1 and PC2 which are rearranged from the k linguistic traits
 through the PCA algorithm. The red number denotes a language. The black arrow
 signifies the evolutionary direction of linguistic traits in a language. The blue arrow
 represents the dispersal direction of a language. The red star denotes the estimated
 dispersal center.

**Figure 2 to Q8. Schematic diagram of language velocity field estimation (LVF)**
 **for inferring the dispersal trajectories and centers of languages.** The
 computational procedures of the LVF comprise two major steps. Subfigures (a) to (e)
 illustrate the first step which is to estimate a velocity field on the PC space to outline
 the diachronic evolutionary trajectories of linguistic traits that shape the observed
 linguistic relatedness. Subfigures (f) to (g) illustrate the second step, which is to
 project the velocity field from PC space into geographic space. Within the velocity
 field in geographic space, the directions of the velocity vectors compose a set of
 continuously changing trajectories that delineate from where these languages diffuse
 to their current locations. These procedures are exemplified using the Bantu language
 family. Comprehensive insights into the underlying principles and computational
 steps can be found in the Materials and Methods section, as well as Supplementary
 Note 1.

**Reference**

[1] Bouckaert, Remco, et al. "Mapping the origins and expansion of the
 Indo-European language family." *Science* 337.6097 (2012): 957-960.

[2] Grollemund, Rebecca, et al. "Bantu expansion shows that habitat alters the route
and pace of human dispersals." Proceedings of the National Academy of Sciences
112.43 (2015): 13296-13301.

[3] Yang, Ziheng. "Maximum-likelihood estimation of phylogeny from DNA
sequences when substitution rates differ over sites." Molecular biology and evolution
10.6 (1993): 1396-1401.

[4] Penny, David, et al. "Mathematical elegance with biochemical realism: the
covarion model of molecular evolution." Journal of Molecular Evolution 53 (2001):
711-723.

[5] Zhang, Menghan, et al. "Phylogenetic evidence for Sino-Tibetan origin in
northern China in the Late Neolithic." Nature 569.7754 (2019): 112-115.

*Q9: Eighth, and final point, the paper is not nice to read, the authors should check*
*their wordings, which are often hard to follow, at times with flaws in grammar, and*
*it would really profit from a complete overhaul and a thorough checking by a proof*
*reader.*

**Replies to Q9:**

We really appreciate the reviewer for pointing this out. In the revised main text,
we have corrected all the typos and grammar flaws. And, we have simplified the long
and wording sentences into the concise and shorten ones. Moreover, we have engaged
the AJE language editing service to thoroughly polish the language of the revised
manuscript (ID: Q2K9ZRSF). We hope that our revised manuscript can be more
readable to native English speakers.

*Q10: Due to all these reservations, I recommend that the paper be rejected, but I*
*emphasize that it is not for poor quality, but for lack of fit. I look forward to see a*
*new methods paper emerging from this, in which the authors work hard to share a*
*useful new approach with the scientific world that they also evaluate rigorously*
*against existing approaches. I am convinced they have the potential to turn their*
*paper into such a study, and I am also very confident that this would be the right*
*way to go, instead of trying to sell this as some study with new insights, or a study*
*with a method that beats all existing approaches, since this is obviously not the*
*case.*

**Replies to Q10:**

We appreciate these comments and are very grateful for the reviewer's
encouragement. According to the reviewer's suggestions, we have carefully rewritten
the contents about the validations of the approach and the comparison with other
approaches. Moreover, we have added a more detailed description of the rationale of
our approach. As supplementary, we have also redrawn the schematic diagram to
more visually demonstrate the rationale and procedure of our approach. Most
importantly, we have restructured the logical flow of our paper, with a focus on
sharing a useful and rigorously validated approach with the science community.

Reviewers' Comments:

Reviewer #1:

Remarks to the Author:

The authors answered all my concerns and I do not have further major comments.

Minor changes that need to be addressed:

- new figure 1: typo in panel b, "langauge"
- Panel c: cordiniate, algrithm
- Panel f: unclear sentence + writings in orange are too small and cannot be read

Reviewer #2:

Remarks to the Author:

I think this is perhaps the fourth time I have reviewed this article. As I stated before, I am neither statistician nor linguist, but I detect that the authors have replied to all previous comments by the referees to the maximum extent possible. So I am happy to see the article go to press.

I am impressed by the authors' claims for their efficacy of their "language velocity field" method (at least for the 4 examples they consider), based on PCA rather than phylogenetic "family tree" distances between language subgroups, even if my understanding of all the algebraic formulae that they present is rather limited. The main point for me is that the conclusions of the authors with respect to the homelands of 4 language families that they consider are virtually identical to those I offer in my two recent books *The Five-Million-Year Odyssey* (Princeton 2022) and *First Farmers* (second edition, Wiley Blackwell 2023).

So I wish the authors the best of luck with publication and scholarly reception of their views.

Reviewer #3:

Remarks to the Author:

Dear Authors. I have now read all your comments and also had a look at the revised paper and I decided that I should no longer stand in the way, preventing your study to be published. What I would like to ask you, however, is one final thing: For transparency and for replicability, please make sure to make a RELEASE of your code on GitHub and please download this release and submit it to an open independent repository that guarantees long-term archival, such as, for example, Zenodo or Open Science Framework. Here, you will receive a DOI and you should add this DOI to your paper, so we can check the very same code you used to produce the final results that you share with us. Since GitHub itself is owned by Microsoft and Microsoft could shut it down any time they please (think of what happened to Twitter), we need to have the data and the code in public hands. This should not be too hard to do for you, so I hope you'll account for it quickly, and I will recommend the publication of your study, once these changes have been made.

As I will ask for my reports to be published along with my name, I emphasize that the fact that I agree with the publication of this study does not mean that I explicitly express full confidence in its results. It rather means that I feel that it is the best if this study is at this point shared with a larger public that can then discuss then findings in due course and may well find that they have some flaws which were overseen during the review process. I myself am not able to find these flaws by now, nor am I able to assess the quality of the study in full, due to the specifics of my own background. But I am confident that this study provides an interesting contribution to the field and therefore deserves to be published and discussed by more qualified colleagues than myself.

Response Letter to Reviewers

Replies to Reviewer 1:

Q1: The authors answered all my concerns and I do not have further major comments. Minor changes that need to be addressed: new figure 1: typo in panel b, "langauge"

Replies to Q1:

We sincerely appreciate your careful examination. We have corrected this typo in the revised manuscript.

Q2: Panel c: cordiniate, algorithm

Replies to Q2

These typos have been corrected in the revised manuscript.

Q3: Panel f: unclear sentence + writings in orange are too small and cannot be read

Replies to Q3

We sincerely appreciate your comments. We have enlarged the texts in orange to ensure that they can be read clearly by the readers.

Replies to Reviewer 2:

Q1: I think this is perhaps the fourth time I have reviewed this article. As I stated before, I am neither statistician nor linguist, but I detect that the authors have replied to all previous comments by the referees to the maximum extent possible. So I am happy to see the article go to press. I am impressed by the authors' claims for their efficacy of their "language velocity field" method (at least for the 4 examples they consider), based on PCA rather than phylogenetic "family tree" distances between language subgroups, even if my understanding of all the algebraic formulae that they

*present is rather limited. The main point for me is that the conclusions of the authors with respect to the homelands of 4 language families that they consider are virtually identical to those I offer in my two recent books *The Five-Million-Year Odyssey* (Princeton 2022) and *First Farmers* (second edition, Wiley Blackwell 2023). So I wish the authors the best of luck with publication and scholarly reception of their views.*

Replies to Q1

We sincerely appreciate your support and affirmation all the time. Moreover, we are also very grateful for your recommendation of your two excellent books to us. We believe that the evidence mentioned within these books can greatly enhance the credibility of our conclusions.

Replies to Reviewer 3:

*Q1: Dear Authors. I have now read all your comments and also had a look at the revised paper and I decided that I should no longer stand in the way, preventing your study to be published. What I would like to ask you, however, is one final thing: For transparency and for replicability, please make sure to make a **RELEASE** of your code on GitHub and please download this release and submit it to an open independent repository that guarantees long-term archival, such as, for example, Zenodo or Open Science Framework. Here, you will receive a DOI and you should add this DOI to your paper, so we can check the very same code you used to produce the final results that you share with us. Since GitHub itself is owned by Microsoft and Microsoft could shut it down any time they please (think of what happened to Twitter), we need to have the data and the code in public hands. This should not be too hard to do for you, so I hope you'll account for it quickly, and I will recommend the publication of your study, once these changes have been made.*

Replies to Q1

We are deeply grateful for your support and encouragement. Your valuable suggestions and comments have greatly improved the quality of our manuscript and the transparency and replicability of our approach. Following your suggestions, we have also uploaded our R package and codes to the Zendo (<https://doi.org/10.5281/zenodo.10223872>).

Q2: As I will ask for my reports to be published along with my name, I emphasize that the fact that I agree with the publication of this study does not mean that I explicitly express full confidence in its results. It rather means that I feel that it is the best if this study is at this point shared with a larger public that can then discuss then findings in due course and may well find that they have some flaws which were overseen during the review process. I myself am not able to find these flaws by now, nor am I able to assess the quality of the study in full, due to the specifics of my own background. But I am confident that this study provides an interesting contribution to the field and therefore deserves to be published and discussed by more qualified colleagues than myself.

Replies to Q2

We sincerely appreciate your support and encouragement. Moreover, we are very grateful that you are willing to publish your reports with your name. We believe that your reports can provoke new thoughts among the readers.